# Cryptographic Hardness of Learning Halfspaces with Massart Noise

**Ilias Diakonikolas**
UW Madison
ilias@cs.wisc.edu

**Daniel M. Kane**
UC San-Diego
dakane@ucsd.edu

**Pasin Manurangsi**
Google Research
pasin@google.com

**Lisheng Ren**
UW Madison
lren29@wisc.edu

## Abstract

We study the complexity of PAC learning halfspaces in the presence of Massart noise. In this problem, we are given i.i.d. labeled examples $(\mathbf{x}, y) \in \mathbb{R}^N \times \{\pm 1\}$, where the distribution of $\mathbf{x}$ is arbitrary and the label $y$ is a Massart corruption of $f(\mathbf{x})$, for an unknown halfspace $f : \mathbb{R}^N \to \{\pm 1\}$, with flipping probability $\eta(\mathbf{x}) \leq \eta < 1/2$. The goal of the learner is to compute a hypothesis with small 0-1 error. Our main result is the first computational hardness result for this learning problem. Specifically, assuming the (widely believed) subexponential-time hardness of the Learning with Errors (LWE) problem, we show that no polynomial-time Massart halfspace learner can achieve error better than $\Omega(\eta)$, even if the optimal 0-1 error is small, namely $\mathrm{OPT} = 2^{-\log^c(N)}$ for any universal constant $c \in (0, 1)$. Prior work had provided qualitatively similar evidence of hardness in the Statistical Query model. Our computational hardness result essentially resolves the polynomial PAC learnability of Massart halfspaces, by showing that known efficient learning algorithms for the problem are nearly best possible.

## 1 Introduction

A halfspace or linear threshold function (LTF) is any function $h_{\mathbf{w},t} : \mathbb{R}^N \to \{\pm 1\}$ of the form $h_{\mathbf{w},t}(\mathbf{x}) := \mathrm{sign}(\langle \mathbf{w}, \mathbf{x} \rangle - t)$, where the vector $\mathbf{w} \in \mathbb{R}^N$ is called the weight vector, $t \in \mathbb{R}$ is called the threshold, and $\mathrm{sign} : \mathbb{R} \to \{\pm 1\}$ is defined by $\mathrm{sign}(t) = 1$ if $t \geq 0$ and $\mathrm{sign}(t) = -1$ otherwise. Halfspaces are a central concept class in machine learning, extensively investigated since the 1950s [Ros58, Nov62, MP68]. Here we study the computational complexity of learning halfspaces in Valiant's (distribution independent) PAC model [Val84], when the labels have been corrupted by *Massart noise* [MN06]. We define the Massart noise model below.

**Definition 1.1** (Massart Noise). We say that a joint distribution $D$ of labeled examples $(\mathbf{x}, y)$, supported on $\mathbb{R}^N \times \{\pm 1\}$, satisfies the Massart noise condition with noise parameter $\eta \in [0, 1/2)$ with respect to a concept class $C$ of Boolean-valued functions on $\mathbb{R}^N$ if there is a concept $c \in C$ such that for all $\mathbf{x}_0 \in \mathbb{R}^N$ we have that $\eta(\mathbf{x}_0) \stackrel{\text{def}}{=} \mathbf{Pr}_{(\mathbf{x},y)\sim D}[c(\mathbf{x}) \neq y \mid \mathbf{x} = \mathbf{x}_0] \leq \eta$.

The Massart PAC learning problem for the concept class $C$ is the following: Given i.i.d. samples from a Massart distribution $D$, as in Definition 1.1, the goal is to output a hypothesis with small 0-1 error. In this work, we study the computational complexity of the Massart PAC learning problem, when the underlying concept class $C$ is the class of halfspaces on $\mathbb{R}^N$.

In its above form, the Massart noise model was defined in [MN06]. An essentially equivalent noise model had been defined in the 80s by Sloan and Rivest [Slo88, RS94, Slo96], and a very similar definition had been considered even earlier by Vapnik [Vap82].

The Massart model is a classical semi-random noise model that is more realistic than Random Classification Noise (RCN) In contrast to RCN, Massart noise allows for variations in misclassification

36th Conference on Neural Information Processing Systems (NeurIPS 2022).

rates (without a priori knowledge of which inputs are more likely to be misclassified). Asymmetric misclassification rates arise in a number of applications, including in human annotation noise [BK09]. Consequently, learning algorithms that can tolerate Massart noise are less brittle than those that depend on the uniformity of RCN. The agnostic model [Hau92, KSS94], where the noise can be fully adversarial, is of course even more robust; unfortunately, it is computationally hard to obtain agnostic learners with any non-trivial guarantees, even for basic settings.

We now return to the class of halfspaces, which is the focus of this work. We recall that PAC learning halfspaces with RCN is known to be solvable in polynomial time (to any desired accuracy) [BFKV96]. On the other hand, agnostic PAC learning of halfspaces is known to computationally hard (even for weak learning) [GR06, FGKP06, Dan16].

The computational task of PAC learning halfspaces corrupted by Massart noise is a classical problem in machine learning theory that has been posed by several authors since the 1980s [Slo88, Coh97, Blu03]. Until recently, no progress had been made on the efficient PAC learnability of Massart halfspaces. [DGT19] made the first algorithmic progress on this problem: they gave a $\text{poly}(N, 1/\epsilon)$-time learning algorithm with error guarantee of $\eta + \epsilon$. Subsequent work made a number of refinements to this algorithmic result, including giving an efficient proper learner [CKMY20] and developing an efficient learner with strongly polynomial sample complexity [DKT21]. In a related direction, [DIK+21] gave an efficient boosting algorithm achieving error $\eta + \epsilon$ for any concept class, assuming the existence of a weak learner for the class.

The error bound of $\eta$ can be very far from the information-theoretically optimum error of OPT, where $\text{OPT} = R_{\text{LTF}}(D) \leq \eta$. Indeed, known polynomial-time algorithms only guarantee error $\approx \eta$ even if OPT is very small, i.e., $\text{OPT} \ll \eta$. This prompts the following question:

**Question 1.1.** *Is there an efficient learning algorithm for Massart halfspaces with a* relative *error guarantee? Specifically, if* $\text{OPT} \ll \eta$ *is it possible to achieve error significantly better than* $\eta$*?*

Our main result (Theorem 1.2) answers this question in the negative, assuming the subexponential-time hardness of the classical Learning with Errors (LWE) problem (Assumption 2.4). In other words, *we essentially resolve the efficient PAC learnability of Massart halfspaces*, under a widely-believed cryptographic assumption.

## 1.1 Our Results

Before we state our main result, we recall the setup of the Learning with Errors (LWE) problem. In the LWE problem, we are given samples $(\mathbf{x}_1, y_1), \ldots, (\mathbf{x}_m, y_m)$ and the goal is to distinguish between the following two cases: (i) Each $\mathbf{x}_i$ is drawn uniformly at random (u.a.r.) from $\mathbb{Z}_q^n$, and there is a hidden secret vector $\mathbf{s} \in \mathbb{Z}_q^n$ such that $y_i = \langle \mathbf{x}_i, \mathbf{s} \rangle + z_i$, where $z_i \in \mathbb{Z}_q$ is discrete Gaussian noise (independent of $\mathbf{x}_i$); (ii) Each $\mathbf{x}_i$ and each $y_i$ are independent and are sampled u.a.r. from $\mathbb{Z}_q^n$ and $\mathbb{Z}_q$ respectively. Formal definitions of LWE (Definition 2.3) and related distributions together with the precise computational hardness assumption (Assumption 2.4) we rely on are given in Section 2.

Our main result can now be stated as follows:

**Theorem 1.2** (Informal Main Theorem). *Assume that LWE cannot be solved in* $2^{n^{1-\Omega(1)}}$ *time. Then, for any constant* $\zeta > 0$*, there is no polynomial-time learning algorithm for Massart halfspaces on* $\mathbb{R}^N$ *that can output a hypothesis with 0-1 error smaller than* $\Omega(\eta)$*, even when* $\text{OPT} \leq 2^{-\log^{1-\zeta} N}$ *and the Massart noise parameter* $\eta$ *is a small positive constant.*

The reader is also referred to Theorem D.1 in the Appendix for a more detailed formal statement. Theorem 1.2 is the first *computational* hardness result for PAC learning halfspaces (and, in fact, any non-trivial concept class) in the presence of Massart noise. Our result rules out even *improper* PAC learning, where the learner is allowed to output any polynomially evaluatable hypothesis. As a corollary, it follows that the algorithm given in [DGT19] is essentially the best possible, even when assuming that OPT is almost inverse polynomially small (in the dimension $N$). We also remark that this latter assumption is also nearly the best possible: if OPT is $o(\epsilon/N)$, then we can just draw $\Omega(N/\epsilon)$ samples and output any halfspace that agrees with these samples.

We note that a line of work has established qualitatively similar hardness in the Statistical Query (SQ) model [Kea98] — a natural, yet restricted, model of computation. Specifically, [CKMY20] established a super-polynomial SQ lower bound for learning within error of $\text{OPT} + o(1)$. Subse-

quently, [DK22] gave a near-optimal super-polynomial SQ lower bound: their result rules out the existence of efficient SQ algorithms that achieve error better than $\Omega(\eta)$, even if $\mathrm{OPT} = 2^{\log^{1-\varsigma} N}$. Building on the techniques of [DK22], more recent work [NT22] established an SQ lower bound for learning to error better than $\eta$, even if $\mathrm{OPT} = 2^{\log^{1-\varsigma} N}$ — matching the guarantees of known algorithms exactly. While the SQ model is quite broad, it is also restricted. That is, the aforementioned prior results do not have any implications for the class of all polynomial-time algorithms. Interestingly, as we will explain in the proceeding discussion, our computational hardness reduction is inspired by the SQ-hard instances constructed in [DK22].

## 1.2 Brief Technical Overview

Here we give a high-level overview of our approach. Our reduction proceeds in two steps. The first is to reduce the standard LWE problem (as described above) to a different "continuous" LWE problem more suitable for our purposes. In particular, we consider the problem where the $\mathbf{x}$ samples are taken uniformly from $\mathbb{R}^n/\mathbb{Z}^n$, $y$ is either taken to be an independent random element of $\mathbb{R}/\mathbb{Z}$ or is taken to be $\langle \mathbf{x}, \mathbf{s} \rangle$ mod 1 plus a small amount of (continuous) Gaussian noise, where $\mathbf{s}$ is some unknown vector in $\{\pm 1\}^n$. This reduction follows from existing techniques [Mic18a, GVV22].

The second step — which is the main technical contribution of our work — is reducing this continuous LWE problem to that of learning halfspaces with Massart noise. The basic idea is to perform a rejection sampling procedure that allows us to take LWE samples $(\mathbf{x}, y)$ and produce some new samples $(\tilde{\mathbf{x}}, \tilde{y})$. We will do this so that if $y$ is independent of $\mathbf{x}$, then $\tilde{y}$ is (nearly) independent of $\tilde{\mathbf{x}}$; but if $y = \langle \mathbf{x}, \mathbf{s} \rangle +$ noise, then $\tilde{y}$ is a halfspace of $\tilde{\mathbf{x}}$ with a small amount of Massart noise. An algorithm capable of learning halfspaces with Massart noise (with appropriate parameters) would be able to distinguish these cases by learning a hypothesis $h$ and then looking at the probability that $h(\tilde{\mathbf{x}}) \neq \tilde{y}$. In the case where $\tilde{y}$ was a halfspace with noise, this would necessarily be small; but in the case where $\tilde{\mathbf{x}}$ and $\tilde{y}$ were independent, it could not be.

In order to manage this reduction, we will attempt to produce a distribution $(\tilde{\mathbf{x}}, \tilde{y})$ similar to the SQ-hard instances of Massart halfspaces constructed in [DK22]. These instances can best be thought of as instances of a random variable $(\mathbf{x}', y')$ in $\mathbb{R}^n \times \{\pm 1\}$, where $y'$ is given by a low-degree polynomial threshold function (PTF) of $\mathbf{x}'$ with a small amount of Massart noise. Then, letting $\tilde{\mathbf{x}}$ be the Veronese map applied to $\mathbf{x}'$, we see that any low-degree polynomial in $\mathbf{x}'$ is a linear function of $\tilde{\mathbf{x}}$, and so $\tilde{y} = y'$ is an LTF of $\tilde{\mathbf{x}}$ plus a small amount of Massart noise.

As for how the distribution over $(\mathbf{x}', y')$ is constructed in [DK22], essentially the conditional distribution of $\mathbf{x}'$ on $y' = 1$ and on $y' = -1$ are carefully chosen mixtures of discrete Gaussians in the $\mathbf{v}$-direction (for some randomly chosen unit vector $\mathbf{v}$), and independent standard Gaussians in the orthogonal directions. () Our goal will be to find a way to perform rejection sampling on the distribution $(\mathbf{x}, y)$ to produce a distribution of this form.

In pursuit of this, for some small real number $b$ and some $a \in [0, b)$, we let $\mathbf{x}'$ be a random Gaussian subject to $\mathbf{x}' \equiv b\mathbf{x} \pmod{b}$ (in the coordinate-wise sense) conditioned on $by \equiv a \pmod{b}$. We note that if we ignore the noise in the definition of $y$, this implies that $\langle \mathbf{x}', \mathbf{s} \rangle \equiv \langle b\mathbf{x}, \mathbf{s} \rangle \equiv b \langle \mathbf{x}, \mathbf{s} \rangle \equiv by \equiv a \pmod{b}$ (recalling that $\mathbf{s} \in \{\pm 1\}^n$). In fact, it is not hard to see that the resulting distribution on $\mathbf{x}'$ is close to a standard Gaussian conditioned on $\langle \mathbf{x}', \mathbf{s} \rangle \equiv a \pmod{b}$. In other words, $\mathbf{x}'$ is close to a discrete Gaussian with spacing $b/\|\mathbf{s}\|_2$ and offset $a/\|\mathbf{s}\|_2$ in the $\mathbf{s}$-direction, and an independent standard Gaussian in orthogonal directions. Furthermore, this $\mathbf{x}'$ can be obtained from $(\mathbf{x}, y)$ samples by rejection sampling: taking many samples until one is found with $by \equiv a \pmod{b}$, and then returning a random $\mathbf{x}'$ with $\mathbf{x}' \equiv b\mathbf{x} \pmod{b}$. By taking an appropriate mixture of these distributions, we can manufacture a distribution close to the hard instances in [DK22]. This intuition is explained in detail in Section 3.1; see Lemma 3.3. (We note that Lemma 3.3 is included only for the purposes of intuition; it is a simpler version of Lemma 3.5, which is one of the main lemmas used to prove our main theorem.)

Unfortunately, as will be discussed in Section 3.2, applying this construction directly does not quite work. This is because the small noise in the definition of $y$ leads to a small amount of noise in the final values of $\langle \mathbf{x}', \mathbf{s} \rangle$. This gives us distributions that are fairly similar to the hard instances of [DK22], but leads to small regions of values for $\mathbf{u}$, where the following condition holds: $\mathbf{Pr}(y' = +1 \mid \mathbf{x}' = \mathbf{u}) = \mathbf{Pr}(y' = -1 \mid \mathbf{x}' = \mathbf{u})$. Unfortunately, the latter condition cannot hold if $y'$ is a function of $\mathbf{x}'$ with Massart noise. In order to fix this issue, we need to modify the construction

by carving intervals out of the support of $\mathbf{x}'$ conditioned on $y' = -1$, in order to eliminate these mixed regions. This procedure is discussed in detail in Section 3.3.2.

### 1.3 Additional Related Work

There have also been several recent works showing reductions from LWE or lattice problems to other learning problems. Concurrent and independent work to ours [Tie22] showed hardness of weakly agnostically learning halfspaces, based on a worst-case lattice problem (via a reduction from "continuous" LWE). Two recent works obtained hardness for the unsupervised problem of learning mixtures of Gaussians (GMMs), assuming hardness of (variants of) the LWE problem. Specifically, [BRST21] defined a continuous version of LWE (whose hardness they established) and reduced it to the problem of learning GMMs. More recently, [GVV22] obtained a direct reduction from LWE to a (different) continuous version of LWE; and leveraged this connection to obtain quantitatively stronger hardness for learning GMMs. It is worth noting that for the purposes of our reduction, we require as a starting point a continuous version of LWE that differs from the one defined in [BRST21]. Specifically, we require that the distribution on $\mathbf{x}$ is uniform on $[0,1]^n$ (instead of a Gaussian, as in [BRST21]) and the secret vector is binary. The hardness of this continuous version essentially follows from [Mic18b, GVV22].

## 2 Preliminaries

For $\mathbf{x}, \mathbf{s} \in \mathbb{R}^n$ with $\mathbf{s} \neq \mathbf{0}$, let $\mathbf{x}^{\mathbf{s}} \stackrel{\text{def}}{=} \langle \mathbf{x}, \mathbf{s} \rangle / \|\mathbf{s}\|_2$ be the length of the projection of $\mathbf{x}$ in the $\mathbf{s}$ direction, and $\mathbf{x}^{\perp \mathbf{s}} \in \mathbb{R}^{n-1}$ be the projection[1] of $\mathbf{x}$ on the orthogonal complement of $\mathbf{s}$. For $f, g : U \to \mathbb{R}$, we write $f(u) \propto g(u)$ if there is $c \in \mathbb{R}$ such that $f(u) = cg(u)$ for all $u \in U$. We use $X \sim D$ to denote a random variable $X$ with distribution $D$. We use $P_D$ or $P_X$ for the corresponding probability mass function (pmf) or density function (pdf), and $\mathbf{Pr}_D$ or $\mathbf{Pr}_X$ for the measure function of the distribution. We use $D_X$ to denote the distribution of the random variable $X$. For $S \subseteq \mathbb{R}^n$, we will use $\lambda(S)$ to denote the $n$-dimensional volume of $S$. Let $U(S)$ denote the uniform distribution on $S$. For a distribution $D$ on $\mathbb{R}^n$ and $S \subseteq \mathbb{R}^n$, we denote by $D \mid S$ the conditional distribution of $X \sim D$ given $X \in S$. Let $D^{\mathbf{s}}$ (resp. $D^{\perp \mathbf{s}}$) be the distribution of $\mathbf{x}^{\mathbf{s}}$ (resp. $\mathbf{x}^{\perp \mathbf{s}}$), where $\mathbf{x} \sim D$. For distributions $D_1, D_2$, we use $D_1 + D_2$ to denote the pseudo-distribution with measure function $\mathbf{Pr}_{D_1+D_2}(A) = \mathbf{Pr}_{D_1}(A) + \mathbf{Pr}_{D_2}(A)$. For $a \in \mathbb{R}$, let $a\,D$ denote the pseudo-distribution with measure function $a\mathbf{Pr}_D$. On the other hand, let $a \circ D$ denote the distribution of $aX$, where $X \sim D$. We use $D_1 \star D_2$ to denote the convolution of distributions $D_1, D_2$.

We will use $\text{LTF}_N$ for the class of halfspaces on $\mathbb{R}^N$; when $N$ is clear from the context, we may discard it and simply write LTF. For $q \in \mathbb{N}$, we use $\mathbb{Z}_q \stackrel{\text{def}}{=} \{0, 1, \cdots, q-1\}$ and $\mathbb{R}_q \stackrel{\text{def}}{=} [0, q)$. We use $\text{mod}_q : \mathbb{R}^n \mapsto \mathbb{R}_q^n$ to denote the function that applies $\text{mod}_q(\mathbf{x})$ on each coordinate of $\mathbf{x}$.

We use $D_{\mathbb{R}^n, \sigma}^{\mathcal{N}}$ to denote the $n$-dimensional Gaussian distribution with mean $\mathbf{0}$ and covariance matrix $\sigma^2 / (2\pi) \cdot \mathbf{I}_n$ and use $D_\sigma^{\mathcal{N}}$ as a short hand for $D_{\mathbb{R}, \sigma}^{\mathcal{N}}$. In some cases, we will use $\mathcal{N}(\mathbf{0}, \mathbf{I_n})$ for the standard (i.e., zero mean and identity covariance) multivariate Gaussian,

**Definition 2.1** (Partially Supported Gaussian Distribution). For $\sigma \in \mathbb{R}_+$ and $\mathbf{x} \in \mathbb{R}^n$, let $\rho_\sigma(\mathbf{x}) \stackrel{\text{def}}{=} \sigma^{-n} \exp\left(-\pi(\|\mathbf{x}\|_2 / \sigma)^2\right)$. For any countable set $S \subseteq \mathbb{R}^n$, we let $\rho_\sigma(S) \stackrel{\text{def}}{=} \sum_{\mathbf{x} \in S} \rho_\sigma(\mathbf{x})$, and let $D_{S,\sigma}^{\mathcal{N}}$ be the distribution supported on $S$ with pmf $P_{D_{S,\sigma}^{\mathcal{N}}}(\mathbf{x}) = \rho_\sigma(\mathbf{x}) / \rho_\sigma(S)$.

**Definition 2.2** (Discrete Gaussian). For $T \in \mathbb{R}_+, y \in \mathbb{R}$ and $\sigma \in \mathbb{R}_+$, we define the "$T$-spaced, $y$-offset discrete Gaussian distribution with $\sigma$ scale" to be the distribution of $D_{T\mathbb{Z}+y,\sigma}^{\mathcal{N}}$.

**Learning with Errors (LWE)** We use the following definition of LWE, which allows for flexible distributions of samples, secrets, and noises. Here $m$ is the number of samples, $n$ is the dimension, and $q$ is the ring size.

**Definition 2.3** (Generic LWE). Let $m, n, q \in \mathbb{N}$, and let $D_{\text{sample}}, D_{\text{secret}}, D_{\text{noise}}$ be distributions on $\mathbb{R}^n, \mathbb{R}^n, \mathbb{R}$ respectively. In the $\text{LWE}(m, D_{\text{sample}}, D_{\text{secret}}, D_{\text{noise}}, \text{mod}_q)$ problem, we are given $m$ independent samples $(\mathbf{x}, y)$ and want to distinguish between the following two cases: (i) **Alternative**

---

[1] More precisely, let $\mathbf{B}_{\perp \mathbf{s}} \in \mathbb{R}^{n \times (n-1)}$ for the matrix whose columns form an (arbitrary) orthonormal basis for the orthogonal complement of $\mathbf{s}$, and let $\mathbf{x}^{\perp \mathbf{s}} \stackrel{\text{def}}{=} (\mathbf{B}_{\perp \mathbf{s}})^T \mathbf{x}$.

**hypothesis**: $\mathbf{s}$ is drawn from $D_{\text{secret}}$. Then, each sample is generated by taking $\mathbf{x} \sim D_{\text{sample}}, z \sim D_{\text{noise}}$, and letting $y = \text{mod}_q(\langle \mathbf{x}, \mathbf{s} \rangle + z)$; and (ii) **Null hypothesis**: $\mathbf{x}, y$ are independent and each has the same marginal distribution as above.

When a distribution in LWE is uniform over some set $S$, we may abbreviate $U(S)$ merely as $S$. Note that $\text{LWE}(m, \mathbb{Z}_q^n, \mathbb{Z}_q^n, D_{\mathbb{Z}, \sigma}^{\mathcal{N}}, \text{mod}_q)$ to the classical LWE problem.

**Computational Hardness Assumption for LWE** As alluded to earlier, the assumption for our hardness result is the hardness of the (classic) LWE problem, with the parameters stated below.

**Assumption 2.4** (Standard LWE Assumption (see, e.g., [LP11])). *Let $c > 0$ be a sufficiently large constant. For any constant $\beta \in (0, 1)$, $\kappa \in \mathbb{N}$, $\text{LWE}(2^{O(n^\beta)}, \mathbb{Z}_q^n, \mathbb{Z}_q^n, D_{\mathbb{Z}, \sigma}^{\mathcal{N}}, \text{mod}_q)$ with $q \leq n^\kappa$ and $\sigma = c\sqrt{n}$ cannot be solved in $2^{O(n^\beta)}$ time with $2^{-O(n^\beta)}$ advantage.*

We recall that [Reg09, Pei09] gave a polynomial-time *quantum* reduction from approximating (the decision version of) the Shortest Vector Problem (GapSVP) to LWE (with similar $n, q, \sigma$ parameters). Our hardness assumption is the widely believed *sub-exponential* hardness of LWE. We note that the fastest known algorithm for GapSVP takes $2^{O(n)}$ time [ALNS20]. Thus, refuting the conjecture would be a major breakthrough. A similar assumption was also used in [GVV22] to establish computational hardness of learning Gaussian mixtures. Our use of a sub-exponential hardness of LWE is not a coincidence; see Section 4.

As mentioned earlier, we will use a different variant of LWE, where the sample is from $\mathbb{R}_1^n$, the secret is from $\{\pm 1\}^n$, and the noise is drawn from a continuous Gaussian distribution. The hardness of this variant is stated below. The proof, which follows from [Mic18a, GVV22], is deferred to Appendix B.

**Lemma 2.5.** *Under Assumption 2.4, for any $\beta \in (0, 1)$ and $\gamma \in \mathbb{R}_+$, there is no $2^{O(n^\beta)}$ time algorithm to solve $\text{LWE}\left(2^{O(n^\beta)}, \mathbb{R}_1^n, \{\pm 1\}^n, D_{O(n^{-\gamma})}^{\mathcal{N}}, \text{mod}_1\right)$ with $2^{-O(n^\beta)}$ advantage.*

**Decisional Massart Halfspace Problem** For a distribution $D$ on labeled examples and a concept class $C$, we let $R_C(D) \stackrel{\text{def}}{=} \min_{h \in C} \mathbf{Pr}_{(\mathbf{x}, y) \sim D}[h(\mathbf{x}) \neq y]$ be the error of the best classifier in $C$ with respect to $D$. We will prove hardness for the following decision version of learning Massart halfspaces. This will directly imply hardness for the corresponding learning (search) problem.

**Definition 2.6** (Testing Halfspaces with Massart Noise). For $n, N \in \mathbb{N}, \eta, \text{OPT} \in (0, 1/2)$, let $\text{Massart}(m, N, \eta, \text{OPT})$ denote the problem of distinguishing, given $m$ i.i.d. samples from $D$ on $\mathbb{R}^N \times \{\pm 1\}$, between the following two cases: (i) **Alternative hypothesis**: $D$ satisfies the Massart halfspace condition with noise parameter $\eta$ and $R_{\text{LTF}}(D) \leq \text{OPT}$; and (ii) **Null hypothesis**: the Bayes optimal classifier has $c\eta$ error, where $c > 0$ is a sufficiently small universal constant.

## 3 Reduction from LWE to Learning Massart Halfspaces

In this section, we establish Theorem 1.2. Some intermediate technical lemmas have been deferred to the Appendix C. Our starting point is the problem $\text{LWE}(m, \mathbb{R}_1^n, \{\pm 1\}^n, D_\sigma^{\mathcal{N}}, \text{mod}_1)$. Note that, by Lemma 2.5, Assumption 2.4 implies the hardness of $\text{LWE}(m, \mathbb{R}_1^n, \{\pm 1\}^n, D_\sigma^{\mathcal{N}}, \text{mod}_1)$. We will reduce this variant of LWE to the decision/testing version of Massart halfspaces (Definition 2.6).

Our reduction will employ multiple underlying parameters, which are required to satisfy a set of conditions. For convenience, we list these conditions below.

**Condition 3.1.** *Let $n, m, m' \in \mathbb{N}$, $t, \epsilon, \sigma \in \mathbb{R}_+$, $\delta \in (0, 1)$, satisfy: (i) $t/\epsilon$ is a sufficiently large even integer, (ii) $\sigma \leq \sqrt{n}$, (iii) $\frac{1}{t\sqrt{n}} \geq \sqrt{c \log(n/\delta)}$, where $c$ is a sufficiently large universal constant, (iv) $\left(\frac{c'\epsilon}{c''t\sigma}\right)^2 \geq \log(m'/\delta)$, where $c' > 0$ is a sufficiently small universal constant and $c'' > 0$ is a sufficiently large universal constant.*

The main theorem of this work is stated below.

**Theorem 3.2.** *Let $n, m, m' \in \mathbb{N}$, $t, \epsilon, \sigma \in \mathbb{R}_+$, $\epsilon', \delta \in (0, 1)$ satisfy Condition 3.1 and $\eta < 1/2$. Moreover, assume that $m' = c(\epsilon/t)m$, where $c > 0$ is a sufficiently small universal constant and $m(\epsilon/t)^2$ is sufficiently large, and $N = (n + 1)^d$, where $d/(t/\epsilon)$ is sufficiently large. Suppose that there is no $T + \text{poly}(m, N, \log(1/\delta))$-time algorithm for solving $\text{LWE}(m, \mathbb{R}_1^n, \{\pm 1\}^n, D_\sigma^{\mathcal{N}}, \text{mod}_1)$*

with $\epsilon' - O(\delta)$ *advantage. Then there is no $T$ time algorithm for solving* $\mathrm{Massart}(m', N, \eta, \mathrm{OPT})$ *with $2\epsilon'$ advantage, where* $\mathrm{OPT} = \exp(-\Omega(t^4/\epsilon^2))$.

Note that Theorem 3.2, combined with Lemma 2.5, can be easily used to prove Theorem 1.2 (e.g., by plugging in $t = n^{-0.5 - \Theta(\zeta)}, \epsilon = \Theta(n^{-1.5})$ in the above statement); see Appendix D. As such, we devote the remainder of the body of this paper to give an overview to the proof of Theorem 3.2.

**High-level Overview** The starting point of our computational hardness reduction is the family of SQ-hard instances obtained in [DK22]. At a high-level, these instances are constructed using mixtures of "hidden direction" discrete Gaussian distributions, i.e., distributions that are discrete Gaussians in a hidden direction and continuous Gaussians on the orthogonal directions.

In Section 3.1, we note that by using an appropriate rejection sampling procedure on the LWE samples (drawn from the alternative hypothesis), we obtain a distribution very similar to the "hidden direction discrete Gaussian". A crucial difference in our setting is the existence of a small amount of additional "noise". A natural attempt is to replace the discrete Gaussians in [DK22] with the noisy ones obtained from our rejection sampling procedure. This produces problems similar to the hard instances from [DK22]. Unfortunately, the extra noise in our construction means that the naive version of this construction will not work; even with small amounts of noise, the resulting distributions will *not* satisfy the assumptions of a PTF with Massart noise. In Section 3.2, we elaborate on this issue and the modifications we need to make to our construction in order to overcome it. In Section 3.3, we provide the complete construction of our Massart PTF hard instance.

**Overview of the [DK22] SQ-hard Construction** [DK22] showed SQ-hardness for the following hypothesis testing version of the problem (which implies hardness for the learning problem): For an input distribution $D$ on $\mathbb{R}^n \times \{\pm 1\}$, distinguish between the cases where $D$ is a specific distribution $D_{\mathrm{null}}$ in which $\mathbf{x}$ and $y$ are independent or where $D$ belongs to a class of alternative hypothesis distributions $\mathcal{D}_{\mathrm{alternative}}$. In particular, for $D \in \mathcal{D}_{\mathrm{alternative}}$, $y$ will be given by a low-degree PTF in $\mathbf{x}$ with a small amount of Massart noise. As we will be trying to reproduce it, it is important for us to understand this alternative hypothesis distribution. Each distribution in $\mathcal{D}_{\mathrm{alternative}}$ is parameterized by a hidden direction $\mathbf{s} \in S^{n-1}$. We will denote the corresponding distribution by $D_{\mathbf{s}}$. $D_{\mathbf{s}}$ is constructed so that $\mathbf{x}^{\perp\mathbf{s}} \sim D_{\mathbb{R}^{n-1},1}^{\mathcal{N}}$ is independent of $\mathbf{x}^{\mathbf{s}}$ and $y$. This means that we can specify $D_{\mathbf{s}}$ by describing the simpler distribution of $(\mathbf{x}^{\mathbf{s}}, y) \in \mathbb{R} \times \{\pm 1\}$. For $(\mathbf{x}^{\mathbf{s}}, y)$, we have that $y = +1$ with probability $1 - \eta$. The distributions of $\mathbf{x}^{\mathbf{s}}$ conditioned on $y = \pm 1$ are defined to be mixtures of discrete Gaussians as follows:

$$D_{\mathbf{x}^{\mathbf{s}}|(y=+1)} = \frac{1}{\epsilon} \int_0^\epsilon D_{u+(t+u)\mathbb{Z},1}^{\mathcal{N}} du \text{ and } D_{\mathbf{x}^{\mathbf{s}}|(y=-1)} = \frac{1}{\epsilon} \int_{t/2}^{t/2+\epsilon} D_{u+(t+u-t/2)\mathbb{Z},1}^{\mathcal{N}} du . \quad (1)$$

As we noted, both $\mathbf{x}^{\mathbf{s}} \mid (y = +1)$ and $\mathbf{x}^{\mathbf{s}} \mid (y = -1)$ are mixtures of discrete Gaussians. Combining this with the fact that $\mathbf{x}^{\perp\mathbf{s}} \sim \mathcal{N}(n, \mathbf{I}_{n-1})$, this indicates that $\mathbf{x} \mid (y = +1)$ and $\mathbf{x} \mid (y = -1)$ are mixtures of "hidden direction discrete Gaussians" — with different spacing and offset for their support on the hidden direction. These conditional distributions were carefully selected to ensure that $y$ is a Massart PTF of $\mathbf{x}$ with small error. To see why this is, notice that the support of $\mathbf{x}^{\mathbf{s}} \mid (y = +1)$ is $\bigcup_{i \in \mathbb{Z}} [it, it + (i+1)\epsilon]$, while the support of $\mathbf{x}^{\mathbf{s}} \mid (y = -1)$ is $\bigcup_{i \in \mathbb{Z}} [it + t/2, it + t/2 + (i+1)\epsilon]$; both supports are unions of intervals. Consider the implications of this for three different ranges of $\mathbf{x}^{\mathbf{s}}$:

1. For $\mathbf{x}^{\mathbf{s}} \in [-t^2/(2\epsilon), t^2/(2\epsilon)]$, the intervals have lengths in $[0, t/2]$; thus, the $+1$ intervals and the $-1$ intervals do not overlap at all.

2. For $\mathbf{x}^{\mathbf{s}} \in [-t^2/\epsilon, -t^2/(2\epsilon)) \cup (t^2/(2\epsilon), t^2/\epsilon]$, the intervals have lengths in $[t/2, t]$; thus, the $+1$ intervals and the $-1$ intervals overlap, so that their union covers the space. We note that in this case there are gaps between the $+1$ intervals; specifically, there are at most $O(t/\epsilon)$ such gaps.

3. For $\mathbf{x}^{\mathbf{s}} \in (-\infty, -t^2/\epsilon) \cup (t^2/\epsilon, \infty)$, the intervals have lengths in $[t, \infty)$, so the $+1$ intervals cover the space by themselves.

Consider the degree-$O(t/\epsilon)$ PTF $\mathrm{sign}(p(\mathbf{x}))$ such that $\mathrm{sign}(p(\mathbf{x})) = +1$ iff $\mathbf{x}^{\mathbf{s}} \in \bigcup_{i \in \mathbb{Z}} [it, it + (i+1)\epsilon]$.

In particular, $\mathrm{sign}(p(\mathbf{x})) = 1$ for $\mathbf{x}$ in the support of the conditional distribution on $y = 1$. Note that the PTF $\mathrm{sign}(p(\mathbf{x}))$ has zero error in the first case; thus, its total 0-1 error is at most $\exp(-\Omega(t^2/\epsilon)^2)$. Moreover, since the probability of $y = 1$ is substantially larger than the probability of $y = -1$, it is not hard to see that for any $x$ with $\mathrm{sign}(p(x)) = 1$ that $\mathbf{Pr}[y = 1 \mid \mathbf{x} = x] > 1 - O(\eta)$. This implies that $y$ is given by $\mathrm{sign}(p(x))$ with Massart noise $O(\eta)$.

## 3.1 Basic Rejection Sampling Procedure

In this subsection, we show that by performing rejection sampling on LWE samples, one can obtain a distribution similar to the "hidden direction discrete Gaussian". For the sake of intuition, we start with the following simple lemma. The lemma essentially states that, doing rejection sampling on LWE samples, gives a distribution with the following properties: On the hidden direction $\mathbf{s}$, the distribution is pointwise close to the convolutional sum of a discrete Gaussian and a continuous Gaussian noise. Moreover, on all the other directions $\perp \mathbf{s}$, the distribution is nearly independent of its value on $\mathbf{s}$, in the sense that conditioning on any value on $\mathbf{s}$, the distribution on $\perp \mathbf{s}$ stays pointwise close to a Gaussian. Note that this distribution closely resembles the "hidden direction discrete Gaussian" in [DK22].

**Lemma 3.3.** *Let $(\mathbf{x}, y)$ be a sample of the $\mathrm{LWE}(m, \mathbb{R}_1^n, \{\pm 1\}^n, D_\sigma^\mathcal{N}, \mathrm{mod}_1)$ from the alternative hypothesis case, let $y'$ be any constant in $[0, 1)$, and let $\mathbf{x}' \sim (1/\sigma_{\mathrm{scale}}) \circ D_{\mathbf{x}+\mathbb{Z}^n, \sigma_{\mathrm{scale}}}^\mathcal{N} \mid (y = y')$. Then we have the following: (i) For $\mathbf{x'^s}$, we have that for any $u \in \mathbb{R}$ it holds that $P_{\mathbf{x'^s}}(u) = (1 \pm O(\delta)) P_{D' \star D_{\sigma_{\mathrm{noise}}}^\mathcal{N}}(u)$, where $D' = D_{T(y'+\mathbb{Z}), \sigma_{\mathrm{signal}}}^\mathcal{N}$, and $T = \mathrm{SR}/(n^{1/2}\sigma_{\mathrm{scale}})$, $\sigma_{\mathrm{signal}} = \sqrt{\mathrm{SR}}$, $\sigma_{\mathrm{noise}} = \sqrt{1 - \mathrm{SR}}$, and $\mathrm{SR} = \frac{\sigma_{\mathrm{scale}}^2}{\sigma_{\mathrm{scale}}^2 + \sigma^2/n}$, (ii) $\mathbf{x'^{\perp s}}$ is "nearly independent" of $\mathbf{x'^s}$, namely for any $l \in \mathbb{R}$ and $\mathbf{u} \in \mathbb{R}^{n-1}$ we have that $P_{\mathbf{x}^\perp \mid \mathbf{x^s}=l}(\mathbf{u}) = (1 \pm O(\delta)) P_{D_{\mathbb{R}^{n-1}, 1}^\mathcal{N}}(\mathbf{u})$.*

Lemma 3.3 is a special case of Lemma 3.5, which is one of the main lemmas required for our proof. We note that the distribution of $\mathbf{x}'$ obtained from the above rejection sampling is very similar to the "hidden direction discrete Gaussian" used in [DK22]. The key differences are as follows: (i) on the hidden direction, $\mathbf{x'^s}$ is close to a discrete Gaussian plus extra Gaussian noise (instead of simply being a discrete Gaussian), (ii) $\mathbf{x'^{\perp s}}$ and $\mathbf{x'^s}$ are not perfectly independent. More importantly, by taking different values for $y'$ and $\sigma_{\mathrm{scale}}$, we can obtain distributions with the same hidden direction, but their discrete Gaussian on the hidden direction has different spacing ($T$) and offset ($y'$).

To obtain a computational hardness reduction, our goal will be to simulate the instances from [DK22] by replacing the hidden direction discrete Gaussians with the noisy versions that we obtain from this rejection sampling. We next discuss this procedure and see why a naive implementation of it does not produce a PTF with Massart noise.

## 3.2 Intuition for the Hard Instance

The natural thing to try is to simulate the conditional distributions from [DK22] by replacing the hidden direction discrete Gaussian terms in (1) with similar distributions obtained from rejection sampling. In particular, Lemma 3.3 says that we can obtain a distribution which is close to this hidden direction Gaussian plus a small amount of Gaussian noise. Unfortunately, this extra noise will cause problems for our construction.

Recall that the support of $\mathbf{x^s} \mid (y = +1)$ was $\bigcup_{i \in \mathbb{Z}} [it, it + (i+1)\epsilon]$, and the support of $\mathbf{x^s} \mid (y = -1)$ was $\bigcup_{i \in \mathbb{Z}} [it + t/2, it + t/2 + (i+1)\epsilon]$ for [DK22]. With the extra noise, there is a decaying density tail in both sides of each $[it, it + (i+1)\epsilon]$ interval in the support of $\mathbf{x^s} \mid (y = +1)$. The same holds for each interval in the support of $\mathbf{x^s} \mid (y = -1)$. Recalling the three cases of these intervals discussed earlier, this leads to the following issue. In the second case, the intervals have length within $[t/2, t]$; thus, the intervals $[it, it + (i+1)\epsilon]$ and $[it + t/2, it + t/2 + (i+1)\epsilon]$ overlap, i.e., $it + (i+1)\epsilon \geq it + t/2$. On the right side of $[it, it + (i+1)\epsilon]$, in the support of $\mathbf{x^s} \mid (y = -1)$, there is a small region of values for $u$, where $\mathbf{Pr}[y' = +1 \mid \mathbf{x^s} = u] = \mathbf{Pr}[y' = -1 \mid \mathbf{x^s} = u]$. This causes the labels $y = +1$ and $y = -1$ to be equally likely over that small region, violating the Massart condition. (We note that for the first case, there is also this kind of small region that $\mathbf{Pr}[y' = +1 \mid \mathbf{x^s} = u] = \mathbf{Pr}[y' = -1 \mid \mathbf{x^s} = u]$ caused by the noise tail. However, the probability density of this region is negligibly small, as we will later see in Lemma 3.9.)

We can address this by carving out empty spaces in the $[it + t/2, it + t/2 + (i+1)\epsilon]$ intervals for $\mathbf{x^s} \mid (y = -1)$, so that these decaying parts can fit into. Since this only needs to be done for intervals of Case 2, at most $O(t/\epsilon)$ many such slots are needed. It should be noted that no finite slot will totally prevent this from occurring. However, we only need the slot to be wide enough so that the decay of the error implies that there is negligible mass in the overlap (which can be treated as an error).

We also need to discuss another technical detail. In the last section, we defined the rejection sampling process as taking $(1/\sigma_{\text{scale}}) \circ D^{\mathcal{N}}_{\mathbf{x}+\mathbb{Z}^n, \sigma_{\text{scale}}} \mid (y = y')$, where we can control the offset by $y'$ and spacing by $\sigma_{\text{scale}}$ (Lemma 3.3). This distribution is effectively a noisy version of a discrete Gaussian. Therefore, we can produce a noisy version of the hard instances of [DK22] by taking a mixture of these noisy discrete Gaussians. Unfortunately the noise rate of one of these instances will be $\sigma_{\text{noise}}$. This quantity depends on the spacing $T$ of the discrete Gaussian, which varies across the mixture we would like to take. This inconsistent noise rate is inconvenient for our analysis. However, we can fix the issue by adding extra noise artificially to each of the discrete Gaussians in our mixture, so that they will all have a uniform noise rate $\sigma_{\text{noise}}$; see Algorithm 1 and Lemma 3.5.

The last bit of technical detail is that instead of doing the rejection for $y = y'$, which has $0$ acceptance probability, we will only reject if $y$ is not corresponding to any discrete Gaussian we need. Then we do another rejection to make sure that the magnitude of discrete Gaussians in the mixture is correct. In the next subsection, we introduce the complete rejection sampling method.

### 3.3 The Full Hard Instance Construction

We first introduce the complete rejection algorithm, and then explain how the hard instance is produced using it. Below we provide proof overviews; omitted proofs can be found in Appendix C.

#### 3.3.1 The Complete Rejection Algorithm

The rejection sampling algorithm is the following. The sampling process produces the noisy variant of the distribution which, for some carefully selected set $B \subseteq [0, 1]$, has PDF function $\frac{1}{\lambda(B)} \int_B D^{\mathcal{N}}_{k+(t+k-\psi)\mathbb{Z}, 1} dk$ in the hidden direction, as we will see in Lemma 3.5.

---

**Algorithm 1** Rejection Sampling Algorithm

**Inputs:** A sample $(\mathbf{x}, y) \in \mathbb{R}^n_1 \times \mathbb{R}_1$ and the input parameters are $t, \epsilon, \psi \in \mathbb{R}_{>0}$, where $\psi + \epsilon \leq t$, $B \subseteq [\psi, \psi + \epsilon]$, $\delta \in (0, 1)$. In addition, the parameters satisfy items (i)-(iii) of Condition 3.1.
**Output:** REJECT or a sample $\mathbf{x}' \in \mathbb{R}^n$.

1. Reject unless there is a $k \in B$ such that $y = \frac{k}{t+k-\psi}$.

2. Furthermore, reject with probability $1 - \frac{t^2}{(t+k-\psi)^2}$.

3. Let $\text{SR} = 1 - 4(t + \epsilon)^2 \sigma^2$, $\sigma_{\text{scale}} = \frac{\text{SR}}{(t+k-\psi)\sqrt{n}}$ and $\sigma_{\text{add}} = \sqrt{\frac{(1-\text{SR})\sigma^2_{\text{scale}} - \text{SR}(\sigma/\sqrt{n})^2}{\text{SR}}}$. Then, sample independent noise $\mathbf{x}_{\text{add}} \sim D^{\mathcal{N}}_{\mathbb{R}^n, \sigma_{\text{add}}}$ and output $\mathbf{x}' \sim (1/\sigma_{\text{scale}}) \circ D^{\mathcal{N}}_{\mathbf{x}+\mathbf{x}_{\text{add}}+\mathbb{Z}^n, \sigma_{\text{scale}}}$.

---

Notice that the parameter $\text{SR}$ does *not* depend on $y$, whereas $\sigma_{\text{scale}}, \sigma_{\text{add}}$ do depend on $y$.

For convenience, let us use the following notation for the output distributions.

**Definition 3.4** (Output Distribution of Rejection Sampling). Let $D^{\text{alternative}}_{t,\epsilon,\psi,B,\delta}$ be the distributions of $\mathbf{x}'$ produced by Algorithm 1 (conditioned that the algorithm accepts) given that $(\mathbf{x}, y)$ are sampled as follows: let $\mathbf{x} \sim U(\mathbb{R}^n_1)$, $z \sim D^{\mathcal{N}}_\sigma$, and then let $y = \text{mod}_1(\langle \mathbf{x}, \mathbf{s} \rangle + z)$, where $\mathbf{s} \in \{\pm 1\}^n$ is the secret. Furthermore, let $D^{\text{null}}_{t,\epsilon,\psi,B,\delta}$ be a similar distribution, but when $\mathbf{x} \sim U(\mathbb{R}^n_1), y \sim U(\mathbb{R}_1)$ are independent.

Note that $D^{\text{alternative}}_{t,\epsilon,\psi,B,\delta}$ depends on $\mathbf{s}$, but we do not explicitly denote this in our notation.

**Alternative Hypothesis Analysis** The main properties of $D^{\text{alternative}}_{t,\epsilon,\psi,B,\delta}$ are summarized in the following lemma. Essentially, the lemma states that for this distribution $D^{\text{alternative}}_{t,\epsilon,\psi,B,\delta}$, the marginal distribution on the hidden direction $\mathbf{s}$ is pointwise close to the convolution sum of $D'$ and a Gaussian noise, where $D'$ is a linear combination of discrete Gaussians. Moreover, on all the other directions $\perp \mathbf{s}$, the distribution is nearly independent of its value on $\mathbf{s}$, in the sense that conditioning on any value on $\mathbf{s}$, the distribution on $\perp \mathbf{s}$ always stays pointwise close to a Gaussian.

**Lemma 3.5.** *Let* $\mathbf{x}' \sim D^{\text{alternative}}_{t,\epsilon,\psi,B,\delta}$. *Then we have the following: (i) For* $\mathbf{x}'^{\mathbf{s}}$, *we have that for any* $u \in \mathbb{R}$, $P_{\mathbf{x}'^{\mathbf{s}}}(u) = (1 \pm O(\delta)) P_{D' \star D^{\mathcal{N}}_{\sigma_{\text{noise}}}}(u)$ , *where* $D' = \frac{1}{\lambda(B)} \int_B D^{\mathcal{N}}_{k+(t+k-\psi)\mathbb{Z}, \sigma_{\text{signal}}} dk$ ,

$\sigma_{\text{signal}} = \sqrt{\text{SR}}$, *and* $\sigma_{\text{noise}} = \sqrt{1 - \text{SR}} = 2(t + \epsilon)\sigma$. *(SR is defined in Algorithm 1), (ii)* $\mathbf{x}'^{\perp \mathbf{s}}$ *is "nearly independent" of* $\mathbf{x}'^{\mathbf{s}}$; *namely, for any* $l \in \mathbb{R}$ *and* $\mathbf{u} \in \mathbb{R}^{n-1}$, *we have that* $P_{\mathbf{x}'^{\perp \mathbf{s}} | \mathbf{x}'^{\mathbf{s}} = l}(\mathbf{u}) = (1 \pm O(\delta)) P_{D_{\mathbb{R}^{n-1}, 1}^{\mathcal{N}}}(\mathbf{u})$ .

**Null Hypothesis Analysis**    For $D_{t,\epsilon,\psi,B,\delta}^{\text{null}}$, we can show that it is pointwise close to $D_{\mathbb{R}^n, 1}^{\mathcal{N}}$:

**Lemma 3.6.** *For any* $\mathbf{u} \in \mathbb{R}^n$, *we have that* $P_{D_{t,\epsilon,\psi,B,\delta}^{\text{null}}}(\mathbf{u}) = (1 \pm O(\delta)) P_{D_{\mathbb{R}^n, 1}^{\mathcal{N}}}(\mathbf{u})$ .

### 3.3.2 The Reduction Algorithm

With the rejection sampling algorithm (Algorithm 1) at our disposal, we can now give the full construction of the hard instance. We use $D_{t,\epsilon,\psi_+,B_+,\delta}$ for $\mathbf{x} \mid y = +1$, $D_{t,\epsilon,\psi_-,B_-,\delta}$ for $\mathbf{x} \mid y = -1$ (with a carefully chosen pair of $(B_+, \psi_+)$ and $(B_-, \psi_-)$, as we discussed in Section 3.2), and take a proper marginal distribution of $y$ to build a joint distribution of $(\mathbf{x}, y)$. We introduce a reduction algorithm that, given samples from our LWE problem (either from the null or the alternative hypothesis), produces i.i.d. samples $(\mathbf{x}, y)$ from a joint distribution with the following properties:

1. If the input LWE problem is the null hypothesis, then $\mathbf{x} \mid y = +1$ and $\mathbf{x} \mid y = -1$ are close in total variation distance. Therefore, no hypothesis for predicting $y$ in terms of $\mathbf{x}$ can do much better than the best constant hypothesis.

2. If the input LWE problem is the alternative hypothesis, then the joint distribution of $(\mathbf{x}, y)$ we build is close to a distribution $D$ that satisfies $O(\eta)$ Massart condition with respect to a degree-$O(t/\epsilon)$ PTF, and there is a degree-$O(t/\epsilon)$ PTF with small error on $D$.

We formalize the idea from Section 3.2 here. For $\mathbf{x} \mid y = +1$, we will use $\psi_+ \overset{\text{def}}{=} 0$ and $B_+ \overset{\text{def}}{=} [0, \epsilon]$. For $\mathbf{x} \mid y = -1$, we take $\psi_- \overset{\text{def}}{=} t/2$, which is also the same as [DK22]; but instead of taking $B_- \overset{\text{def}}{=} [t/2, t/2 + \epsilon]$, we will need to carve out the slots on $B_-$. First, we define the mapping $g : \mathbb{R} - [-1.5t, 0.5t] \mapsto [0.5t, t]$, as follows: for $i \in \mathbb{Z}$ and $b \in \mathbb{R}_t$, we have that

$$g(it + t/2 + b) \overset{\text{def}}{=} \begin{cases} \frac{b}{i+1} + t/2 & \text{if } i \geq 0; \\ \frac{b-t}{i+2} + t/2 & \text{if } i < 0. \end{cases}$$

This function maps a location $it + t/2 + b$ to the corresponding place we need to carve out on $B_-$, which is defined in Algorithm 2. These intervals are chosen so that the decaying density part of $+1$ can fit in, as we discussed in Section 3.2. Now we introduce the algorithm that reduces LWE to learning Massart PTFs.

We similarly define the output distributions of the algorithms in the two cases as follows:

**Definition 3.7.** Let $D_{\text{PTF}}^{\text{alternative}}$ be mixture of $D_{t,\epsilon,\psi_+,B_+,\delta}^{\text{alternative}}$ and $D_{t,\epsilon,\psi_-,B_-,\delta}^{\text{alternative}}$ with $+1$ and $-1$ labels and weights $1-\eta$ and $\eta$ respectively. Similarly, let $D_{\text{PTF}}^{\text{null}}$ be mixture of $D_{t,\epsilon,\psi_+,B_+,\delta}^{\text{null}}$ and $D_{t,\epsilon,\psi_-,B_-,\delta}^{\text{null}}$ with $+1$ and $-1$ labels and weights $1 - \eta$ and $\eta$ respectively.

The following observation is immediate from the algorithm.

**Observation 3.8.** *In the alternative (resp. null) hypothesis case, the output distribution of Algorithm 2, conditioned on not failing, is the same as* $m'$ *i.i.d. samples drawn from* $D_{\text{PTF}}^{\text{alternative}}$ *(resp.* $D_{\text{PTF}}^{\text{null}}$*).*

**Alternative Hypothesis Analysis**    We prove that there exists a degree-$O(t/\epsilon)$ PTF such that $D_{\text{PTF}}^{\text{alternative}}$ is close to (in total variation distance) satisfying the $O(\eta)$ Massart noise condition with respect to this PTF, and this PTF has small error with respect to $D_{\text{PTF}}^{\text{alternative}}$.

**Lemma 3.9.** $D_{\text{PTF}}^{\text{alternative}}$ *is* $O(\delta/m')$ *close in total variation distance to a distribution* $D^{\text{truncated}}$ *such that there is a degree-$O(t/\epsilon)$ PTF* $\text{sign}(p(\mathbf{x}))$ *that: (i)* $\mathbf{E}_{(\mathbf{x},y) \sim D^{\text{truncated}}}[\text{sign}(p(\mathbf{x})) \neq y] \leq \exp(-\Omega(t^4/\epsilon^2))$, *(ii)* $D^{\text{truncated}}$ *satisfies the* $O(\eta)$ *Massart noise condition with respect to* $\text{sign}(p(\mathbf{x}))$.

**Null Hypothesis Analysis**    The reader is referred to Lemma C.8 in Appendix C for the null hypothesis analysis.

---

**Algorithm 2** Reducing LWE to Learning PTFs with Massart Noise

---

**Inputs:** $m$ samples from an instance of $\text{LWE}(m, \mathbb{R}_1^n, \{\pm 1\}^n, \mathcal{N}_\sigma, \text{mod}_1)$. The input parameters are $m' \in \mathbb{N}, t, \epsilon \in \mathbb{R}_{>0}, \delta \in (0, 1)$, and $\eta > 0$ a sufficiently small value. In addition, the parameters satisfy Condition 3.1.

**Output:** $m'$ many samples in $\mathbb{R}^n \times \{\pm 1\}$ or FAIL.

1. We take $\psi_+ = 0$, $B_+ = [0, \epsilon]$, $\psi_- = t/2$ and

$$B_- \overset{\text{def}}{=} [t/2, t/2 + \epsilon] - \bigcup_{i=\frac{t}{2\epsilon}-1}^{\frac{t}{\epsilon}-1} g([it - 2c'\epsilon, it]) - \bigcup_{i=\frac{t}{2\epsilon}-1}^{\frac{t}{\epsilon}-1} g([it + (i+1)\epsilon, it + (i+1)\epsilon + 2c'\epsilon])$$

$$- \bigcup_{i=-\frac{t}{\epsilon}-1}^{-\frac{t}{2\epsilon}-1} g([it + (i+1)\epsilon - 2c'\epsilon, it + (i+1)\epsilon]) - \bigcup_{i=-\frac{t}{\epsilon}-1}^{-\frac{t}{2\epsilon}-1} g([it, it + 2c'\epsilon]) \; .$$

2. Repeat the following $m'$ times. If at any point the algorithm attempts to use more than $m$ LWE samples from the input, then output FAIL.

   (a) With probability $1 - \eta$, repeat the following until Algorithm 1 accepts and output $\mathbf{x}'$: run Algorithm 1 with the next unused LWE sample from the input and parameters $t, \epsilon, \psi = \psi_+, B = B_+, \delta$. Add $(\mathbf{x}', +1)$ to the output samples.

   (b) With probability $\eta$, repeat the following until Algorithm 1 accepts and output $\mathbf{x}'$: run Algorithm 1 with the next unused LWE sample from the input and parameters $t, \epsilon, \psi = \psi_-, B = B_-, \delta$. Add $(\mathbf{x}', -1)$ to the output samples.

---

**Putting Everything Together** Having reduced LWE to learning Massart PTFs, we can apply a Veronese mapping on the samples; this PTF becomes an LTF on the Veronese mapping. Since we use degree-$O(t/\epsilon)$ Veronese mapping, the dimension for the Massart LTF problem is $N = (n+1)^{O(t/\epsilon)}$.

## 4 Discussion

Our result rules out the existence of polynomial time algorithms achieving error smaller than $\Omega(\eta)$, where $\eta$ is the upper bound on the noise rate, even of the optimal error is very small, assuming the subexponential time hardness of LWE. A technical open question is whether the constant factor in the $\Omega(\eta)$-term of our lower bound can be improved to the value $C = 1$; this would match known algorithms exactly. (As mentioned in the introduction, such a sharp lower bound has been recently established in the SQ model [NT22], improving on [DK22].)

It is also worth noting that our reduction rules out polynomial-time algorithms, but does not rule out, e.g., subexponential or even quasipolynomial time algorithms with improved error guarantees. We believe that obtaining stronger hardness for these problems would require substantially new ideas, as our runtime lower bounds are essentially the same as the best time lower bounds for learning in the (much stronger) agnostic noise model or in restricted models of computation (like SQ). This seems related to the requirement that our bounds require subexponential hardness of LWE in our assumption. As the strongest possible assumptions only allow us to prove quasi-polynomial lower bounds, any substantially weaker assumption will likely fail to prove super-polynomial ones.

## Acknowledgments

Ilias Diakonikolas was supported by NSF Medium Award CCF-2107079, NSF Award CCF-1652862 (CAREER), a Sloan Research Fellowship, and a DARPA Learning with Less Labels (LwLL) grant. Daniel M. Kane was supported by NSF Medium Award CCF-2107547, NSF Award CCF-1553288 (CAREER), a Sloan Research Fellowship, and a grant from CasperLabs. Lisheng Ren was supported by NSF Award CCF-1652862 (CAREER) and a DARPA Learning with Less Labels (LwLL) grant.

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
