## Appendix

**Additional Notation** For $d \in \mathbb{N}$, we use $V_d : \mathbb{R}^n \mapsto \mathbb{R}^{(n+1)^d}$ to denote the degree-$d$ Veronese mapping, where the outputs are corresponding to monomials of degrees at most $d$.

## A  Additional Technical Background

For completeness, we start with the definition of the classic LWE problem. Note that Definition 2.3 generalizes this definition, so we will not use the definition below directly.

**Definition A.1** (Classic Learning with Errors Problem). Let $m, n, q \in \mathbb{N}$, and let $D_{\text{sample}}$, $D_{\text{secret}}$, $D_{\text{noise}}$ be distributions on $\mathbb{Z}_q^n$, $\mathbb{Z}_q^n$, $\mathbb{Z}_q$ respectively. In the $\text{LWE}_{\text{classic}}(m, D_{\text{sample}}, D_{\text{secret}}, D_{\text{noise}}, \text{mod}_q)$ problem, we are given $m$ independent samples $(\mathbf{x}, y)$ and want to distinguish between the following two cases:

(i) **Alternative hypothesis**: $\mathbf{s}$ is drawn from $D_{\text{secret}}$. Then, each sample is generated by taking $\mathbf{x} \sim D_{\text{sample}}, z \sim D_{\text{noise}}$ and letting $y = \text{mod}_q(\langle \mathbf{x}, \mathbf{s} \rangle + z)$.

(ii) **Null hypothesis**: $\mathbf{x}, y$ are independent and each has the same marginal distribution as above.

Throughout our proofs, we need to manipulate discrete Gaussian distributions that are taken modulo 1 and those with noise added. Due to this, it will be convenient to introduce the following definitions.

**Definition A.2** (Expanded Gaussian Distribution from $\mathbb{R}_1^n$). For $\sigma \in \mathbb{R}_+$, let $D_{\mathbb{R}_1^n, \sigma}^{\text{expand}}$ denote the distribution of $\mathbf{x}'$ drawn as follows: first sample $\mathbf{x} \sim U(\mathbb{R}_1^n)$ (using the Lebesgue measure on $\mathbb{R}_1^n$), and then sample $\mathbf{x}' \sim D_{\mathbb{Z}^n + \mathbf{x}, \sigma}^{\mathcal{N}}$.

**Definition A.3** (Collapsed Gaussian Distribution on $\mathbb{R}_1^n$). We will use $D_{\mathbb{R}_1^n, \sigma}^{\text{collapse}}$ to denote the distribution of $\text{mod}_1(\mathbf{x})$ on $\mathbb{R}_1^n$, where $\mathbf{x} \sim D_{\mathbb{R}^n, \sigma}^{\mathcal{N}}$.

We will also need the following fact, which can be easily derived from known bounds in literature. For completeness, we provide the full proof in Appendix A.1.

**Fact A.4.** *Let $n \in \mathbb{N}, \sigma \in \mathbb{R}_+, \epsilon \in (0, 1/3)$ be such that $\sigma \geq \sqrt{\ln(2n(1 + 1/\epsilon))/\pi}$. Then, we have*

$$\frac{P_{D_{\mathbb{R}_1^n, \sigma}^{\text{expand}}/\sigma}(\mathbf{t})}{P_{D_{\mathbb{R}^n, 1}^{\mathcal{N}}}(\mathbf{t})} = \frac{P_{U(\mathbb{R}_1^n)}(\text{mod}_1(\sigma \mathbf{t}))}{P_{D_{\mathbb{R}_1^n, \sigma}^{\text{collapse}}}(\text{mod}_1(\sigma \mathbf{t}))} = 1 \pm O(\epsilon) \ ,$$

*for all $\mathbf{t} \in \mathbb{R}^n$, and*

$$d_{\text{TV}}\left(\frac{D_{\mathbb{R}_1^n, \sigma}^{\text{expand}}}{\sigma}, D_{\mathbb{R}^n, 1}^{\mathcal{N}}\right), d_{\text{TV}}\left(D_{\mathbb{R}_1^n, \sigma}^{\text{collapse}}, U(\mathbb{R}_1^n)\right) = \exp\left(-\Omega(\sigma^2)\right) \ .$$

We will also need the following well-known fact in our proof.

**Lemma A.5** (Corollary 3.10 of [Reg09]). *Let $\mathbf{z} \in \mathbb{R}^n$, $\sigma_1, \sigma_2 \in \mathbb{R}_{>0}$. Assume that*

$$1/\sqrt{1/\sigma_1^2 + \|\mathbf{z}\|_2^2/\sigma_2^2} \geq \eta_\epsilon(L) \ ,$$

*and further suppose that $\mathbf{x} \sim D_{\mathbb{Z}^n, \sigma_1}^{\mathcal{N}}$ and $\mathbf{x}' \sim D_{\sigma_2}^{\mathcal{N}}$. Then the distribution of $\langle \mathbf{x}, \mathbf{z} \rangle + \mathbf{x}'$ is within $O(\epsilon)$ total variation distance to $D_{\sqrt{\|\mathbf{z}\|_2^2 \sigma_1^2 + \sigma_2^2}}^{\mathcal{N}}$. ($\eta_\epsilon(L)$ is the smoothing parameter of a lattice, and is defined in Definition A.7.)*

### A.1  Proof of Fact A.4

We start by recalling the definition of a lattice.

**Definition A.6** (Lattice). Let $\mathcal{B} = (\mathbf{v}_1, \mathbf{v}_2, \cdots, \mathbf{v}_n)$ be a set of $n$ linearly independent vectors in $\mathbb{R}^n$. The lattice $L = L(\mathcal{B})$ defined by $\mathcal{B}$ is the set of all integer linear combinations of vectors in $\mathcal{B}$, i.e., the set $\{\mathbf{v} \in \mathbb{R}^n : \mathbf{v} = \sum_{j=1}^n \alpha_j \mathbf{v}_j, \alpha_j \in \mathbb{Z}\}$.

Since we only use the integer lattices $\mathbb{Z}^n$, we will only introduce notation as necessary. For a more detailed introduction about lattices, the reader is referred to [MG02].

Partially supported Gaussian distributions (Definition 2.1) behave similarly to continuous Gaussian distributions. The similarity can be quantified based on the so-called smoothing parameter of a lattice defined below.

**Definition A.7** (see, e.g., Definition 2.10 of [Reg09]). For an $n$-dimensional lattice $L$ and $\epsilon \in \mathbb{R}_+$, we define the *smoothing parameter* $\eta_\epsilon(L)$ to be the smallest $s$ such that[2] $\rho_{1/s}(L^* \setminus \{0\}) \leq \epsilon$.

**Lemma A.8** (see, e.g., Lemma 2.12 of [Reg09]). *For any $n \in \mathbb{N}$ and $\epsilon \in \mathbb{R}_+$, we have that* $\eta_\epsilon(\mathbb{Z}^n) \leq \sqrt{\frac{\ln(2n(1+1/\epsilon))}{\pi}}$ .

The main lemma we will use here is that, when $\sigma$ is larger than the smoothing parameter, the normalizing factor remains roughly the same after a shift by an arbitrary vector $\mathbf{v}$, as formalized below. (Note that in the discrete Gaussian case, the two sides would have been equal.) This property follows from the proof of [MR07, Lemma 4.4], in which it was shown that $\rho_\sigma(L + \mathbf{v}) \in [(1-\epsilon)\det(L^*), (1+\epsilon)\det(L^*)]$; the lemma then follows since $\det((\mathbb{Z}^n)^*) = 1$.

**Lemma A.9** ([MR07]). *Let $n \in \mathbb{N}, \epsilon \in (0,1)$ and $\sigma \geq \eta_\epsilon(\mathbb{Z}^n)$. Then, for any $\mathbf{v} \in \mathbb{R}^n$, we have that* $\rho_\sigma(\mathbb{Z}^n + \mathbf{v}) \in [1 - \epsilon, 1 + \epsilon]$.

We are now ready to prove Fact A.4.

*Proof of Fact A.4.* Let $\mathbf{r} = \mathrm{mod}_1(\sigma\mathbf{t})$. Notice that

$$
\begin{aligned}
\frac{P_{(1/\sigma)\circ D^{\mathrm{expand}}_{\mathbb{R}_1^n,\sigma}}(\mathbf{t})}{P_{D^{\mathcal{N}}_{\mathbb{R}^n,1}}(\mathbf{t})} &= \frac{1}{P_{D^{\mathcal{N}}_{\mathbb{R}^n,1}}(\mathbf{t})} \cdot P_{U(\mathbb{R}_1^n)}(\mathbf{r}) \cdot P_{D^{\mathcal{N}}_{\mathbb{Z}^n+\mathbf{r},\sigma}}(\sigma\mathbf{t}) \\
&= \frac{1}{P_{D^{\mathcal{N}}_{\mathbb{R}^n,1}}(\mathbf{t})} \cdot P_{D^{\mathcal{N}}_{\mathbb{Z}^n+\mathbf{r},\sigma}}(\sigma\mathbf{t}) \\
&= \frac{1}{P_{D^{\mathcal{N}}_{\mathbb{R}^n,1}}(\mathbf{t})} \cdot \frac{\rho_\sigma(\sigma\mathbf{t})}{\rho_\sigma(\mathbb{Z}^n + \mathbf{r})} \\
&= \frac{1}{P_{D^{\mathcal{N}}_{\mathbb{R}^n,1}}(\mathbf{t})} \cdot \frac{p_1(\mathbf{t})}{\rho_\sigma(\mathbb{Z}^n + \mathbf{r})} \\
&= \frac{1}{\rho_\sigma(\mathbb{Z}^n + \mathbf{r})} \\
&\stackrel{\text{(Lemmas A.9, A.8)}}{=} \frac{1}{1 \pm \epsilon} \\
&= 1 \pm O(\epsilon) \ .
\end{aligned}
$$

Notice also that

$$
\begin{aligned}
\frac{P_{D^{\mathrm{collapse}}_{\mathbb{R}_1^n,\sigma}}(\mathbf{r})}{P_{U(\mathbb{R}_1^n)}(\mathbf{r})} &= P_{D^{\mathrm{collapse}}_{\mathbb{R}_1^n,\sigma}}(\mathbf{r}) \\
&= \sum_{\mathbf{u}\in\mathbb{Z}^n} \rho_\sigma(\mathbf{u} + \mathbf{r}) \\
&= \rho_\sigma(\mathbb{Z}^n + \mathbf{r}) \\
&= \frac{P_{D^{\mathcal{N}}_{\mathbb{R}^n,1}}(\mathbf{t})}{P_{(1/\sigma)\circ D^{\mathrm{expand}}_{\mathbb{R}_1^n,\sigma}}(\mathbf{t})} \ ,
\end{aligned}
$$

where the last equality follows from the previously derived equality above. These prove the first equality in Fact A.4.

---

[2]Note that $L^*$ denotes the *dual lattice* of $L$; it contains all $\mathbf{y}$ such that $\langle \mathbf{x}, \mathbf{y} \rangle$ is an integer for all $\mathbf{x} \in L$.

Next, note that pointwise closeness immediately yields the same bound on the TV distance. Therefore, we have

$$d_{\mathrm{TV}}\left((1/\sigma) \circ D_{\mathbb{R}_1^n,\sigma}^{\mathrm{expand}}, D_{\mathbb{R}^n,1}^{\mathcal{N}}\right), d_{\mathrm{TV}}\left(D_{\mathbb{R}_1^n,\sigma}^{\mathrm{collapse}}, U(\mathbb{R}_1^n)\right) \le \epsilon \le \exp(-\Omega(\sigma^2)),$$

where the second inequality follows from our choice of $\sigma$. $\qquad\square$

## B  Hardness of LWE with Binary Secret, Continuous Samples, and Continuous Noise

The main theorem of this section is a reduction from $\mathrm{LWE}(n, \mathbb{Z}_q^l, \mathbb{Z}_q^l, D_{\mathbb{Z},\sigma}^{\mathcal{N}}, \mathrm{mod}_q)$ to $\mathrm{LWE}(m, \mathbb{R}_1^n, \{\pm 1\}^n, D_{\sigma'}^{\mathcal{N}}, \mathrm{mod}_1)$. The purpose of this reduction is to massage the LWE problem into its variant with binary secret, continuous samples and continuous noise, so we can further reduce it to the MASSART problem.

We reiterate that this reduction between LWE problems follows from the previous works [Mic18a, GVV22]; we provide the full proof here for completeness. The main theorem regarding the reduction is presented below.

**Theorem B.1.** *Let $n, m, l, q \in \mathbb{N}$, $\sigma, \sigma' \in \mathbb{R}$, where the parameters satisfy:*

1. *$\log(q)/2^l = \delta^{\omega(1)}$ (where $\omega(1)$ goes to infinity as $n$ goes to infinity),*

2. *$\sigma = \omega(\sqrt{\log(mn/\delta)})$*

3. *$n \ge 2l \log_2(q) + \omega(\log(1/\delta))$, and*

4. *$\sigma' = c\sqrt{n}\,\sigma/q$, where $c$ is a sufficiently large constant.*

*Suppose there is no $T + \mathrm{poly}(m, n, l, \log q, \log(1/\delta))$ time distinguisher for $\mathrm{LWE}(n, \mathbb{Z}_q^l, \mathbb{Z}_q^l, D_{\mathbb{Z},\sigma}^{\mathcal{N}}, \mathrm{mod}_q)$ with $\epsilon/m$ advantage. Then there is no $T$ time distinguisher for $\mathrm{LWE}(m, \mathbb{R}_1^n, \{\pm 1\}^n, \mathcal{N}_{\sigma'}, \mathrm{mod}_1)$ with $2\epsilon + O(\delta)$ advantage.*

Before we prove Theorem B.1, let us note that combining it with Assumption 2.4 yields Lemma 2.5.

*Proof of Lemma 2.5.* We take $n = l^\alpha$, $m = 2^{O(l^{\beta'})}$, $q = l^{\gamma'}$, and $\sigma = \sqrt{l}$, where $\alpha > 1$ and $\beta'' > \beta' \in (0,1)$. Then, from Assumption 2.4, it follows that there is no $2^{O(n^{\beta''/\alpha})}$ time algorithm to solve $\mathrm{LWE}\left(n, \mathbb{Z}_q^l, \mathbb{Z}_q^l, D_{\mathbb{Z},n^{\alpha/2}}^{\mathcal{N}}, \mathrm{mod}_q\right)$ with $2^{-O(n^{\beta''/\alpha})}$ advantage.

We take $\delta$ to be a sufficiently small constant and apply Theorem B.1. Then we have that no $2^{O(n^{\beta'/\alpha})}$ time algorithm can solve $\mathrm{LWE}\left(2^{O(n^{\beta'/\alpha})}, \mathbb{R}_1^n, \{\pm 1\}^n, D_{\mathbb{Z},O\left(n^{1/(2\alpha)+1/2-\gamma'/\alpha}\right)}^{\mathcal{N}}, \mathrm{mod}_1\right)$ with $1/3$ advantage. We rename $\beta = \beta'/\alpha$ and $\gamma = 1/(2\alpha) + 1/2 - \gamma'/\alpha$. By taking $\alpha > 1$ and $\beta' < 1$ to be arbitrarily close to $1$ and $\gamma'$ to be arbitrarily large constant, we can have $\beta \in (0,1)$ to be arbitrarily close to $1$ and $\gamma$ to be arbitrarily large. Then there is no $2^{O(n^\beta)}$ time algorithm to solve $\mathrm{LWE}\left(2^{O(n^\beta)}, \mathbb{R}_1^n, \{\pm 1\}^n, D_{O(n^{-\gamma})}^{\mathcal{N}}, \mathrm{mod}_1\right)$ with $1/3$ advantage.

Given the above, by a standard boosting argument, it follows that there is no $2^{O(n^\beta)}$ time algorithm to solve $\mathrm{LWE}\left(2^{O(n^\beta)}, \mathbb{R}_1^n, \{\pm 1\}^n, D_{O(n^{-\gamma})}^{\mathcal{N}}, \mathrm{mod}_1\right)$ with $2^{-O(n^\beta)}$ advantage. $\qquad\square$

**From $\mathbb{Z}_q$ Secret to Binary Secret**  We require the following lemma from [GVV22] that reduces the classic LWE problem with $\mathbb{Z}_q$ secret to an LWE problem with binary secret.

**Lemma B.2** (Theorem 7 from [GVV22]). *Let $q, l, n, m \in \mathbb{N}$ and $\sigma \in \mathbb{R}_+$. Assuming that there is no time $T + \mathrm{poly}(l, n, \log(q), \log(1/\delta))$ algorithm for solving $\mathrm{LWE}(n + 1, \mathbb{Z}_q^l, \mathbb{Z}_q^l, D_{\mathbb{Z},\sigma}^{\mathcal{N}}, \mathrm{mod}_q)$ with advantage $(\epsilon - \delta^{\omega_n(1)})/(2m)$, there is no time $T$ algorithm for solving $\mathrm{LWE}(m, \mathbb{Z}_q^{n+1}, \{\pm 1\}^{n+1}, D_{\mathbb{Z},\sigma'}^{\mathcal{N}}, \mathrm{mod}_q)$ with $\epsilon$ advantage, as long as the following holds: $\log(q)/2^l = \delta^{\omega_n(1)}$, $\sigma \ge 4\sqrt{\omega(\log(1/\delta)) + \log m + \log n}$, $n \ge 2l \log_2(q) + \omega(\log(1/\delta))$, and $\sigma' = 2\sigma\sqrt{n+1}$.*

**From Discrete To Continuous** In the next step, we show that adding a small amount of Gaussian noise on $y$ will render the discrete Gaussian noise close to continuous Gaussian noise.

**Lemma B.3** (Lemma 15 from [GVV22]). *Let $n, m, q \in \mathbb{N}$, $\sigma \in \mathbb{R}_+$, $c$ be a sufficiently large constant and suppose that $\sigma > \sqrt{c \log(m/\delta)}$. Suppose there is no distinguisher for $\mathrm{LWE}(m, \mathbb{Z}_q^n, \{\pm 1\}^n, D_{\mathbb{Z}, \sigma}^{\mathcal{N}}, \mathrm{mod}_q)$ running in time $T + \mathrm{poly}(m, n \log(q), \log(1/\delta))$ with $\epsilon$ advantage. Then there is no $T$-time distinguisher for $\mathrm{LWE}(m, \mathbb{Z}_q^n, \{\pm 1\}^n, D_{\sigma'}^{\mathcal{N}}, \mathrm{mod}_q)$ with $\epsilon + O(\delta)$ advantage, where*

$$\sigma' \geq \sqrt{\sigma^2 + c \log(m/\delta)} \ .$$

*Proof.* We will give a reduction argument. Take $\sigma_{\mathrm{add}} = \sqrt{\sigma'^2 - \sigma^2}$. Then for each sample $(\mathbf{x}, y)$ from $\mathrm{LWE}(m, \mathbb{Z}_q^n, \{\pm 1\}^n, D_{\mathbb{Z}, \sigma}^{\mathcal{N}}, \mathrm{mod}_q)$, we return

$$(\mathbf{x}, \mathrm{mod}_1(y + e)) \text{ where } e \sim D_{\sigma_{\mathrm{add}}}^{\mathcal{N}}$$

as a sample for $\mathrm{LWE}(m, \mathbb{Z}_q^n, \{\pm 1\}^n, D_{\sigma'}^{\mathcal{N}}, \mathrm{mod}_q)$.

Suppose that the input instance is in the alternative hypothesis case. We need to argue that after running the reduction algorithm, the new noise $z + e$ has at most $O(\delta/m)$ total variation distance from $D_{\sigma'}^{\mathcal{N}}$. From Lemma A.5, we have that for a sufficiently large constant $c$, $(z + e)$ is within $O(\delta/m)$ total variation distance to $D_{\sigma'}^{\mathcal{N}}$. With $m$ samples, this only decreases the distinguishing advantage by at most $O(\delta)$.

Suppose that input instance is from the null hypothesis case. We need to show after the reduction algorithm, both $\mathbf{x}$ and $y$ have the same marginal as in the previous case. It is easy to verify that after the reduction, $\mathbf{x}$ will have the same marginal as in the previous case. For $y$, the marginal distribution of $y$ in the previous case is $U(\mathbb{R}_q)$ by symmetry. Similarly, in the null hypothesis case, the distribution of $y$ is also $U(\mathbb{R}_q)$ by symmetry. $\qquad\square$

We now show that adding a small amount of Gaussian noise on the samples will render the samples continuous; at the same time, this extra Gaussian noise on samples can be interpreted as some extra Gaussian noise on the labels.

**Lemma B.4** (Lemma 16 from [GVV22]). *Let $n, m, q \in \mathbb{N}$, $\sigma \in \mathbb{R}$, $c$ be a sufficiently large constant. Suppose that $\sigma \geq c n^{1/2} \sqrt{\log(mn/\delta)}$. Suppose there is no $T + \mathrm{poly}(m, n, \log(q), \log(1/\delta))$-time distinguisher for $\mathrm{LWE}(m, \mathbb{Z}_q^n, \{\pm 1\}, D_{\sigma}^{\mathcal{N}}, \mathrm{mod}_q)$ with $\epsilon$ advantage. Then there is no $T$-time distinguisher for $\mathrm{LWE}(m, \mathbb{R}_q^n, \{\pm 1\}, D_{\sigma'}^{\mathcal{N}}, \mathrm{mod}_q)$ with $\epsilon + \delta$ advantage, where*

$$\sigma' = \sqrt{\sigma^2 + cn \log(mn/\delta)} \ .$$

*Proof.* We will give a reduction argument. Taking $\sigma_{\mathrm{add}} = \sqrt{\frac{\sigma'^2 - \sigma^2}{n}}$, then for each sample $(\mathbf{x}, y)$ from $\mathrm{LWE}(m, \mathbb{Z}_q^n, \{\pm 1\}^n, D_{\sigma}^{\mathcal{N}}, \mathrm{mod}_q)$, we return

$$(\mathrm{mod}_q(\mathbf{x} + \mathbf{x}'), y) \text{ ,where } \mathbf{x}' \sim D_{\mathbb{R}^n, \sigma_{\mathrm{add}}}^{\mathcal{N}}$$

as a sample for $\mathrm{LWE}(m, \mathbb{R}_q^n, \{\pm 1\}^n, D_{\sigma'}^{\mathcal{N}}, \mathrm{mod}_q)$.

Suppose the input instance is in the alternative hypothesis case. We need to show the following:

(a) $\mathrm{mod}_q(\mathbf{x} + \mathbf{x}')$ is close to $U(\mathbb{R}_q^n)$; and

(b) $y = \mathrm{mod}_q(\langle \mathrm{mod}_q(\mathbf{x} + \mathbf{x}'), \mathbf{S} \rangle + z')$, where $z'$ is the noise in this new LWE instance we generated. We need to show $z'$ has distribution close to a independent $D_{\sigma'}^{\mathcal{N}}$ noise (independent of $\mathrm{mod}_q(\mathbf{x} + \mathbf{x}')$).

For (a), from the symmetry of $\mathbf{x}$ and $\mathbf{x} + \mathbf{u}$ where $\mathbf{u} \in \mathbb{Z}_q^n$, we have

$$d_{\mathrm{TV}}(\mathrm{mod}_q(\mathbf{x} + \mathbf{x}'), U(\mathbb{R}_q^n)) = d_{\mathrm{TV}}(\mathrm{mod}_q(\mathbf{x} + \mathbf{x}') | \mathrm{mod}_q(\mathbf{x} + \mathbf{x}') \in [0,1]^n, U(\mathbb{R}_1^n))$$
$$= d_{\mathrm{TV}}(\mathrm{mod}_1(\mathbf{x}'), U(\mathbb{R}_1^n)) \ .$$

From Fact A.4, we know that for a sufficiently large constant $c$, the distribution of $\mathrm{mod}_q(\mathbf{x}')$ is $O(\delta/m)$ close to $U(\mathbb{R}_q^n)$.

For (b), consider $\mathbf{x} \sim U(\mathbb{Z}_q^n), s \sim U(\{\pm 1\}^n), z \sim D_\sigma^{\mathcal{N}}$ and $y = \mathrm{mod}_q(\langle \mathbf{x}, \mathbf{S}\rangle + z)$. Then the new sample satisfies

$$y = \mathrm{mod}_q(\langle \mathrm{mod}_q(\mathbf{x} + \mathbf{x}'), \mathbf{S}\rangle + (-\langle \mathbf{x}', \mathbf{S}\rangle + z)) ,$$

where the new noise is $z' = -\langle \mathbf{x}', \mathbf{S}\rangle + z$. We now verify the distribution of noise. Conditioned on a fixed $\mathbf{x} + \mathbf{x}'$, we have noise as $-\langle \mathbf{x}', \mathbf{S}\rangle + z$, where $\mathbf{x}' \sim D_{\mathbf{x}+\mathbf{x}'+\mathbb{Z}^n, \sigma_{\mathrm{add}}}^{\mathrm{partial}}$ and $z \sim D_\sigma^{\mathcal{N}}$. From Lemma A.5, we have that the distribution of noise is $O(\delta/m)$ close to $D_{\sigma'}^{\mathcal{N}}$, for sufficiently large $c$. Overall, with $m$ samples, the distinguishing advantage has decreased by at most $O(\delta)$.

If the input instance is from the null hypothesis case, it is easy to verify that after the reduction, both $\mathbf{x}$ and $\mathbf{y}$ will have the same marginal as in the alternative hypothesis case. $\qquad\square$

The final step is to rescale the sample and noise by $1/q$.

**Lemma B.5.** *Suppose there is no $T + \mathrm{poly}(m, n, \log q)$ time distinguisher for the distribution* $\mathrm{LWE}(m, \mathbb{R}_q^n, \{\pm 1\}^n, D_\sigma^{\mathcal{N}}, \mathrm{mod}_q)$ *with advantage $\epsilon$. Then there is no $T$ time distinguisher for the distribution* $\mathrm{LWE}(m, \mathbb{R}_1^n, \{\pm 1\}^n, D_{\sigma'}^{\mathcal{N}}, \mathrm{mod}_1)$ *with advantage $\epsilon$, where $\sigma' = \sigma/q$.*

*Proof.* This follows simply by rescaling samples by $1/q$ and changing $\mathrm{mod}_q$ to $\mathrm{mod}_1$. Note the size of the secret remains unchanged here, but the noise is scaled by $1/q$. $\qquad\square$

**Putting Things Together: Proof of Theorem B.1**   Now we are ready to prove Theorem $B.1$.

*Proof of Theorem $B.1$.* Suppose there is no $T + \mathrm{poly}(m, n, l, \log q, \log(1/\delta))$ time distinguisher for $\mathrm{LWE}(n, \mathbb{Z}_q^l, \mathbb{Z}_q^l, D_{\mathbb{Z},\sigma}^{\mathcal{N}}, \mathrm{mod}_q)$ with $\epsilon/m$ advantage. Then, by applying Lemma B.2, we have that there is no $T + \mathrm{poly}(m, n, \log q, \log(1/\delta))$ time distinguisher for $\mathrm{LWE}(n, \mathbb{Z}_q^l, \{\pm 1\}^l, D_{\mathbb{Z},\sigma_1}^{\mathcal{N}}, \mathrm{mod}_q)$ with $2\epsilon + \delta^{\omega_n(1)}$ advantage, where $\sigma_1 = 2\sigma\sqrt{n+1}$.

Then, we apply Lemma B.3, Lemma B.4, and Corollary B.5. It follows that there is no time $T$ distinguisher for $\mathrm{LWE}(m, \mathbb{R}_1^n, \{\pm 1\}^n, \mathcal{N}_{\sigma'}, \mathrm{mod}_1)$ with $2\epsilon + O(\delta)$ advantage, where $\sigma' \geq \sqrt{\sigma_1^2 + cn\log(mn/\delta)}/q$.

Recalling that $\sigma = \omega(\sqrt{\log(mn/\delta)})$, $\sigma_1 = 2\sigma\sqrt{n+1}$ and $\sigma' \geq \sqrt{\sigma_1^2 + cn\log(mn/\delta)}/q$, we have that $\sigma' = c\sqrt{n}\sigma/q$ is sufficient. $\qquad\square$

# C  Omitted Proofs from Section 3

This section includes additional details of our Massart halfspace hardness reduction, and contains the proofs omitted from Section 3.

We start with Figure 1, which shows a rough flow chart of the reduction algorithm and its relation with the relevant theorems and lemmas.

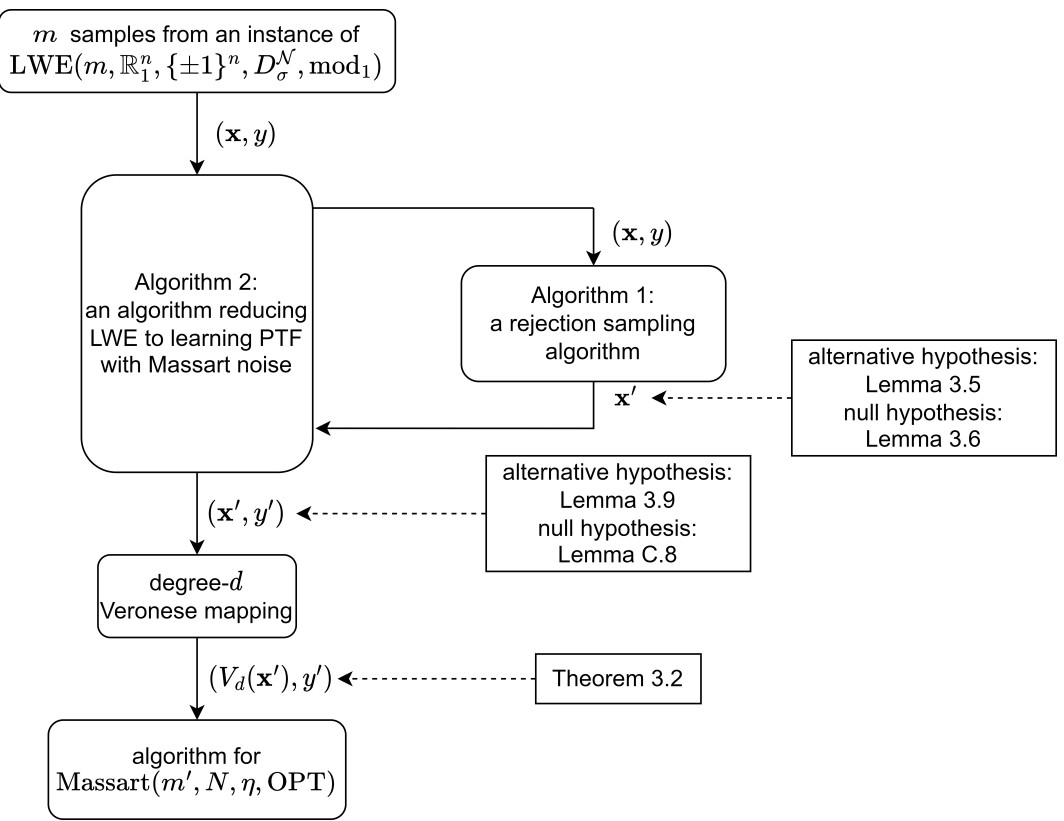

Figure 1: **Reducing LWE to Learning Halfspaces with Massart Noise.** The diagram shows which step of the analysis each lemma is used for and which case of the input LWE instance (alternative/null hypothesis case) for the reduction algorithm. Lemma 3.5 (resp. Lemma 3.6) analyzes the properties of $\mathbf{x}'$ when the input LWE instance is from the alternative hypothesis case (resp. null hypothesis case). Lemma 3.9 (resp. Lemma C.8) analyzes the properties of $(\mathbf{x}', y')$ when the input LWE instance is from the alternative hypothesis case (resp. null hypothesis case). Theorem 3.2 analyzes the properties of $(V_d(\mathbf{x}'), y')$ for the input LWE instance from both cases.

## C.1   Illustration of Hard Instances

For the sake of intuition, we additionally present the following figures to illustrate the ideas behind our construction. Figure 2 illustrates the original [DK22] construction, as discussed in Section 3.

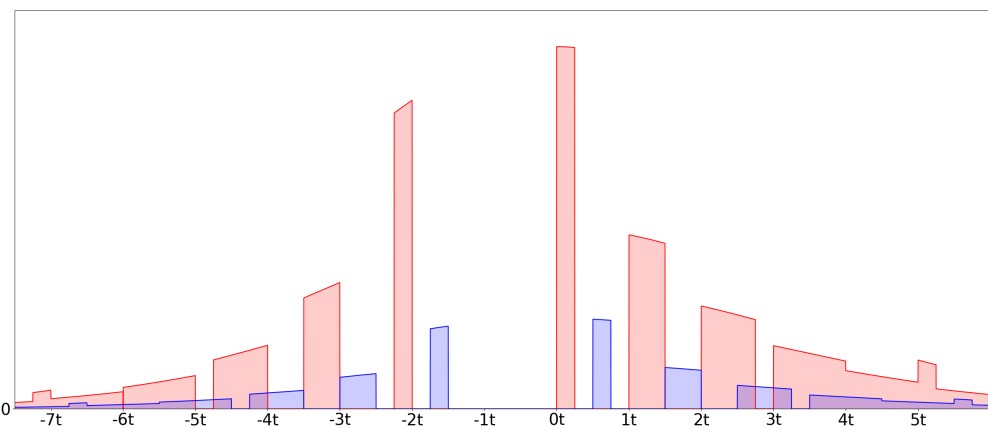

Figure 2: The original SQ-hard construction in [DK22] (specifically, the univariate distribution in the hidden direction). The red part corresponds to $\mathbf{Pr}[y=+1]P_{[\mathbf{x}^s|(y=+1)]}(\cdot)$; the blue part corresponds to $\mathbf{Pr}[y=-1]P_{[\mathbf{x}^s|(y=-1)]}(\cdot)$.

As discussed in Section 3.2, if we try to replace the "hidden direction discrete Gaussian" with its noisy variant, we will get a construction as the one illustrated in Figure 3.

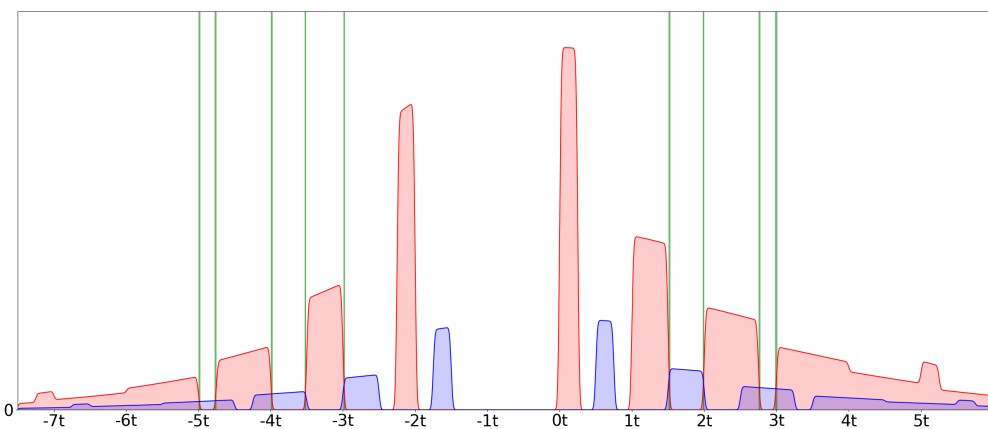

Figure 3: The [DK22] construction with a noisy "hidden direction discrete Gaussian". The red part corresponds to $\mathbf{Pr}[y=+1]P_{[\mathbf{x}^s|(y=+1)]}(\cdot)$, and the blue part corresponds to $\mathbf{Pr}[y=-1]P_{[\mathbf{x}^s|(y=-1)]}(\cdot)$. Notice the thin green intervals where $\frac{\mathbf{Pr}[y=+1]P_{[\mathbf{x}^s|(y=+1)]}(\cdot)}{\mathbf{Pr}[y=-1]P_{[\mathbf{x}^s|(y=-1)]}(\cdot)}$ is close to 1; the labels $+1$ and $-1$ are close to equally likely in these regions, and thus violate the Massart noise condition. (Notice that, as we explained in Section 3.2, there are other places where $\frac{\mathbf{Pr}[y=+1]P_{[\mathbf{x}^s|(y=+1)]}(\cdot)}{\mathbf{Pr}[y=-1]P_{[\mathbf{x}^s|(y=-1)]}(\cdot)}$ is close to 1; however, the density of these regions is negligible.)

Our idea is to modify the above construction by carving empty slots on the support of $\mathbf{x}^s \mid (y=-1)$, so the "clean" version of the construction (using the "hidden direction discrete Gaussian" without noise) is as presented in Figure 4.

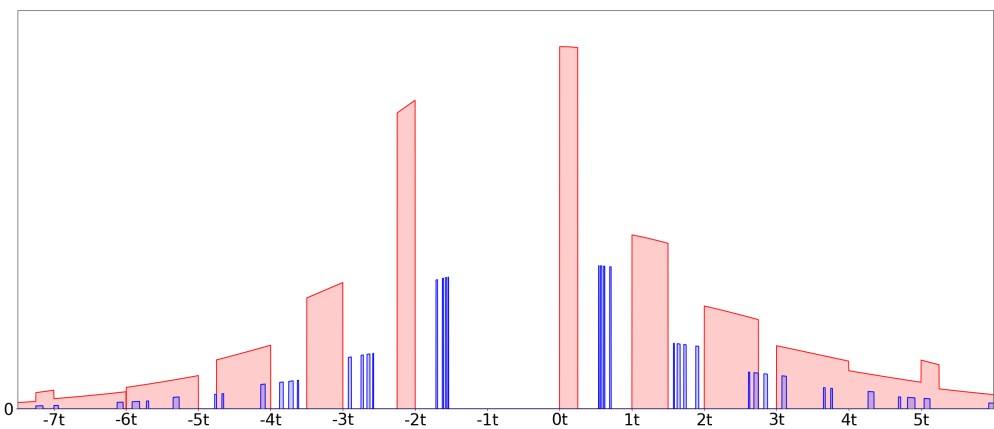

Figure 4: The modified hardness construction with the clean "hidden direction discrete Gaussian". The red part corresponds to $\mathbf{Pr}[y = +1]P_{[\mathbf{x}^s|(y=+1)]}(\cdot)$ and the blue part corresponds to $\mathbf{Pr}[y = -1]P_{[\mathbf{x}^s|(y=-1)]}(\cdot)$. Compared with the original construction, we carve empty slots on $\mathbf{x}^s \mid (y = -1)$ for the problematic part to fit in. Since the total mass we carve out is at most a constant fraction, this causes the density to increase by at most a constant multiplicative factor.

With the noisy "hidden direction discrete Gaussian", the modified construction is as presented in Figure 5.

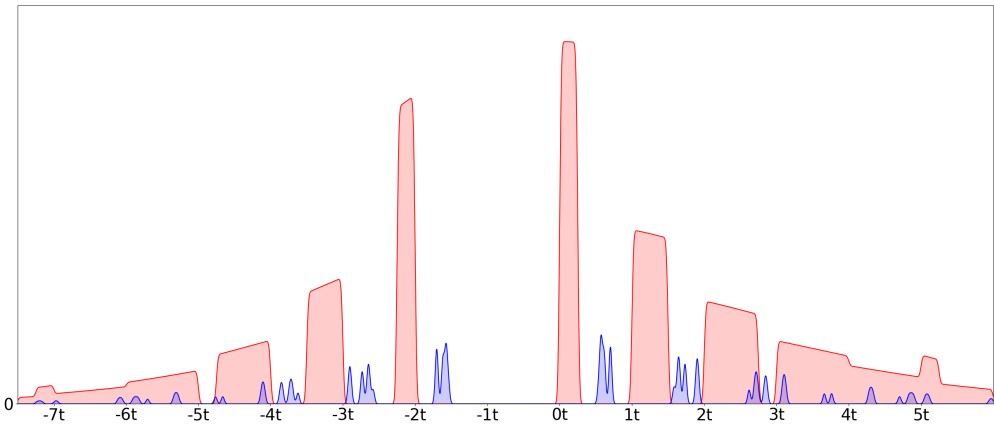

Figure 5: The modified construction with the noisy "hidden direction discrete Gaussian". The red part corresponds to $\mathbf{Pr}[y = +1]P_{[\mathbf{x}^s|(y=+1)]}(\cdot)$ and the blue part corresponds to $\mathbf{Pr}[y = -1]P_{[\mathbf{x}^s|(y=-1)]}(\cdot)$.

It is worth noting that even this modified construction does not perfectly satisfy the Massart condition. However, the part that violates the Massart condition is negligibly small, thus it is very close to a distribution that perfectly satisfies the Massart condition.

## C.2 Omitted Proofs from Section 3.1

Here we give the proof for Lemma 3.3. Lemma 3.3 is stated only for the sake of intuition, and is not needed for the proof of our main theorem.

*Proof of Lemma 3.3.* Notice $y = \mathrm{mod}_1(\langle \mathbf{s}, \mathbf{x}\rangle + z)$. We rename the variables as $\mathbf{x}_{\mathrm{new}} \stackrel{\mathrm{def}}{=} \mathbf{x}$, $\sigma'_{\mathrm{scale}} \stackrel{\mathrm{def}}{=} \sigma_{\mathrm{scale}}$ and $z_{\mathrm{new}} = z$ then we have

$$\mathbf{x}' \sim (1/\sigma'_{\mathrm{scale}}) \circ D^{\mathcal{N}}_{\mathbf{x}_{\mathrm{new}}+\mathbb{Z}^n, \sigma'_{\mathrm{scale}}} | (y = y')$$

where $\mathbf{x}_{\mathrm{new}} \sim U(\mathbb{R}^n_1)$, $z_{\mathrm{new}} \sim D^{\mathcal{N}}_\sigma$ are independent and $y = \mathrm{mod}_1(\langle \mathbf{s}, \mathbf{x}_{\mathrm{new}}\rangle + z_{\mathrm{new}})$. This is the exact same distribution we calculated in the proof of Lemma 3.5 (after we rewrote the distribution in Lemma 3.5). The same calculation gives the lemma statement here. $\square$

## C.3 Omitted Proofs from Section 3.3

### C.3.1 Analysis of Algorithm 1

We first prove that the parameters defined in Algorithm 1 are well posed. In particular, we need to show that the expression inside the square root in Step 3 of Algorithm 1 is positive.

**Observation C.1.** *Let* $\mathrm{SR}, \sigma_{\mathrm{scale}}$ *be as in Algorithm 1. Then* $\mathrm{SR} \geq 1/2$ *and* $(1 - \mathrm{SR})\sigma^2_{\mathrm{scale}} \geq \mathrm{SR}(\sigma/\sqrt{n})^2$.

*Proof.* The first three conditions of the parameters in Condition 3.1 yield $(t + \epsilon)\sigma \leq 2t\sigma \leq 2t\sqrt{n} \leq 2/\sqrt{c\log(n/\delta)}$, which is at most $1/8$ for sufficiently large $c$. This implies that $\mathrm{SR} \geq 1/2$.

Moreover, from our choice of $\sigma_{\mathrm{scale}}$, we have

$$\frac{(1 - \mathrm{SR})\sigma^2_{\mathrm{scale}}}{\mathrm{SR}(\sigma/\sqrt{n})^2} = \frac{\mathrm{SR}(1 - \mathrm{SR})}{((t+k-\psi)\sigma)^2} \geq \frac{(1/2)(4(t+\epsilon)^2\sigma^2)}{((t+\epsilon)\sigma)^2} = 2.$$

$\square$

We note that Observation C.1 implies that the expression inside the square root in Step 3 of Algorithm 1 is positive; therefore, $\sigma_{\mathrm{add}}$ in Step 3 is real. It is also useful to observe some other properties of the parameters that will be useful later in the proof of Lemma 3.5.

**Observation C.2.** *Let* $\mathrm{SR}, \sigma_{\mathrm{scale}}$ *be as in Algorithm 1. Then* $\sigma_{\mathrm{scale}} \geq \frac{1}{2(t+k-\psi)\sqrt{n}}$ *and* $\mathrm{SR} = \frac{\sigma^2_{\mathrm{scale}}}{\sigma^2_{\mathrm{scale}}+\sigma^2_{\mathrm{add}}+\sigma^2/n}$.

*Proof.* The observation $\sigma_{\mathrm{scale}} \geq \frac{1}{2(t+k-\psi)\sqrt{n}}$ follows from $\mathrm{SR} \geq 1/2$ (Observation C.1) and $\sigma_{\mathrm{scale}} = \frac{\mathrm{SR}}{(t+k-\psi)\sqrt{n}}$. Then $\mathrm{SR} = \frac{\sigma^2_{\mathrm{scale}}}{\sigma^2_{\mathrm{scale}}+\sigma^2_{\mathrm{add}}+\sigma^2/n}$ follows from the definition of $\sigma_{\mathrm{add}}$ in Algorithm 1. $\square$

We next prove a lower bound on the acceptance probability of Algorithm 1.

**Lemma C.3.** *If* $y \sim U(\mathbb{R}_1)$, *then Algorithm 1 accepts with probability at least* $\Omega(\frac{\lambda(B)(t-\psi)}{t^2})$.

*Proof.* The probability of a sample not being rejected by the first rejection in Step 1 of Algorithm 1 is

$$\lambda\left(\left\{\frac{k}{t+(k-\psi)}, k \in B\right\}\right) \geq \min_{k \in B}\left[\frac{d\left(\frac{k}{t+(k-\psi)}\right)}{dk}\right] \cdot \lambda(B) \geq \frac{\lambda(B)(t-\psi)}{(t+\epsilon)^2} = \Omega\left(\frac{\lambda(B)(t-\psi)}{t^2}\right),$$

where the inequality holds because $\frac{k}{t+(k-\psi)}$ is monotone increasing for $k \in B$. Conditioned on passing the first rejection, the second round rejects with probability $1 - \frac{t^2}{(t+k-\psi)^2}$, which is at most $3/4$ since $t + k - \psi \leq t + \epsilon \leq 2t$. Therefore, a sample is accepted with probability at least $\Omega\left(\frac{\lambda(B)(t-\psi)}{t^2}\right)$. $\square$

### C.3.2 Proof of Lemma 3.5

*Proof of Lemma 3.5.* **We first prove property (i).** We first review the definition of $D_{t,\epsilon,\psi,B,\delta}^{\text{alternative}}$. We have $(\mathbf{x}, y)$ from the alternative hypothesis case of the LWE problem. Namely, this means we have an unknown $\mathbf{s} \in \{\pm 1\}^n$ and independently sampled $\mathbf{x} \sim U(\mathbb{R}_1^n)$ and $z \sim D_\sigma^{\mathcal{N}}$. Then $y = \text{mod}_1(\langle \mathbf{x}, \mathbf{s} \rangle + z)$. We reject unless $y = \frac{k}{t+(k-\psi)}$ for some $k \in B$. Then the secondary rejection step rejects with probability $1 - \frac{t^2}{(t+k-\psi)^2}$. We calculate $\sigma_{\text{add}}$ and $\sigma_{\text{scale}}$ which only depend on $y$ and output a sample

$$\mathbf{x}' \sim (1/\sigma_{\text{scale}}) \circ D_{\mathbf{x}+\mathbf{x}_{\text{add}}+\mathbb{Z}^n, \sigma_{\text{scale}}}^{\mathcal{N}},$$

where $\mathbf{x}_{\text{add}} \sim D_{\mathbb{R}^n, \sigma_{\text{add}}}^{\mathcal{N}}$. Let $k$ be the value such that $y' = \frac{k}{t+(k-\psi)}$. Then since $y \sim U(\mathbb{R}_1)$

$$D_{t,\epsilon,\psi,B,\delta}^{\text{alternative}} \propto \int_0^1 \mathbf{1}(k \in B) \left( \frac{t^2}{(t+k-\psi)^2} \right) \left[ (1/\sigma_{\text{scale}}) \circ D_{\mathbf{x}+\mathbf{x}_{\text{add}}+\mathbb{Z}^n, \sigma_{\text{scale}}}^{\mathcal{N}} \Big| (y = y') \right] dy'$$

$$\propto \int_B \left[ (1/\sigma_{\text{scale}}) \circ D_{\mathbf{x}+\mathbf{x}_{\text{add}}+\mathbb{Z}^n, \sigma_{\text{scale}}}^{\mathcal{N}} \Big| \left( y = \frac{k}{t+(k-\psi)} \right) \right] dk.$$

Taking the proper scaling factor gives

$$D_{t,\epsilon,\psi,B,\delta}^{\text{alternative}} = \frac{1}{\lambda(B)} \int_B \left[ (1/\sigma_{\text{scale}}) \circ D_{\mathbf{x}+\mathbf{x}_{\text{add}}+\mathbb{Z}^n, \sigma_{\text{scale}}}^{\mathcal{N}} \Big| \left( y = \frac{k}{t+(k-\psi)} \right) \right] dk.$$

We will prove that for any fixed $k$, letting $\mathbf{x}'_k \sim (1/\sigma_{\text{scale}}) \circ D_{\mathbf{x}+\mathbf{x}_{\text{add}}+\mathbb{Z}^n, \sigma_{\text{scale}}}^{\mathcal{N}} \Big| \left( y = \frac{k}{t+(k-\psi)} \right)$, that

$$P_{\mathbf{x}'_k\text{'s}}(u) = (1 \pm O(\delta)) P_{D_{k+(t+k-\psi)\mathbb{Z}, \sigma_{\text{signal}}}^{\mathcal{N}} \star D_{\sigma_{\text{noise}}}^{\mathcal{N}}}(u).$$

We use $\sigma'_{\text{add}}$ and $\sigma'_{\text{scale}}$ to denote the specific values of $\sigma_{\text{add}}$ and $\sigma_{\text{scale}}$ for $y = \frac{k}{t+(k-\psi)}$. Then

$$D_{\mathbf{x}'_k} = (1/\sigma_{\text{scale}}) \circ D_{\mathbf{x}+\mathbf{x}_{\text{add}}+\mathbb{Z}^n, \sigma_{\text{scale}}}^{\mathcal{N}} \Big| \left( y = \frac{k}{t+(k-\psi)} \right)$$

$$= (1/\sigma_{\text{scale}}) \circ D_{\text{mod}_1(\mathbf{x}+\mathbf{x}_{\text{add}})+\mathbb{Z}^n, \sigma_{\text{scale}}}^{\mathcal{N}} \Big| \left( y = \frac{k}{t+(k-\psi)} \right)$$

$$= (1/\sigma'_{\text{scale}}) \circ D_{\text{mod}_1(\mathbf{x}+\mathbf{x}'_{\text{add}})+\mathbb{Z}^n, \sigma'_{\text{scale}}}^{\mathcal{N}} \Big| \left( y = \frac{k}{t+(k-\psi)} \right),$$

where $\mathbf{x}'_{\text{add}} \sim D_{\mathbb{R}^n, \sigma'_{\text{add}}}^{\mathcal{N}}$. Now we will attempt to reason about this random process and replace it with something equivalent. Letting $\mathbf{x}_{\text{new}} = \text{mod}_1(\mathbf{x} + \mathbf{x}'_{\text{add}})$ and $z_{\text{new}} = -\langle \mathbf{x}'_{\text{add}}, \mathbf{s} \rangle + z$, we notice that

$$y = \text{mod}_1(\langle \mathbf{x}_{\text{new}}, \mathbf{s} \rangle + z_{\text{new}}).$$

We show that $\mathbf{x}_{\text{new}}$ and $z_{\text{new}}$ are independent before conditioning on $y$, therefore we can instead consider the random process as sampling independent $\mathbf{x}_{\text{new}}$ and $z_{\text{new}}$ and then conditioning on $y$. Note that $\mathbf{x}_{\text{new}} = \text{mod}_1(\mathbf{x} + \mathbf{x}'_{\text{add}}) \sim U(\mathbb{R}_1^n)$. If we condition on any fixed $\mathbf{x}'_{\text{add}}$, the conditional distribution of $\mathbf{x}_{\text{new}}$ is always $U(\mathbb{R}_1^n)$, therefore $\mathbf{x}'_{\text{add}}$ and $\mathbf{x}_{\text{new}}$ are independent. Since both $\mathbf{x}'_{\text{add}}$ and $z$ are independent of $\mathbf{x}_{\text{new}}$, we have that $z_{\text{new}} = -\langle \mathbf{x}'_{\text{add}}, \mathbf{s} \rangle + z$ is also independent of $\mathbf{x}_{\text{new}}$. Therefore,

$$D_{\mathbf{x}'_k} = (1/\sigma'_{\text{scale}}) \circ D_{\mathbf{x}_{\text{new}}+\mathbb{Z}^n, \sigma'_{\text{scale}}}^{\mathcal{N}} \Big| (\text{mod}_1(\langle \mathbf{x}_{\text{new}}, \mathbf{s} \rangle + z_{\text{new}}) = y'),$$

where we can interpret $\mathbf{x}_{\text{new}}$ and $z_{\text{new}}$ as independent samples $\mathbf{x}_{\text{new}} \sim U(\mathbb{R}_1^n)$ and $z_{\text{new}} \sim D_{\sqrt{\sigma^2 + \|\mathbf{s}\|_2^2 \sigma'^2_{\text{add}}}}^{\mathcal{N}}$.

Now let $\mathbf{w} \sim (1/\sigma'_{\text{scale}}) \circ D_{\mathbf{x}_{\text{new}}+\mathbb{Z}^n, \sigma'_{\text{scale}}}^{\mathcal{N}}$. Since $\mathbf{x}_{\text{new}} \sim U(\mathbb{R}_1^n)$, we have $D_{\mathbf{x}_{\text{new}}+\mathbb{Z}^n, \sigma'_{\text{scale}}}^{\mathcal{N}} = D_{\mathbb{R}_1^n, \sigma'_{\text{scale}}}^{\text{expand}}$. From the lower bound of $\sigma_{\text{scale}}$ in Observation C.2 and Condition (iii) in Condition 3.1, we have

$$\sigma'_{\text{scale}} \geq \frac{1}{2(t+k-\psi)\sqrt{n}} \geq \frac{1}{4t\sqrt{n}} \geq \sqrt{c \log(n/\delta)}.$$

Therefore, applying Fact A.4 yields

$$P_{\mathbf{w}}(\mathbf{u}) = (1 \pm O(\delta)) P_{D^{\mathcal{N}}_{\mathbb{R}^n,1}}(\mathbf{u}) \,.$$

Since $\mathbf{s} \in \{\pm 1\}^n$ and $\mathrm{mod}_1(\mathbf{x}_{\mathrm{new}}) = \mathrm{mod}_1(\sigma'_{\mathrm{scale}}\mathbf{w})$, we have that

$$\mathrm{mod}_1(\langle \mathbf{x}_{\mathrm{new}}, \mathbf{s} \rangle) = \mathrm{mod}_1(\langle \sigma'_{\mathrm{scale}}\mathbf{w}, \mathbf{s} \rangle) \,.$$

We can now rewrite the distribution as

$$
\begin{aligned}
D_{\mathbf{x}'_k} &= (1/\sigma'_{\mathrm{scale}}) \circ D^{\mathcal{N}}_{\mathrm{mod}_1(\mathbf{x}+\mathbf{x}'_{\mathrm{add}})+\mathbb{Z}^n, \sigma'_{\mathrm{scale}}} \;\bigg|\; \left( y = \frac{k}{t+(k-\psi)} \right) \\
&= (1/\sigma'_{\mathrm{scale}}) \circ D^{\mathcal{N}}_{\mathbf{x}_{\mathrm{new}}+\mathbb{Z}^n, \sigma'_{\mathrm{scale}}} \;\bigg|\; \left( \mathrm{mod}_1(\langle \mathbf{x}_{\mathrm{new}}, \mathbf{s} \rangle + z_{\mathrm{new}}) = \frac{k}{t+(k-\psi)} \right) \\
&= \mathbf{w} \;\bigg|\; \left( \mathrm{mod}_1(\langle \sigma'_{\mathrm{scale}}\mathbf{w}, \mathbf{s} \rangle + z_{\mathrm{new}}) = \frac{k}{t+(k-\psi)} \right) \\
&= \mathbf{w} \;\bigg|\; \left( \mathrm{mod}_1(\sigma'_{\mathrm{scale}}\|\mathbf{s}\|_2 \mathbf{w}^{\mathbf{s}} + z_{\mathrm{new}}) = \frac{k}{t+(k-\psi)} \right) \,,
\end{aligned}
$$

where

$$P_{\mathbf{w}}(\mathbf{u}) = (1 \pm O(\delta)) P_{D^{\mathcal{N}}_{\mathbb{R}^n,1}}(\mathbf{u}) \,,$$

and

$$z_{\mathrm{new}} \sim D^{\mathcal{N}}_{\sqrt{\sigma^2 + \|\mathbf{s}\|_2^2 \sigma'^2_{\mathrm{add}}}}$$

are independently sampled (this follows from $\mathbf{x}_{\mathrm{new}}$ and $z_{\mathrm{new}}$ being independent, since $\mathbf{w}$ solely depends on $\mathbf{x}_{\mathrm{new}}$). The above form allows us the explicitly express the PDF function as follows:

$$
\begin{aligned}
P_{\mathbf{x}'^{\mathbf{s}}_k}(u) &\propto P_{\mathbf{w}^{\mathbf{s}}}(u) \sum_{i \in \mathbb{Z}+\frac{k}{t+(k-\psi)}} P_{z_{\mathrm{new}}}(i - \sigma'_{\mathrm{scale}}\|\mathbf{s}\|_2 u) \\
&\propto (1 \pm O(\delta)) \sum_{i \in \mathbb{Z}+\frac{k}{t+(k-\psi)}} P_{D^{\mathcal{N}}_1}(u) P_{D^{\mathcal{N}}_{\sqrt{\sigma^2+\|\mathbf{s}\|_2^2\sigma'^2_{\mathrm{add}}}}}(i - \sigma'_{\mathrm{scale}}\|\mathbf{s}\|_2 u) \\
&\propto (1 \pm O(\delta)) \sum_{i \in \sigma'^{-1}_{\mathrm{scale}} n^{-1/2}\left(\mathbb{Z}+\frac{k}{t+(k-\psi)}\right)} P_{D^{\mathcal{N}}_1}(u) P_{D^{\mathcal{N}}_{\sqrt{\frac{\sigma^2/n+\sigma'^2_{\mathrm{add}}}{\sigma'^2_{\mathrm{scale}}}}}}(i - u) \,.
\end{aligned}
$$

Letting $\alpha \stackrel{\mathrm{def}}{=} \sqrt{\frac{\sigma^2/n+\sigma'^2_{\mathrm{add}}}{\sigma'^2_{\mathrm{scale}}}}$, we notice that $\alpha = \sqrt{1/\mathrm{SR} - 1}$ from Observation C.2. Thus, we can write

$$
\begin{aligned}
P_{\mathbf{x}'^{\mathbf{s}}_k}(u) &\propto (1 \pm O(\delta)) \sum_{i \in \sigma'^{-1}_{\mathrm{scale}} n^{-1/2}\left(\mathbb{Z}+\frac{k}{t+(k-\psi)}\right)} P_{D^{\mathcal{N}}_1}(u) P_{D^{\mathcal{N}}_\alpha}(i - u) \\
&\propto (1 \pm O(\delta)) \sum_{i \in \sigma'^{-1}_{\mathrm{scale}} n^{-1/2}\left(\mathbb{Z}+\frac{k}{t+(k-\psi)}\right)} \exp\left(-\pi\left(u^2 + \frac{(i-u)^2}{\alpha^2}\right)\right) \\
&\propto (1 \pm O(\delta)) \sum_{i \in \sigma'^{-1}_{\mathrm{scale}} n^{-1/2}\left(\mathbb{Z}+\frac{k}{t+(k-\psi)}\right)} \exp\left(-\pi\left(\frac{(\alpha^2+1)u^2}{\alpha^2} + \frac{i^2}{\alpha^2} - \frac{2iu}{\alpha^2}\right)\right) \\
&\propto (1 \pm O(\delta)) \sum_{i \in \sigma'^{-1}_{\mathrm{scale}} n^{-1/2}\left(\mathbb{Z}+\frac{k}{t+(k-\psi)}\right)} \exp\left(-\pi\left(\frac{i^2}{\alpha^2+1} + \frac{\left((\alpha^2+1)^{1/2}u - (\alpha^2+1)^{-1/2}i\right)^2}{\alpha^2}\right)\right) \\
&\propto (1 \pm O(\delta)) \sum_{i \in \sigma'^{-1}_{\mathrm{scale}} n^{-1/2}\left(\mathbb{Z}+\frac{k}{t+(k-\psi)}\right)} \exp\left(-\pi\left(\frac{\left((\alpha^2+1)^{-1}i\right)^2}{(\alpha^2+1)^{-1}} + \frac{\left(u - (\alpha^2+1)^{-1}i\right)^2}{\alpha^2(\alpha^2+1)^{-1}}\right)\right) \\
&\propto (1 \pm O(\delta)) \sum_{i \in \sigma'^{-1}_{\mathrm{scale}} n^{-1/2}\left(\mathbb{Z}+\frac{k}{t+(k-\psi)}\right)} P_{D^{\mathcal{N}}_{(\alpha^2+1)^{-1/2}}}((\alpha^2+1)^{-1}i) P_{D^{\mathcal{N}}_{\alpha(\alpha^2+1)^{-1/2}}}(u - (\alpha^2+1)^{-1}i) \,.
\end{aligned}
$$

Notice that $(\alpha^2 + 1)^{-1} = \mathrm{SR}$ from Observation C.2. Thus, we get

$$P_{\mathbf{x}_k'^{\mathbf{s}}}(u) \propto (1 \pm O(\delta)) \sum_{i \in \sigma_{\mathrm{scale}}^{-1} n^{-1/2}\left(\mathbb{Z} + \frac{k}{t+(k-\psi)}\right)} P_{D^{\mathcal{N}}_{\sqrt{\mathrm{SR}}}}(\mathrm{SR}i) P_{D^{\mathcal{N}}_{\sqrt{1-\mathrm{SR}}}}(\mathrm{SR}i - u)$$

$$\propto (1 \pm O(\delta)) \sum_{i \in \mathrm{SR}\sigma_{\mathrm{scale}}^{-1} n^{-1/2}\left(\mathbb{Z} + \frac{k}{t+(k-\psi)}\right)} P_{D^{\mathcal{N}}_{\sqrt{\mathrm{SR}}}}(i) P_{D^{\mathcal{N}}_{\sqrt{1-\mathrm{SR}}}}(i - u) .$$

Since $\sigma_{\mathrm{scale}} = \frac{\mathrm{SR}}{(t+k-\psi)\sqrt{n}}$ from Step 3 of Algorithm 1,

$$P_{\mathbf{x}_k'^{\mathbf{s}}}(u) \propto (1 \pm O(\delta)) \sum_{i \in k+(t+k-\psi)\mathbb{Z}} P_{D^{\mathcal{N}}_{\sqrt{\mathrm{SR}}}}(i) P_{D^{\mathcal{N}}_{\sqrt{1-\mathrm{SR}}}}(i - u) .$$

Notice the expression here is propotional to the convolution of $D^{\mathcal{N}}_{k+(t+k-\psi)\mathbb{Z}, \sqrt{\mathrm{SR}}}$ and $D^{\mathcal{N}}_{\sqrt{1-\mathrm{SR}}}$. Therefore, we have

$$P_{\mathbf{x}_k'^{\mathbf{s}}}(u) = (1 \pm O(\delta)) P_{D^{\mathcal{N}}_{k+(t+k-\psi)\mathbb{Z}, \sqrt{\mathrm{SR}}} \star D^{\mathcal{N}}_{\sqrt{1-\mathrm{SR}}}}(u)$$

$$= (1 \pm O(\delta)) P_{D^{\mathcal{N}}_{k+(t+k-\psi)\mathbb{Z}, \sigma_{\mathrm{signal}}} \star D^{\mathcal{N}}_{\sigma_{\mathrm{noise}}}}(u) .$$

Now we can plug this back and calculate $\left[D^{\mathrm{alternative}}_{t,\epsilon,\psi,B,\delta}\right]^{\mathbf{s}}$, as follows:

$$P_{\left[D^{\mathrm{alternative}}_{t,\epsilon,\psi,B,\delta}\right]^{\mathbf{s}}}(u) = \frac{1}{\lambda(B)} \int_B P_{\mathbf{x}_k'^{\mathbf{s}}}(u) dk$$

$$= \frac{1}{\lambda(B)} (1 \pm O(\delta)) \int_B P_{D^{\mathcal{N}}_{k+(t+k-\psi)\mathbb{Z}, \sigma_{\mathrm{signal}}} \star D^{\mathcal{N}}_{\sigma_{\mathrm{noise}}}}(u) dk .$$

Note that inside the integration is the convolution of two distributions, $D^{\mathcal{N}}_{k+(t+k-\psi)\mathbb{Z}, \sigma_{\mathrm{signal}}}$ and $D^{\mathcal{N}}_{\sigma_{\mathrm{noise}}}$. Since $D^{\mathcal{N}}_{\sigma_{\mathrm{noise}}}$ is independent of $k$, we can interpret it as added after the integration. Therefore,

$$P_{\left[D^{\mathrm{alternative}}_{t,\epsilon,\psi,B,\delta}\right]^{\mathbf{s}}}(u) = (1 \pm O(\delta)) P_{\left[\int_B \frac{1}{\lambda(B)} D^{\mathcal{N}}_{k+(t+k-\psi)\mathbb{Z}, \sigma_{\mathrm{signal}}} dk \star D^{\mathcal{N}}_{\sigma_{\mathrm{noise}}}\right]}(u)$$

$$= (1 \pm O(\delta)) P_{D' \star D^{\mathcal{N}}_{\sigma_{\mathrm{noise}}}}(u) ,$$

where

$$\sigma_{\mathrm{signal}} = \sqrt{\mathrm{SR}} ,$$

and

$$\sigma_{\mathrm{noise}} = \sqrt{1 - \mathrm{SR}} = 2(t + \epsilon)\sigma .$$

This proves property (i).

**Now we prove Property (ii).** We prove that for any fixed $y = \frac{k}{t+(k-\psi)}$, letting $\mathbf{x}_k' \sim D^{\mathrm{alternative}}_{t,\epsilon,\psi,B,\delta} \mid \left(y = \frac{k}{t+(k-\psi)}\right)$, then $\mathbf{x}_k'^{\perp \mathbf{s}}$ is nearly independent of $\mathbf{x}_k'^{\mathbf{s}}$ in the sense that for any $l \in \mathbb{R}$ and $\mathbf{u} \in \mathbb{R}^{n-1}$, we have that

$$P_{\mathbf{x}_k'^{\perp \mathbf{s}} \mid \mathbf{x}_k'^{\mathbf{s}} = l}(\mathbf{u}) = (1 \pm O(\delta)) P_{D^{\mathcal{N}}_{\mathbb{R}^{n-1}, 1}}(\mathbf{u}) .$$

Following the same notation we used for proving Property (i), we can rewrite the random process as

$$(1/\sigma_{\mathrm{scale}}) \circ D^{\mathcal{N}}_{\mathbf{x}+\mathbf{x}_{\mathrm{add}}+\mathbb{Z}^n, \sigma_{\mathrm{scale}}} \mid \left(y = \frac{k}{t+(k-\psi)}\right) = \mathbf{w} \mid \left(\mathrm{mod}_1(\sigma_{\mathrm{scale}}' \|\mathbf{s}\|_2 \mathbf{w}^{\mathbf{s}} + z_{\mathrm{new}}) = \frac{k}{t+(k-\psi)}\right) ,$$

where

$$P_{\mathbf{w}}(\mathbf{u}) = (1 \pm O(\delta)) P_{D^{\mathcal{N}}_{\mathbb{R}^n, 1}}(\mathbf{u}) ,$$

and

$$z_{\mathrm{new}} \sim D^{\mathcal{N}}_{\sqrt{\sigma^2 + \|\mathbf{s}\|_2^2 \sigma_{\mathrm{add}}'^2}}$$

are independently sampled (before conditioning). Therefore, we have that

$$P_{\mathbf{x}_k'^{\perp \mathbf{s}}|\mathbf{x}_k'^{\mathbf{s}}=l}(\mathbf{u}) = P_{\left[\mathbf{w}^{\perp \mathbf{s}}\middle|\left(\mathrm{mod}_1\left(\sigma_{\mathrm{scale}}'\|\mathbf{s}\|_2\mathbf{w}^{\mathbf{s}}+z_{\mathrm{new}}\right)=\frac{k}{t+(k-\psi)}\right)\wedge(\mathbf{w}^{\mathbf{s}}=l)\right]}(\mathbf{u})$$

$$= P_{\left[\mathbf{w}^{\perp \mathbf{s}}\middle|\left(\mathrm{mod}_1\left(\sigma_{\mathrm{scale}}'\|\mathbf{s}\|_2 l+z_{\mathrm{new}}\right)=\frac{k}{t+(k-\psi)}\right)\wedge(\mathbf{w}^{\mathbf{s}}=l)\right]}(\mathbf{u})$$

$$= P_{[\mathbf{w}^{\perp \mathbf{s}}|\mathbf{w}^{\mathbf{s}}=l]}(\mathbf{u}) .$$

Recall that the PDF of $\mathbf{w}$ is pointwise close to a Gaussian, as shown earlier in proof for Property (i), therefore, we obtain

$$P_{\mathbf{x}_k'^{\perp \mathbf{s}}|\mathbf{x}_k'^{\mathbf{s}}=l}(\mathbf{u}) = P_{\mathbf{w}^{\perp \mathbf{s}}|\mathbf{w}^{\mathbf{s}}=l}(\mathbf{u}) = (1 \pm O(\delta)) P_{D^{\mathcal{N}}_{\mathbb{R}^{n-1},1}}(\mathbf{u}) .$$

Considering that $\mathbf{x}' \sim D_{t,\epsilon,\psi,B,\delta}^{\mathrm{alternative}}$ is a mixture of $\mathbf{x}_k' \sim D_{t,\epsilon,\psi,B,\delta}^{\mathrm{alternative}} \mid \left(y = \frac{k}{t+(k-\psi)}\right)$ for different values of $k$, it follows that

$$P_{\mathbf{x}'^{\perp \mathbf{s}}|\mathbf{x}'^{\mathbf{s}}=l}(\mathbf{u}) = (1 \pm O(\delta)) P_{D^{\mathcal{N}}_{\mathbb{R}^{n-1},1}}(\mathbf{u}) .$$

$\square$

We also give the following claim obtained from Lemma 3.5, which will be useful when we prove the Massart condition in Lemma 3.9.

**Claim C.4.** *The exact PDF function of the distribution $D'$ in Lemma 3.5 is*

$$P_{D'}(u) = \frac{\Theta(t)}{\lambda(B)} \sum_{i \in \mathbb{Z}} \frac{1}{|i+1|} \mathbf{1}(u \in it + \psi + (i+1)(B-\psi)) \rho_{\sigma_{\mathrm{signal}}}(u) .$$

*Proof.* This follows from expanding the expression of $D'$ in Lemma 3.5. We have

$$P_{D'}(u) = P_{\left[\frac{1}{\lambda(B)} \int_B D^{\mathcal{N}}_{k+(t+k-\psi)\mathbb{Z},\sigma_{\mathrm{signal}}} dk\right]}(u)$$

$$= \frac{1}{\lambda(B)} \int_B P_{\left[D^{\mathcal{N}}_{k+(t+k-\psi)\mathbb{Z},\sigma_{\mathrm{signal}}}\right]}(u) dk .$$

Notice that

$$P_{\left[D^{\mathcal{N}}_{k+(t+k-\psi)\mathbb{Z},\sigma_{\mathrm{signal}}}\right]}(u) = \mathbf{1}\left(u \in k + (t+k-\psi)\mathbb{Z}\right) \frac{\rho_{\sigma_{\mathrm{signal}}}(u)}{\rho_{\sigma_{\mathrm{signal}}}(k + (t+k-\psi)\mathbb{Z})} .$$

Then using that $\sigma_{\mathrm{signal}} = \sqrt{\mathrm{SR}} \geq 1/2$ (Observation C.1) and $\frac{1}{t\sqrt{n}} \geq \sqrt{c \log(n/\delta)}$ for a sufficiently large $c$ (Condition 3.1), we have that $\sigma_{\mathrm{signal}}/(t+k-\psi) \geq \sqrt{\tilde{c} \log(1/\delta)}$ for a sufficiently large $\tilde{c}$. After that, an application of Lemma A.9 gives that

$$\rho_{\sigma_{\mathrm{signal}}}(k+(t+k-\psi)\mathbb{Z}) = (t+k-\psi)^{-1} \rho_{\sigma_{\mathrm{signal}}/(t+k-\psi)}(k/(t+k-\psi)+\mathbb{Z}) = (1\pm O(\delta))(t+k-\psi)^{-1} .$$

Plugging this back, we obtain

$$P_{D'}(u) = \frac{1 \pm O(\delta)}{\lambda(B)} \int_B (t+k-\psi) \mathbf{1}\left(u \in k + (t+k-\psi)\mathbb{Z}\right) \rho_{\sigma_{\mathrm{signal}}}(u) dk$$

$$= \frac{1 \pm O(\delta)}{\lambda(B)} \int_B (t+k-\psi) \sum_{i \in \mathbb{Z}} \left[\mathbf{1}(u = k + (t+k-\psi)i)\right] \rho_{\sigma_{\mathrm{signal}}}(u) dk$$

$$= \frac{\Theta(t)}{\lambda(B)} \sum_{i \in \mathbb{Z}} \left[\int_B \mathbf{1}(u = k + (t+k-\psi)i) dk\right] \rho_{\sigma_{\mathrm{signal}}}(u)$$

$$= \frac{\Theta(t)}{\lambda(B)} \sum_{i \in \mathbb{Z}} \frac{1}{|i+1|} \mathbf{1}(u \in it + \psi + (i+1)(B-\psi)) \rho_{\sigma_{\mathrm{signal}}}(u) . \qquad \square$$

### C.3.3 Proof of Lemma 3.6

*Proof of Lemma 3.6.* We first review the definition of $D^{\mathrm{null}}_{t,\epsilon,\psi,B,\delta}$. Let $(\mathbf{x}, y)$ be drawn from the null hypothesis case of the LWE problem. This means that we have $\mathbf{x} \sim U(\mathbb{R}^n_1)$ and $y \sim U(\mathbb{R}_1)$ independently. In this case, we reject unless $y = \frac{k}{t+(k-\psi)}$ for some $k \in B$. Then the secondary rejection step rejects with probability $1 - \frac{t^2}{(t+k-\psi)^2}$. We calculate $\sigma_{\mathrm{add}}$ and $\sigma_{\mathrm{scale}}$ which only depend on $y$, and output a sample

$$\mathbf{x}' \sim (1/\sigma_{\mathrm{scale}}) \circ D^{\mathcal{N}}_{\mathbf{x}+\mathbf{x}_{\mathrm{add}}+\mathbb{Z}^n, \sigma_{\mathrm{scale}}},$$

where $\mathbf{x}_{\mathrm{add}} \sim D^{\mathcal{N}}_{\mathbb{R}^n, \sigma_{\mathrm{add}}}$. Therefore, the distribution $D^{\mathrm{null}}_{t,\epsilon,\psi,B,\delta}$ is a mixture of

$$(1/\sigma_{\mathrm{scale}}) \circ D^{\mathcal{N}}_{\mathbf{x}+\mathbf{x}_{\mathrm{add}}+\mathbb{Z}^n, \sigma_{\mathrm{scale}}} \mid (y = y'),$$

for different values of $y'$.

We prove that for any fixed $y'$, the following holds:

$$P_{\left[(1/\sigma_{\mathrm{scale}}) \circ D^{\mathcal{N}}_{\mathbf{x}+\mathbf{x}_{\mathrm{add}}+\mathbb{Z}^n, \sigma_{\mathrm{scale}}} \mid (y=y')\right]}(\mathbf{u}) = (1 \pm O(\delta)) \cdot P_{D^{\mathcal{N}}_{\mathbb{R}^n,1}}(\mathbf{u}).$$

We use $\sigma'_{\mathrm{add}}$ and $\sigma'_{\mathrm{scale}}$ to denote the specific values of $\sigma_{\mathrm{add}}$ and $\sigma_{\mathrm{scale}}$ for $y = y'$. Then

$$
\begin{aligned}
(1/\sigma_{\mathrm{scale}}) \circ D^{\mathcal{N}}_{\mathbf{x}+\mathbf{x}_{\mathrm{add}}+\mathbb{Z}^n, \sigma_{\mathrm{scale}}} \mid (y = y') &= (1/\sigma_{\mathrm{scale}}) \circ D^{\mathcal{N}}_{\mathrm{mod}_1(\mathbf{x}+\mathbf{x}_{\mathrm{add}})+\mathbb{Z}^n, \sigma_{\mathrm{scale}}} \mid (y = y')\\
&= (1/\sigma'_{\mathrm{scale}}) \circ D^{\mathcal{N}}_{\mathrm{mod}_1(\mathbf{x}+\mathbf{x}'_{\mathrm{add}})+\mathbb{Z}^n, \sigma'_{\mathrm{scale}}} \mid (y = y')\\
&= (1/\sigma'_{\mathrm{scale}}) \circ D^{\mathcal{N}}_{\mathrm{mod}_1(\mathbf{x}+\mathbf{x}'_{\mathrm{add}})+\mathbb{Z}^n, \sigma'_{\mathrm{scale}}},
\end{aligned}
$$

where $\mathbf{x}'_{\mathrm{add}} \sim D^{\mathcal{N}}_{\mathbb{R}^n,\sigma'_{\mathrm{scale}}}$. Since $\mathbf{x}$ is always drawn from $U(\mathbb{R}^n_1)$ and $\mathbf{x}'_{\mathrm{add}}$ is independent of $\mathbf{x}$, then $\mathrm{mod}_1(\mathbf{x} + \mathbf{x}'_{\mathrm{add}})$ is also drawn from $U(\mathbb{R}^n_1)$. Thus, we can write

$$D^{\mathcal{N}}_{\mathbf{x}+\mathbf{x}'_{\mathrm{add}}+\mathbb{Z}^n, \sigma'^2_{\mathrm{scale}}} = D^{\mathrm{expand}}_{\mathbb{R}^n_1, \sigma'_{\mathrm{scale}}}.$$

Applying Fact A.4, the lower bound of $\sigma_{\mathrm{scale}}$ in Observation C.2 and Condition (iii) in Condition 3.1 yields

$$P_{D^{\mathrm{expand}}_{\mathbb{R}^n_1, \sigma'_{\mathrm{scale}}}}(\mathbf{u}) = (1 \pm O(\delta)) \cdot P_{D^{\mathcal{N}}_{\mathbb{R}^n, \sigma'_{\mathrm{scale}}}}(\mathbf{u}).$$

Plugging this back, we obtain that

$$P_{\left[(1/\sigma_{\mathrm{scale}}) \circ D^{\mathcal{N}}_{\mathbf{x}+\mathbf{x}_{\mathrm{add}}+\mathbb{Z}^n, \sigma_{\mathrm{scale}}} \mid (y=y')\right]}(\mathbf{u}) = (1 \pm O(\delta)) \cdot P_{D^{\mathcal{N}}_{\mathbb{R}^n,1}}(\mathbf{u}).$$

Then, since $D^{\mathrm{null}}_{t,\epsilon,\psi,B,\delta}$ is a mixture of $(1/\sigma_{\mathrm{scale}}) \circ D^{\mathcal{N}}_{\mathbf{x}+\mathbf{x}_{\mathrm{add}}+\mathbb{Z}^n, \sigma_{\mathrm{scale}}} \mid (y = y')$ for different $y'$, we get that

$$P_{D^{\mathrm{null}}_{t,\epsilon,\psi,B,\delta}}(\mathbf{u}) = (1 \pm O(\delta)) \cdot P_{D^{\mathcal{N}}_{\mathbb{R}^n,1}}(\mathbf{u}).$$

$\square$

### C.3.4 Proof of Lemma 3.9

Before we prove Lemma 3.9, we restate the definition for $B_-$ as a refresher. $B_-$ is defined by carving out $O(t/\epsilon)$ many empty slots in $[t/2, t]$ in order to make the problematic part of $\mathbf{x^s}|y = +1$ to fit in (see Figure 4 and Figure 5). To define $B_-$, we need to first define a mapping $g$ that maps a location to the corresponding place we need to carve out on $B_-$. The function $g : \mathbb{R} - [-1.5t, 0.5t] \mapsto [0.5t, t]$ is defined as follows: for $i \in \mathbb{Z}$ and $b \in \mathbb{R}_t$, we have that

$$g(it + t/2 + b) \stackrel{\mathrm{def}}{=} \begin{cases} \frac{b}{i+1} + t/2 & \text{if } i \geq 0; \\ \frac{b-t}{i+2} + t/2 & \text{if } i < 0. \end{cases}$$

Then $B_-$ is defined as follows:

$$
\begin{aligned}
B_- \stackrel{\mathrm{def}}{=}\ & [t/2, t/2 + \epsilon]\\
& - \bigcup_{i=\frac{t}{2\epsilon}-1}^{\frac{t}{\epsilon}-1} g([it - 2c'\epsilon, it]) - \bigcup_{i=\frac{t}{2\epsilon}-1}^{\frac{t}{\epsilon}-1} g([it + (i+1)\epsilon, it + (i+1)\epsilon + 2c'\epsilon])\\
& - \bigcup_{i=-\frac{t}{\epsilon}-1}^{-\frac{t}{2\epsilon}-1} g([it + (i+1)\epsilon - 2c'\epsilon, it + (i+1)\epsilon]) - \bigcup_{i=-\frac{t}{\epsilon}-1}^{-\frac{t}{2\epsilon}-1} g([it, it + 2c'\epsilon]).
\end{aligned}
$$

*Proof of Lemma 3.9.* We will prove that there is a distribution $D^{\text{truncated}}$ such that $d_{\text{TV}}(D^{\text{truncated}}, D_{\text{PTF}}^{\text{alternative}}) = O(\delta/m')$ and there is a degree-$O(t/\epsilon)$ PTF $\text{sign}(p(\cdot))$ such that

1. $\mathbf{Pr}_{(\mathbf{x},y) \sim D^{\text{truncated}}}[\text{sign}(p(\mathbf{x})) \neq y] = \exp(-\Omega(t^4/\epsilon^2))$; and,

2. $D^{\text{truncated}}$ satisfies the $O(\eta)$ Massart condition with respect to $\text{sign}(p(\mathbf{x}))$.

We first give some high level intuition for $D^{\text{truncated}}$. First, we recall that $D_{\text{PTF}}^{\text{alternative}}$ is the $1 - \eta : \eta$ mixture of $D_{t,\epsilon,\psi_+,B_+,\delta}^{\text{alternative}}$ and $D_{t,\epsilon,\psi_-,B_-,\delta}^{\text{alternative}}$ with $+1$ and $-1$ labels, respectively [3]. Also we recall that both $D_{t,\epsilon,\psi_+,B_+,\delta}^{\text{alternative}}$ and $D_{t,\epsilon,\psi_-,B_-,\delta}^{\text{alternative}}$ are noisy "hidden direction discrete Gaussians" which are close to continuous Gaussian on all except the hidden direction $\mathbf{s}$, and on the hidden direction, they are close to a linear combination of discrete Gaussian plus an extra continuous Gaussian noise (as shown in Lemma 3.5). The idea here is to truncate that extra continuous Gaussian noise on the hidden direction to obtain $D^{\text{truncated}}$.

Now we formally define $D^{\text{truncated}}$ in the following way. We will first define distributions $D_+^{\text{truncated}}$ and $D_-^{\text{truncated}}$ such that

$$d_{\text{TV}}(D_+^{\text{truncated}}, D_{t,\epsilon,\psi_+,B_+,\delta}^{\text{alternative}}) = O(\delta/m') ,$$

and

$$d_{\text{TV}}(D_-^{\text{truncated}}, D_{t,\epsilon,\psi_-,B_-,\delta}^{\text{alternative}}) = O(\delta/m') .$$

Then we take $D^{\text{truncated}}$ as the $1 - \eta : \eta$ mixture of $D_+^{\text{truncated}}$ and $D_-^{\text{truncated}}$ with $+1$ and $-1$ labels respectively.

We define $D_+^{\text{truncated}}$ below; $D_-^{\text{truncated}}$ is defined analogously. We specify $D_+^{\text{truncated}}$ to satisfy the following requirement: Let $\mathbf{x}' \sim D_+^{\text{truncated}}$ and $\mathbf{x} \sim D_{t,\epsilon,\psi_+,B_+,\delta}^{\text{alternative}}$. For any $l \in \mathbb{R}$, we specify the conditional distribution of $P_{\mathbf{x}'\perp\mathbf{s}|\mathbf{x}'\mathbf{s}=l}$ to be:

$$P_{\mathbf{x}'\perp\mathbf{s}|\mathbf{x}'\mathbf{s}=l} = P_{\mathbf{x}\perp\mathbf{s}|\mathbf{x}\mathbf{s}=l} .$$

Since the conditional distributions on $\perp \mathbf{s}$ are the same, to define $D_+^{\text{truncated}}$, it remains to specify $[D_+^{\text{truncated}}]^{\mathbf{s}}$. According to Lemma 3.5 property (i), $\left[D_{t,\epsilon,\psi_+,B_+,\delta}^{\text{alternative}}\right]^{\mathbf{s}}$ is pointwise close to the convolutional sum of a distribution $D'$ and noise drawn from $D_{\sigma_{\text{noise}}}^{\mathcal{N}}$. Here we use $D'_+$ (resp. $D'_-$) to denote the corresponding $D'$ for $\left[D_{t,\epsilon,\psi_+,B_+,\delta}^{\text{alternative}}\right]^{\mathbf{s}}$ (resp. $\left[D_{t,\epsilon,\psi_-,B_-,\delta}^{\text{alternative}}\right]^{\mathbf{s}}$). This is

$$P_{\left[D_{t,\epsilon,\psi_+,B_+,\delta}^{\text{alternative}}\right]^{\mathbf{s}}}(u) = (1 \pm O(\delta))P_{D'_+ \star D_{\sigma_{\text{noise}}}^{\mathcal{N}}}(u) .$$

We define a truncated noise distribution $\mathcal{N}^{\text{truncated}}$ to replace $D_{\sigma_{\text{noise}}}^{\mathcal{N}}$, as follows:

$$P_{\mathcal{N}^{\text{truncate}}}(u) \propto \begin{cases} P_{D_{\sigma_{\text{noise}}}^{\mathcal{N}}}(u) & \text{if } |u| \leq c'\epsilon, \\ 0 & \text{otherwise}, \end{cases}$$

where $c'$ is the parameter in Condition 3.1. We now define $[D_+^{\text{truncated}}]^{\mathbf{s}}$ as pointwise close to the convolution sum of the distribution $D'_+$ and noise from $\mathcal{N}^{\text{truncated}}$, namely,

$$P_{[D_+^{\text{truncated}}]^{\mathbf{s}}}(u) \propto \frac{P_{\left[D_{t,\epsilon,\psi_+,B_+,\delta}^{\text{alternative}}\right]^{\mathbf{s}}}(u)}{P_{D'_+ \star D_{\sigma_{\text{noise}}}^{\mathcal{N}}}(u)} P_{D'_+ \star \mathcal{N}^{\text{truncated}}}(u) ,$$

where $\frac{P_{\left[D_{t,\epsilon,\psi_+,B_+,\delta}^{\text{alternative}}\right]^{\mathbf{s}}}(u)}{P_{D'_+ \star D_{\sigma_{\text{noise}}}^{\mathcal{N}}}(u)} = (1 \pm O(\delta))$. $D_-^{\text{truncated}}$ is defined in the same manner.

Now we bound $d_{\text{TV}}(D_+^{\text{truncated}}, D_{t,\epsilon,\psi_+,B_+,\delta}^{\text{alternative}})$. We first define

$$f(u) = \frac{P_{\left[D_{t,\epsilon,\psi_+,B_+,\delta}^{\text{alternative}}\right]^{\mathbf{s}}}(u)}{P_{D'_+ \star D_{\sigma_{\text{noise}}}^{\mathcal{N}}}(u)} P_{D'_+ \star \mathcal{N}^{\text{truncated}}}(u) ,$$

---

[3]That is, $D_{\text{PTF}}^{\text{alternative}}$ is the joint distribution of $(\mathbf{x}, y)$ such that with probability $1 - \eta$ we sample $\mathbf{x} \sim D_{t,\epsilon,\psi_+,B_+,\delta}^{\text{alternative}}$ and $y = +1$; and with probability $\eta$ we sample $\mathbf{x} \sim D_{t,\epsilon,\psi_-,B_-,\delta}^{\text{alternative}}$ and $y = -1$

and notice that
$$P_{D_+^{\text{truncated}}}(u) = \left(\int_{\mathbb{R}} f(u)du\right)^{-1} f(u) .$$

We then bound $\int_{\mathbb{R}} f(u)du$. Note that

$$\int_{\mathbb{R}} \left| f(u) - P_{\left[D_{t,\epsilon,\psi_+,B_+,\delta}^{\text{alternative}}\right]^{\mathbf{s}}}(u) \right| du = (1 \pm O(\delta)) \int_{\mathbb{R}} \left| \int_{\mathbb{R}} P_{D_+}(t) P_{\mathcal{N}^{\text{truncated}}}(u-t)dt - \int_{\mathbb{R}} P_{D_+}(t) P_{D_{\sigma_{\text{noise}}}^{\mathcal{N}}}(u-t)dt \right| du$$

$$\leq (1 \pm O(\delta)) \int_{\mathbb{R}} P_{D_+}(t) \int_{\mathbb{R}} \left| P_{\mathcal{N}^{\text{truncated}}}(u-t) - P_{D_{\sigma_{\text{noise}}}^{\mathcal{N}}}(u-t) \right| dt du$$

$$= (1 \pm O(\delta)) \int_{\mathbb{R}} \left| P_{\mathcal{N}^{\text{truncated}}}(u) - P_{D_{\sigma_{\text{noise}}}^{\mathcal{N}}}(u) \right| du$$

$$= (1 \pm O(\delta)) 2d_{\text{TV}}(\mathcal{N}^{\text{truncated}}, D_{\sigma_{\text{noise}}}^{\mathcal{N}}) .$$

Therefore, we have that $\int_{\mathbb{R}} f(u)du = 1 \pm O(d_{\text{TV}}(\mathcal{N}^{\text{truncated}}, D_{\sigma_{\text{noise}}}^{\mathcal{N}}))$. To bound our objective $d_{\text{TV}}(D_+^{\text{truncated}}, D_{t,\epsilon,\psi_+,B_+,\delta}^{\text{alternative}})$, we have

$$d_{\text{TV}}(D_+^{\text{truncated}}, D_{t,\epsilon,\psi_+,B_+,\delta}^{\text{alternative}}) = d_{\text{TV}}\left(\left[D_+^{\text{truncated}}\right]^{\mathbf{s}}, \left[D_{t,\epsilon,\psi_+,B_+,\delta}^{\text{alternative}}\right]^{\mathbf{s}}\right)$$

$$= \frac{1}{2} \int_{\mathbb{R}} \left| P_{\left[D_+^{\text{truncated}}\right]^{\mathbf{s}}}(u) - P_{\left[D_{t,\epsilon,\psi_+,B_+,\delta}^{\text{alternative}}\right]^{\mathbf{s}}}(u) \right| du .$$

By the triangle inequality, we can write

$$d_{\text{TV}}(D_+^{\text{truncated}}, D_{t,\epsilon,\psi_+,B_+,\delta}^{\text{alternative}}) \leq \frac{1}{2} \int_{\mathbb{R}} \left| f(u) - P_{\left[D_{t,\epsilon,\psi_+,B_+,\delta}^{\text{alternative}}\right]^{\mathbf{s}}}(u) \right| du + \frac{1}{2} \int_{\mathbb{R}} \left| f(u) - P_{\left[D_+^{\text{truncated}}\right]^{\mathbf{s}}}(u) \right| du$$

$$= O\left(d_{\text{TV}}\left(\mathcal{N}^{\text{truncated}}, D_{\sigma_{\text{noise}}}^{\mathcal{N}}\right)\right) + \frac{1}{2} \left| \int_{\mathbb{R}} f(u)du - 1 \right|$$

$$= O\left(d_{\text{TV}}\left(\mathcal{N}^{\text{truncated}}, D_{\sigma_{\text{noise}}}^{\mathcal{N}}\right)\right)$$

$$= O\left(\exp\left(-\frac{(c'\epsilon)^2}{2\sigma_{\text{noise}}^2}\right)\right) .$$

From Condition (iv) in Condition 3.1 and $\sigma_{\text{noise}} = 2(t+\epsilon)\sigma$, we have that $\frac{(c'\epsilon)^2}{2\sigma_{\text{noise}}^2} \geq \log(m'/\delta)$; thus,

$$d_{\text{TV}}(D_+^{\text{truncated}}, D_{t,\epsilon,\psi_+,B_+,\delta}^{\text{alternative}}) = O(\delta/m') .$$

The same holds for $D_-^{\text{truncated}}$. Therefore, we have that

$$d_{\text{TV}}(D^{\text{truncated}}, D_{\text{PTF}}^{\text{alternative}}) = (1-\eta)d_{\text{TV}}(D_+^{\text{truncated}}, D_{t,\epsilon,\psi_+,B_+,\delta}^{\text{alternative}}) + \eta d_{\text{TV}}(D_-^{\text{truncated}}, D_{t,\epsilon,\psi_-,B_-,\delta}^{\text{alternative}})$$

$$= O(\delta/m') .$$

Before we continue with the rest of the proof, we require the following claim about the support of these distributions.

**Claim C.5.** *The support of distribution $D'_+$ is*

$$\bigcup_{i\in\mathbb{Z}_-} [it + (i+1)\epsilon, it] \cup \bigcup_{i\in\mathbb{Z}_+} [it, it+(i+1)\epsilon] ,$$

*and the support of distribution $D_+^{\text{truncated}}$ is*

$$\bigcup_{i\in\mathbb{Z}_-} [it + (i+1)\epsilon - c'\epsilon, it + c'\epsilon] \cup \bigcup_{i\in\mathbb{Z}_+} [it - c'\epsilon, it + (i+1)\epsilon + c'\epsilon] .$$

*Similarly, the support of distribution $D'_-$ is **a subset of***

$$\bigcup_{i\in\mathbb{Z}_-} [it + t/2 + (i+1)\epsilon, it + t/2] \cup \bigcup_{i\in\mathbb{Z}_+} [it + t/2, it + t/2 + (i+1)\epsilon] ,$$

*and the support of distribution $D_+^{\text{truncated}}$ is **a subset of***

$$\bigcup_{i\in\mathbb{Z}_-} [it + t/2 + (i+1)\epsilon - c'\epsilon, it + t/2 + c'\epsilon] \cup \bigcup_{i\in\mathbb{Z}_+} [it + t/2 - c'\epsilon, it + t/2 + (i+1)\epsilon + c'\epsilon] .$$

The above claim directly follows from Claim C.4, the definition of these distributions, and the fact that the support of $\mathcal{N}^{\text{truncated}}$ is $[-c'\epsilon, c'\epsilon]$.

Now we continue with our proof. With our definition of $D^{\text{truncated}}$, it remains to prove that there is a PTF $\text{sign}(p(\cdot))$ satisfying the following:

1. $\mathbf{Pr}_{(\mathbf{x},y)\sim D^{\text{truncated}}}[\text{sign}(p(\mathbf{x})) \neq y] = \exp(-\Omega(t^4/\epsilon^2))$; and

2. $D^{\text{truncated}}$ satisfies the $O(\eta)$ Massart condition with respect to $\text{sign}(p(\mathbf{x}))$.

From the way we defined $D^{\text{truncated}}$ and Property (ii) of Lemma 3.5, we know that for $\mathbf{u} \in \mathbb{R}^n$

$$\mathbf{Pr}_{D^{\text{truncated}}}[y = +1 \mid \mathbf{x} = \mathbf{u}] = (1 \pm O(\delta))\mathbf{Pr}_{D^{\text{truncated}}}[y = +1 \mid \mathbf{x^s} = \mathbf{u^s}] .$$

Since $(1 \pm O(\delta)) = O(1)$ and $t/\epsilon = \Omega(1)$, therefore, it suffices to prove these statements on the subspace spanned by $\mathbf{s}$: there is a degree-$O(t/\epsilon)$ PTF $\text{sign}(p(\langle \mathbf{s}, \cdot \rangle))$ such that

1. $\mathbf{Pr}_{(x,y)\sim [D^{\text{truncated}}]^{\mathbf{s}}}[\text{sign}(p(\langle \mathbf{s}, x \rangle)) \neq y] = \exp(-\Omega(t^4/\epsilon^2))$; and

2. $\left[D^{\text{truncated}}\right]^{\mathbf{s}}$ satisfies the $O(\eta)$ Massart condition with respect to $\text{sign}(p(\langle \mathbf{s}, x \rangle))$,

where we abuse the notation slightly and use $\left[D^{\text{truncated}}\right]^{\mathbf{s}}$ to denote the $1 - \eta : \eta$ mixture of $[D_+^{\text{truncated}}]^{\mathbf{s}}$ and $[D_-^{\text{truncated}}]^{\mathbf{s}}$ with $+1$ and $-1$ labels repectively. We consider a degree-$O(t/\epsilon)$ PTF $\text{sign}(p(u))$ such that

$$\text{sign}(p(u)) = \begin{cases} +1 & \text{if } u \in \bigcup_{i\in\mathbb{Z}_-} [it + (i+1)\epsilon - c'\epsilon, it + c'\epsilon] \cup \bigcup_{i\in\mathbb{Z}_+} [it - c'\epsilon, it + (i+1)\epsilon + c'\epsilon] ; \\ -1 & \text{otherwise} . \end{cases}$$

Notice that according to Claim C.7, the domain with value $+1$ can be written as the union of $O(t/\epsilon)$ many intervals, thus the above function must be realizable by taking $p$ as a degree-$O(t/\epsilon)$ PTF.

For item 1, from Claim C.5, the support of $+1$ samples ($\left[D_+^{\text{truncated}}\right]^{\mathbf{s}}$) is

$$\bigcup_{i\in\mathbb{Z}_-} [it + (i+1)\epsilon - c'\epsilon, it + c'\epsilon] \cup \bigcup_{i\in\mathbb{Z}_+} [it - c'\epsilon, it + (i+1)\epsilon + c'\epsilon] ,$$

and the support for -1 samples ($\left[D_-^{\text{truncated}}\right]^{\mathbf{s}}$) is a subset of

$$\bigcup_{i\in\mathbb{Z}_-} [it + t/2 + (i+1)\epsilon - c'\epsilon, it + t/2 + c'\epsilon] \cup \bigcup_{i\in\mathbb{Z}_+} [it + t/2 - c'\epsilon, it + t/2 + (i+1)\epsilon + c'\epsilon] .$$

(The above follows from Claim C.4.) For $|i + 1| \leq t/(2\epsilon) - 1$, these intervals have length at most $t/2 - \epsilon + 2c'\epsilon < t/2$, thus do not overlap (for sufficiently small $c'$). Therefore, this PTF makes no mistake for $i \in [-t/(2\epsilon), t/(2\epsilon) - 2]$, i.e., at least for $u \in [-t^2/(2\epsilon), t^2/(2\epsilon) - 2t]$. Thus, its error is at most $\exp(-\Omega(t^2/\epsilon)^2)$.

For item 2, we show the following:

(i) If $u \in \bigcup_{i\in\mathbb{Z}_-} [it + (i+1)\epsilon - c'\epsilon, it + c'\epsilon] \cup \bigcup_{i\in\mathbb{Z}_+} [it - c'\epsilon, it + (i+1)\epsilon + c'\epsilon]$ then

$$\mathbf{Pr}_{(x,y)\sim[D^{\text{truncated}}]^{\mathbf{s}}}[y = +1|x = u] \geq 1 - O(\eta) .$$

(ii) Otherwise,
$$P_{[D^{\text{truncated}}]^{\mathbf{s}}|(y=+1)}(u) = 0 .$$

Item (ii) is straightforward, as from Claim C.5, the support of $D^{\text{truncated}} \mid (y = +1)$ (i.e., $D_+^{\text{truncated}}$) is $\bigcup_{i\in\mathbb{Z}_-} [it + (i+1)\epsilon - c'\epsilon, it + c'\epsilon] \cup \bigcup_{i\in\mathbb{Z}_+} [it - c'\epsilon, it + (i+1)\epsilon + c'\epsilon]$. Since all the support is in item (i), in the "otherwise" case, it must be the case that $P_{[D^{\text{truncated}}]^{\mathbf{s}}|(y=+1)}(u) = 0$.

For item (i), we recall that $\left[D^{\text{truncated}}\right]^{\mathbf{s}}$ is the $1 - \eta : \eta$ mixture of $D_+^{\text{truncated}}$ and $D_-^{\text{truncated}}$ with $+1$ and $-1$ labels respectively. Therefore, we need to show for

$$u \in \bigcup_{i \in \mathbb{Z}_-} [it + (i+1)\epsilon - c'\epsilon, it + c'\epsilon] \cup \bigcup_{i \in \mathbb{Z}_+} [it - c'\epsilon, it + (i+1)\epsilon + c'\epsilon] \, ,$$

we have that

$$P_{D_+^{\text{truncated}}}(u) = \Omega(P_{D_-^{\text{truncated}}}(u)) \, .$$

Since $D_+^{\text{truncated}}$ and $D_-^{\text{truncated}}$ are pointwise close to the convolution sums of $D'_+$ or $D'_-$ and $\mathcal{N}^{\text{truncated}}$ respectively, the above amounts to

$$\int_{-c'\epsilon}^{c'\epsilon} P_{\mathcal{N}^{\text{truncated}}}(u') P_{D'_+}(u - u') du' = \Omega\left(\int_{-c'\epsilon}^{c'\epsilon} P_{\mathcal{N}^{\text{truncated}}}(u') P_{D'_-}(u - u') du'\right) \, .$$

Therefore, it suffices to prove for $u \in \bigcup_{i \in \mathbb{Z}_-} [it + (i+1)\epsilon - 2c'\epsilon, it + 2c'\epsilon] \cup \bigcup_{i \in \mathbb{Z}_+} [it - 2c'\epsilon, it + (i+1)\epsilon + 2c'\epsilon]$, it holds that

$$P_{D'_+}(u) = \Omega(P_{D'_-}(u)) \, .$$

We will consider

$$u \in \bigcup_{i \in \mathbb{Z}_-} [it + (i+1)\epsilon - 2c'\epsilon, it + 2c'\epsilon] \cup \bigcup_{i \in \mathbb{Z}_+} [it - 2c'\epsilon, it + (i+1)\epsilon + 2c'\epsilon]$$

for three different cases:

(A) $|i+1| \leq t/(2\epsilon) - 1$,

(B) $t/(2\epsilon) - 1 < |i+1| \leq t/\epsilon$, and,

(C) $|i+1| > t/\epsilon$.

For case (A), we begin by recalling that the support of $D'_-$ is

$$\bigcup_{i \in \mathbb{Z}_-} [it + t/2 + (i+1)\epsilon, it + t/2] \cup \bigcup_{i \in \mathbb{Z}_+} [it + t/2, it + t/2 + (i+1)\epsilon] \, .$$

For case (A), the support of both $D'_+$ and $D'_-$ are intervals with length at most $t/2 - \epsilon$ (see Claim C.5). Thus, the gaps between $D'_+$ intervals and $D'_-$ intervals are at least $\epsilon$. Any $u$ for case (A) is at most $2c'\epsilon$ far from a $D'_+$ support interval, where $c'$ is sufficiently small, combining with the fact that gaps between $D'_+$ intervals and $D'_-$ intervals are at least $\epsilon$, this makes $u$ not inside the support of $D'_-$. Therefore, for any $u$ in case (A), we have that $P_{D'_-}(u) = 0$, which implies $P_{D'_+}(u) = \Omega(P_{D'_-}(u))$.

For case (B), we note by Claim C.4, we have

$$P_{D'_-}(u) = \frac{\Theta(t)}{\lambda(B_-)} \sum_{i \in \mathbb{Z}} \frac{1}{|i+1|} \mathbf{1}(u \in it + t/2 + (i+1)(B_- - t/2)) \rho_{\sigma_{\text{signal}}}(u) \, ,$$

and

$$P_{D'_+}(u) = \frac{\Theta(t)}{\lambda(B_+)} \sum_{i \in \mathbb{Z}} \frac{1}{|i+1|} \mathbf{1}(u \in it + (i+1)[0, \epsilon]) \rho_{\sigma_{\text{signal}}}(u) \, .$$

Note that since $t/(2\epsilon - 1) < |i+1| \leq t/\epsilon$, thus these intervals has lenghth at most $t$, the $D_+$ (resp. $D_-$) intervals do not overlap with other $D_+$ (resp. $D_-$) intervals. Therefore taking $i = \lfloor u/t \rfloor$,

$$P_{D'_-}(u) = \frac{\Theta(t)}{\lambda(B_-)|i+1|} \mathbf{1}\left(u \in \bigcup_{i \in \mathbb{Z}} it + t/2 + (i+1)(B_- - t/2)\right) \, ,$$

and

$$P_{D'_+}(u) = \frac{\Theta(t)}{\lambda(B_+)|i+1|} \mathbf{1}\left(u \in \bigcup_{i \in \mathbb{Z}} it + (i+1)[0, \epsilon]\right) \, .$$

Let $S_- = \bigcup_{i\in\mathbb{Z}} it + t/2 + (i+1)(B_- - t/2)$ and $S_+ = \bigcup_{i\in\mathbb{Z}} it + (i+1)[0,\epsilon]$. Therefore it suffices to show that

$$\lambda(B_-) = \Omega(\lambda(B_+)) = \Omega(\epsilon) ,$$

and for any $u$ in case (B)

$$u \in S_- \text{ implies } u \in S_+ .$$

$\lambda(B_-) = \Omega(\epsilon)$ follows from the definition of $B_-$, since

$$B_- \overset{\text{def}}{=} [t/2, t/2 + \epsilon] - \bigcup_{i=\frac{t}{2\epsilon}-1}^{\frac{t}{\epsilon}-1} g([it - 2c'\epsilon, it]) - \bigcup_{i=\frac{t}{2\epsilon}-1}^{\frac{t}{\epsilon}-1} g([it + (i+1)\epsilon, it + (i+1)\epsilon + 2c'\epsilon])$$

$$- \bigcup_{i=-\frac{t}{\epsilon}-1}^{-\frac{t}{2\epsilon}-1} g([it + (i+1)\epsilon - 2c'\epsilon, it + (i+1)\epsilon]) - \bigcup_{i=-\frac{t}{\epsilon}-1}^{-\frac{t}{2\epsilon}-1} g([it, it + 2c'\epsilon]) .$$

Therefore, recall the definition of the mapping $g$, we get that

$$\lambda(B_-) \geq \epsilon - 4\sum_{i=\frac{t}{2\epsilon}}^{\frac{t}{\epsilon}} \frac{2c'\epsilon}{|i|} \geq \epsilon - 4\frac{t}{\epsilon}\frac{2c'\epsilon}{\frac{t}{2\epsilon}} = \epsilon - 16c'\epsilon = \Omega(\epsilon) ,$$

where the last equality follows from the fact that $c'$ is a sufficiently small constant. To prove for any $u$ in case (B)

$$u \in S_- \text{ implies } u \in S_+ .$$

We prove the contrapositive, let $u$ be in case (B) and $u \notin S_+$, then calculations show

$$u \in \bigcup_{i=\frac{t}{2\epsilon}-1}^{\frac{t}{\epsilon}-1} g([it - 2c'\epsilon, it]) \cup \bigcup_{i=\frac{t}{2\epsilon}-1}^{\frac{t}{\epsilon}-1} g([it + (i+1)\epsilon, it + (i+1)\epsilon + 2c'\epsilon]) \tag{2}$$

$$\cup \bigcup_{i=-\frac{t}{\epsilon}-1}^{-\frac{t}{2\epsilon}-1} g([it + (i+1)\epsilon - 2c'\epsilon, it + (i+1)\epsilon]) \cup \bigcup_{i=-\frac{t}{\epsilon}-1}^{-\frac{t}{2\epsilon}-1} g([it, it + 2c'\epsilon]) . \tag{3}$$

Notice that those intervals are exactly the intervals we carved out. Thus, $u \notin S_-$ . (We note that the intuition behind this is the following. For the interval $[g(a), g(b)]$ we carved out on $B-$, we made $[a, b]$ missing from the support of $D'_-$. So what we did can be thought of as carving these intervals to make the supports of $D'_+$ and $D'_-$ having $2c'\epsilon$ gaps between each other for $t/(2\epsilon) - 1 < |i+1| \leq t/\epsilon$. This ensures that after applying the $\mathcal{N}^{\text{truncated}}$ noise, their supports still do not overlap.) This completes the proof for case (B).

It remains to analyze case (C). By Claim C.4, we have that

$$\frac{P_{D'_+}(u)}{P_{D'_-}(u)} = \frac{\frac{\Theta(t)}{\lambda(B_+)} \sum_{i\in\mathbb{Z}} \frac{1}{|i+1|} \mathbf{1}(u \in it + \psi_+ + (i+1)(B_+ - \psi_+))\rho_{\sigma_{\text{signal}}}(u)}{\frac{\Theta(t)}{\lambda(B_-)} \sum_{i\in\mathbb{Z}} \frac{1}{|i+1|} \mathbf{1}(u \in it + \psi_- + (i+1)(B_- - \psi_-))\rho_{\sigma_{\text{signal}}}(u)}$$

$$= \Omega\left( \frac{\sum_{i\in\mathbb{Z}} \frac{1}{|i+1|} \mathbf{1}(u \in it + \psi_+ + (i+1)(B_+ - \psi_+))}{\sum_{i\in\mathbb{Z}} \frac{1}{|i+1|} \mathbf{1}(u \in it + \psi_- + (i+1)(B_- - \psi_-))} \right) ,$$

where the second equality follows from $\lambda(B_-) = \Omega(\lambda(B_+)) = \Omega(\epsilon)$. Since $B_-$ is a subset of $[t/2, t/2 + \epsilon]$, thus

$$\sum_{i\in\mathbb{Z}} \frac{1}{|i+1|} \mathbf{1}(u \in it + \psi_- + (i+1)(B_- - \psi_-)) \leq \sum_{i\in\mathbb{Z}} \frac{1}{|i+1|} \mathbf{1}\left(u \in it + t/2 + (i+1)\left([t/2, t/2+\epsilon] - t/2\right)\right) ,$$

therefore

$$\frac{P_{D'_+}(u)}{P_{D'_-}(u)} = \Omega\left( \frac{\sum_{i\in\mathbb{Z}} \frac{1}{|i+1|} \mathbf{1}(u \in it + (i+1)[0,\epsilon])}{\sum_{i\in\mathbb{Z}} \frac{1}{|i+1|} \mathbf{1}(u \in it + t/2 + (i+1)([t/2, t/2+\epsilon] - t/2))} \right) .$$

Thus we just need to show the following claim to finish case (C).

**Claim C.6.** *For*

$$u \in \bigcup_{i \in \{\mathbb{Z}_- : |i+1| > t/\epsilon\}} [it + (i+1)\epsilon - 2c'\epsilon, it + 2c'\epsilon] \cup \bigcup_{i \in \{\mathbb{Z}_+ : |i+1| > t/\epsilon\}} [it - 2c'\epsilon, it + (i+1)\epsilon + 2c'\epsilon] \,,$$

*we have that*

$$\frac{\sum_{i \in \mathbb{Z}} \frac{1}{|i+1|} \mathbf{1}(u \in it + (i+1)[0, \epsilon])}{\sum_{i \in \mathbb{Z}} \frac{1}{|i+1|} \mathbf{1}(u \in it + t/2 + (i+1)([t/2, t/2 + \epsilon] - t/2))} = \Omega(1) \,.$$

The idea is to determine for both the numerator and denominator which values of $i$ cause the corresponding indicator function to be 1. Notice for $(-\infty, -t^2/\epsilon - t) \cup (t^2/\epsilon - t, \infty)$, the indicator function in the numerator $it + (i+1)[0, \epsilon/2]$ will have length at least $t$. Thus for $u \in (-\infty, -t^2/\epsilon - 2t + 2c'\epsilon) \cup (t^2/\epsilon - 2c'\epsilon, \infty)$, $u$ must be in at least one of these intervals in the numerator. In particular, there must be $j \leq k$ such that the non-zero terms in the numerator correspond to the indicator terms with $j \leq i \leq k$. Therefore, the numerator is $\sum_{i=j}^{k} \frac{1}{|i+1|}$.

Then we examine the terms for the denominator. For convenience, we use $l_i^{\text{numerator}}$ and $r_i^{\text{numerator}}$ (resp. $l_i^{\text{denominator}}$ and $r_i^{\text{denominator}}$) to denote the left endpoint and right endpoint of the $i$th term in the numerator (resp. denominator). Since the $j - 1$th term in the numerator is not satisfied, $u > r_{j-1}^{\text{numerator}}$. Since $r_{j-1}^{\text{numerator}} > r_{j-2}^{\text{denominator}}$, we know that $u > r_{j-2}^{\text{denominator}}$, thus the $j - 2$th term in the denominator cannot be satisfied. Then, since the $(k + 1)$-st term in the numerator is not satisfied, we know that $u < l_{k+1}^{\text{numerator}}$. Since $l_{k+1}^{\text{numerator}} < l_{k+1}^{\text{denominator}}$, $u < l_{k+1}^{\text{denominator}}$, thus the $(k + 1)$-st term in the denominator is not satisfied. Now we have that the denominator is at most $\sum_{i=j-1}^{k} \frac{1}{|i+1|}$.

Then, we just need to prove for any $j, k \in \mathbb{Z}$ and $j \leq k$ (since from the $j$-th to $k$-th term in the numerator are satisfied, it must be that $k < -1$ or $j > -1$), it holds that

$$\frac{\sum_{i=j}^{k} \frac{1}{|i+1|}}{\sum_{i=j-1}^{k} \frac{1}{|i+1|}} = \Omega(1) \,.$$

This is easy to see, since the denominator has at most one term more than the numerator and all terms are of comparable size. This completes the proof for Lemma 3.9. $\qquad\square$

Now we prove the following helper lemma for Lemma 3.9.

**Claim C.7.**

$$\bigcup_{i \in \mathbb{Z}_-} [it + (i+1)\epsilon - c'\epsilon, it + c'\epsilon] \cup \bigcup_{i \in \mathbb{Z}_+} [it - c'\epsilon, it + (i+1)\epsilon + c'\epsilon]$$

*can be equivalently written as the union of $O(t/\epsilon)$ many intervals on $\mathbb{R}$.*

*Proof.* We notice that for $|i + 1| \geq t/\epsilon$, the interval terms inside the union has length at least $t$, therefore they overlap with each other and cover the whole space. Thus, we can rewrite the expression as

$$\bigcup_{i \in \mathbb{Z}_-} [it + (i+1)\epsilon - c'\epsilon, it + c'\epsilon] \cup \bigcup_{i \in \mathbb{Z}_+} [it - c'\epsilon, it + (i+1)\epsilon + c'\epsilon]$$

$$= (-\infty, -t^2/\epsilon - t + c'\epsilon]$$

$$\cup \bigcup_{i \in \{\mathbb{Z}_- | i+1| < t/\epsilon\}} [it + (i+1)\epsilon - c'\epsilon, it + c'\epsilon] \cup \bigcup_{i \in \{\mathbb{Z}_+ | i+1| < t/\epsilon\}} [it - c'\epsilon, it + (i+1)\epsilon + c'\epsilon]$$

$$\cup [-t^2/\epsilon - t - c'\epsilon, \infty) \,,$$

which is the union over $O(t/\epsilon)$ many intervals. $\qquad\square$

### C.3.5 Null Hypothesis Analysis for Algorithm 2

Here we analyze the null hypothesis case by showing that for $(\mathbf{x}, y) \sim D_{\mathrm{PTF}}^{\mathrm{null}}$, $\mathbf{x}$ and $y$ are almost independent. For $D_{\mathrm{PTF}}^{\mathrm{null}}$, we establish the following property.

**Lemma C.8.** *For any $\mathbf{u} \in \mathbb{R}^n$, we have that*

$$\mathbf{Pr}_{(\mathbf{x},y) \sim D_{\mathrm{PTF}}^{\mathrm{null}}}[y = +1 \mid \mathbf{x} = \mathbf{u}] = (1 \pm O(\delta))(1 - \eta) \,.$$

*Proof.* This lemma follows directly from Lemma 3.6, since

$$P_{\mathbf{x}|(y=+1)}(\mathbf{u}), P_{\mathbf{x}|(y=-1)}(\mathbf{u}) = (1 \pm O(\delta)) P_{D_{\mathbb{R}^n,1}^{\mathcal{N}}}(\mathbf{u}) \,.$$

Also the marginal probability is $\mathbf{Pr}[y = +1] = 1 - \eta$. Therefore, for any $\mathbf{u} \in \mathbb{R}^n$,

$$\mathbf{Pr}_{(\mathbf{x},y) \sim D_{\mathrm{PTF}}^{\mathrm{null}}}[y = +1 \mid \mathbf{x} = \mathbf{u}] = (1 \pm O(\delta))(1 - \eta) \,.$$

$\square$

### C.4 Proof of Theorem 3.2

*Proof of Theorem 3.2.* We give a reduction from $\mathrm{LWE}(m, \mathbb{R}_1^n, \{\pm 1\}^n, \mathcal{N}_\sigma, \mathrm{mod}_1)$ to $\mathrm{Massart}(m', N, \eta, \mathrm{OPT})$. Suppose there is an algorithm $A$ for the problem $\mathrm{Massart}(m', N, \eta, \mathrm{OPT})$ with $2\epsilon'$ distinguishing advantage such that (a) if the input is from the alternative hypothesis case, $A$ outputs "alternative hypothesis" with probability $\alpha$; and (b) if the input is from the null hypothesis case, $A$ outputs "alternative hypothesis" with probability at most $\alpha - 2\epsilon'$.

Given $m$ samples from $\mathrm{LWE}(m, \mathbb{R}_1^n, \{\pm 1\}^n, \mathcal{N}_\sigma, \mathrm{mod}_1)$, we run Algorithm 2 with input parameters $(m', t, \epsilon, \delta, \eta')$, where $\delta$ and $\eta'/\eta$ are sufficiently small positive constants. If Algorithm 2 fails, we output "alternative hypothesis" with probability $\alpha$ and "null hypothesis" with probability $1 - \alpha$. Otherwise, Algorithm 2 succeeds, and we get $m'$ many i.i.d. samples $(\mathbf{x}_1, y_1), (\mathbf{x}_2, y_2) \cdots, (\mathbf{x}_{m'}, y_{m'}) \in \mathbb{R}^n \times \{\pm 1\}$ from $D_{\mathrm{PTF}}$ (they are i.i.d. according to Observation 3.8). With $d = c(t/\epsilon)$, where $c$ is a sufficiently large constant, we apply the degree-$d$ Veronese mapping $V_d : \mathbb{R}^n \mapsto \mathbb{R}^N$ on the samples to get $(V_d(\mathbf{x}_1), y_1), (V_d(\mathbf{x}_2), y_2) \cdots, (V_d(\mathbf{x}_{m'}), y_{m'}) \in \mathbb{R}^N \times \{\pm 1\}$. Then we give these samples to $A$ and argue that the above process can distinguish $\mathrm{LWE}(m, \mathbb{R}_1^n, \{\pm 1\}^n, \mathcal{N}_\sigma, \mathrm{mod}_1)$ with at least $\epsilon' - O(\delta)$ advantage.

We let $D_{\mathrm{LTF}}$ denote the distribution of $(V_d(\mathbf{x}), y)$, where $(\mathbf{x}, y) \sim D_{\mathrm{PTF}}$ ($D_{\mathrm{PTF}}$ depends on $\mathbf{s}$ and which case the original LWE samples come from). We note that the samples we provide to the Massart distinguisher are $m'$ many i.i.d. samples from $D_{\mathrm{LTF}}$. We claim that if we can prove the following items, then we are done.

1. In the alternative (resp. null) hypothesis case, Algorithm 2 in the above process fails with probability at most $1/2$.

2. **Completeness:** If the LWE instance is from the alternative hypothesis case, then $D_{\mathrm{LTF}}$ has at most $O(\delta/m') \, d_{\mathrm{TV}}$ distance from a distribution $D$ and there is an LTF $h$ such that

    (a) $\mathbf{Pr}_{(\mathbf{x},y) \sim D}[h(\mathbf{x}) \neq y] = \exp(-\Omega(t^4/\epsilon^2))$, and
    (b) $D$ satisfies the $\eta$ Massart condition with respect to to the LTF $h$.

3. **Soundness:** If the LWE instance is from the null hypothesis case, then $R_{\mathrm{OPT}}(D_{\mathrm{LTF}}) = \Omega(\eta)$.

Suppose we have proved the above items. In the alternative hypothesis case, the above process outputs "alternative hypothesis" with probability at least $\alpha - O(\delta)$, since $D$ is at most $O(\delta/m')$ in $d_{\mathrm{TV}}$ distance from $D_{\mathrm{LTF}}$. Then, in the null hypothesis case, the process outputs "alternative hypothesis" with probability at most $\alpha - \epsilon'$. Therefore, the distinguishing advantage is at least $\epsilon' - O(\delta)$. Now it remains to prove these three items.

For the first item, recall that for each sample, Algorithm 2 uses Algorithm 1 to generate a new sample. If Algorithm 1 accepts, it outputs a new sample. According to Lemma C.3, it accepts with probability $\Omega(\frac{\lambda(B)(t-\psi)}{t^2})$. Under our choice of $\psi$, this is at least $\Omega(\epsilon/t)$ (since we have shown that $\lambda(B_+)$, $\lambda(B_-) = \Omega(\epsilon)$ in the proof of Lemma 3.9 and $\psi$ is either $\psi_+ = 0$ or $\psi_- = t/2$). With

$m' = c(\epsilon/t)m$, where $c > 0$ is sufficiently small and $m(\epsilon/t)^2$ is sufficiently large, by applying the Hoeffding bound, one can see that Algorithm 2 succeeds with probability at least $1/2$.

For the second item, suppose that the initial LWE samples are from the alternative hypothesis case. Then, in the proof of Lemma 3.9 (at the beginning of the proof), we have shown that there exists a distribution $D_{\mathrm{PTF}}^{\mathrm{truncated}}$ (denoted by $D^{\mathrm{truncated}}$ in the proof of Lemma 3.9) within total variation distance $O(\delta/m')$ from $D_{\mathrm{PTF}}^{\mathrm{alternative}}$, and a degree-$d$ PTF $\mathrm{sign}(p(\mathbf{x}))$ such that

1. $\mathbf{Pr}_{(\mathbf{x},y)\sim D_{\mathrm{PTF}}^{\mathrm{truncated}}}[\mathrm{sign}(p(\mathbf{x})) \neq y] = \exp(-\Omega(t^4/\epsilon^2)]$; and

2. $D_{\mathrm{PTF}}^{\mathrm{truncated}}$ satisfies the $O(\eta')$ Massart noise condition with respect to $\mathrm{sign}(p(\mathbf{x}))$.

With $\eta'/\eta$ being a sufficiently small constant, the second item satisfies the $\eta$ Massart noise condition. Then, letting $D_{\mathrm{LTF}}^{\mathrm{truncated}}$ denote the distribution of $(V_d(\mathbf{x}), y)$ where $(\mathbf{x}, y) \sim D_{\mathrm{PTF}}^{\mathrm{truncated}}$, one can see that $D_{\mathrm{LTF}}^{\mathrm{truncated}}$ is at most $O(\delta/m')$ in total variation distance from $D_{\mathrm{LTF}}$. Let $h$ denote the corresponding LTF such that $h(V_d(\mathbf{x})) = \mathrm{sign}(p(\mathbf{x}))$. Then, after the Veronese mapping, it must be the case that:

1. $\mathbf{Pr}_{(\mathbf{x},y)\sim D_{\mathrm{LTF}}^{\mathrm{truncated}}}[h(\mathbf{x}) \neq y] = \exp(-\Omega(t^4/\epsilon^2))$; and

2. $D_{\mathrm{LTF}}^{\mathrm{truncated}}$ satisfies the $\eta$ Massart noise condition with respect to $h(\mathbf{x})$.

This gives the second item.

For the third item, if the initial LWE samples are from the null hypothesis case, then according to Lemma C.8, for any $\mathbf{u} \in \mathbb{R}^n$,

$$\mathbf{Pr}_{(\mathbf{x},y)\sim D_{\mathrm{PTF}}^{\mathrm{null}}}[y = +1 \mid \mathbf{x} = \mathbf{u}] = (1 \pm O(\delta))(1 - \eta') .$$

Therefore, $R_{\mathrm{OPT}}(D_{\mathrm{PTF}}) = \Omega(\eta') = \Omega(\eta)$, thus $R_{\mathrm{OPT}}(D_{\mathrm{LTF}}) = \Omega(\eta)$. This completes the proof. $\qquad\square$

# D   Putting Everything Together: Proof of Main Hardness Result

Applying Theorem 3.2 together with Lemma 2.5 yields our main theorem:

**Theorem D.1.** *Under Assumption 2.4, for any $\zeta \in (0,1)$, there exists $\chi > 0$ such that there is no $N^{O(\log^\chi N)}$-time algorithm that solves* Massart $\left(m' = N^{O(\log^\chi N)}, N, \eta = 1/3, \mathrm{OPT} = 1/2^{\log^{1-\zeta} N}\right)$ *with $1/3$ advantage.*

*Proof.* We first take $\chi = 0.01\zeta$. For Lemma 2.5, we take $\beta = 1 - 0.1\zeta$ and $\gamma = -5$. For Theorem 3.2, we take

1. $t = n^{-0.5-0.2\zeta}$,

2. $\epsilon = \Theta(n^{-1.5})$,

3. $\delta$ be a sufficiently small constant,

4. $\eta = 1/3$, and,

5. $m' = c(\epsilon/t)m$, where $c$ is a sufficiently small positive constant.

One can easily check that the conditions in Condition 3.1 are satisfied. According to our parameters of choice, the remaining parameters for the hardness are:

1. $N = n^{O(t/\epsilon)} \leq n^{O(n^{1-0.2\zeta})}$,

2. $\mathrm{OPT} = \exp(-\Omega(t^4/\epsilon^2)) = 2^{-\Omega(n^{1-0.8\zeta})} \leq c2^{-n^{1-\zeta}} \leq c2^{-\log^{1-\zeta} N}$, for any constant c,

3. $m' = c(\epsilon/t)m = \Omega(n^{-\Theta(1-0.2\zeta)}2^{O(n^\beta)}) \geq 2^{O(n^{1-0.11\zeta})} \geq N^{O\left(\log^{\frac{1-0.11\zeta}{1-0.19\zeta}} N\right)} \geq N^{O(\log^\chi N)}$, and

4. the time complexity lower bound is

$$2^{\Omega(n^{1-0.1\zeta})} = N^{\omega(\log^\chi N)} .$$

Therefore, according to Theorem 3.2, there is no $N^{O(\log^\chi N)}$-time algorithm that solves $\mathrm{Massart}(m' = N^{O(\log^\chi N)}, N, \eta = 1/3, \mathrm{OPT} = 1/2^{\log^{1-\zeta} N})$ with 1/3 advantage. This completes the proof. $\qquad\square$

The above theorem gives the following corollary. We note that Corollary D.2 implies our informal Theorem 1.2, and Corollary D.2 has stronger parameters.

**Corollary D.2.** *Under Assumption 2.4, for any $\zeta \in (0,1)$, there exists $\chi > 0$ such that no $N^{O(\log^\chi N)}$ time and sample complexity algorithm can achieve an error of $2^{O(\log^{1-\zeta} N)} \cdot \mathrm{OPT}$ in the task of learning LTFs on $\mathbb{R}^N$ with $\eta = 1/3$ Massart noise.*