# OpenReview forum: "Cryptographic Hardness of Learning Halfspaces with Massart Noise"
_NeurIPS.cc/2022/Conference — NeurIPS 2022 Accept_

### Official Review · Reviewer_k4Eo · 2022-07-09

**Rating:** 6
**Confidence:** 3
**Soundness:** 4 excellent
**Presentation:** 4 excellent
**Contribution:** 3 good

**Summary:**

The paper provides cryptographic hardness results for the problem of learning halfspaces with Massart noise. The authors give a reduction of LWE to Massart halfspace learning and show that no polynomial time algorithm for Massart halfspaces can achieve error better than the one achieved by the existing algorithms. Prior to this work, there were only SQ lower bounds for this task.

**Questions:**

While I find the contribution nice, it is not clear to me how significant is such a hardness result given the existing SQ hardness results for the Massart halfspaces problem. Which algorithmic approaches that bypass the SQ lower bounds could be potentially used to efficiently tackle this task? Is it only Gaussian elimination and some LLL based algorithms?

**Limitations:**

The authors address the limitations and propose future directions.

**Strengths And Weaknesses:**

The paper provides a cryptographic hardness result for the problem of learning Massart halfspaces. Such cryptographic hardness results have been recently obtained for various other learning tasks. Also, recent work has established hardness results for Massart halfspaces in the SQ model. The main result of this work is that, assuming the sub-exponential hardness of the LWE problem, there is no efficient learning algorithm for Massart halfspaces with error better than $\eta$ even if the optimal classifier can achieve much smaller error.

The paper is well-written and the result is quite clear. The technical contribution of the paper is a reduction from continuous LWE to Massart halfspaces. In general, this result is an interesting addition to the literature of robust supervised statistics.

I am in general positive towards acceptance; however, I would like to understand the importance of this hardness result, given the existing SQ lower bounds.

---

> ### Author Response · Authors · 2022-08-02
> **Author Response to Reviewer k4Eo**
>
> We thank the reviewer for the detailed and constructive comments. To answer the question, Gaussian elimination for parity and LLL-based algorithms do not fall into the SQ model, and there are no logical implications that these are the only exceptions. We also fundamentally disagree with the suggestion that the existence of SQ lower bounds detracts from the significance of our computational hardness result. While the SQ model captures a range of natural algorithms, at the end of the day it is a restricted model of computation, i.e., does *not* capture all efficient algorithms. This can be a significant limitation of the SQ model. Here we overcome this limitation, by establishing a lower bound for *all* polynomial time algorithms, subject to a widely believed cryptographic assumption — which is plausibly *worst-case* hard. (There is a quantum reduction from GapSVP to our LWE assumption.)
>
>
> We would also like to give our perspective on the pros and cons between SQ hardness and computational hardness. Broadly speaking, the ultimate goal of lower bounds is ruling out all algorithms for solving a specific task. Therefore, computational hardness results (under a reasonable hardness assumption) are the best for this purpose, as they rule out all efficient algorithms. However, computational hardness results for statistical problems are usually hard to come by for multiple reasons, including 1) one needs the right kind of assumption to start with, and 2) unlike worst-case computational hardness, average-case computational hardness requires more delicate techniques to preserve the background distribution. On the other hand, SQ lower bounds (or other restricted computational models) only rule out restricted classes of algorithms, but are usually easier to obtain and give us a starting point to probe into the structure of these problems. We also want to point out that algorithms like Gaussian elimination and LLL-based algorithms simply do not fall into the SQ model, and there are no logical implications that these are the only exceptions.

---

### Official Review · Reviewer_EeJy · 2022-07-10

**Rating:** 7
**Confidence:** 3
**Soundness:** 3 good
**Presentation:** 3 good
**Contribution:** 3 good

**Summary:**

Learning halfspaces from samples with noisy labels is a classical problem in learning theory. In the Massart noise model with parameter eta, for some unknown halfspace f_w(x) = sign(<x,w>), the samples (x,y) are drawn as x ~ D (for some arbitrary unknown D) and y in {-1,1} such that Pr_{x ~ D}[f_w(x) = y] >= 1-eta. A recent sequence of works has shown how to learn (with polynomial time and sample complexity) a halfspace with error at most eta + eps. But in polynomial samples (with no time complexity restriction) it's possible to learn a halfspace with error at most OPT + eps, where OPT is the error of the best halfspace (it could be that OPT << eta). It remains open whether it's possible to do in polynomial time.

There is evidence via statistical query lower bounds suggesting that this goal is impossible. The current paper presents additional evidence, in the form of a conditional (but unrestricted) lower bound under the Learning With Errors assumption. Specifically, the main result of this paper is that if LWE cannot be solved in strongly subexponential time 2^{n^{1-Omega(1)}}, then no polynomial time learning algorithm can learn halfspaces with eta-Massart noise in polynomial time with error better than Omega(eta), even under the promise that OPT << eta (concretely, eta is constant and OPT < 2^{-log^{1-zeta}(n)} for constant zeta>0).

**Questions:**

line 54: should be 2^{-log^{1-zeta}(n)}?
line 224: why is this exp(-Omega(t^2/epsilon)^2)? Should it be exp(-Omega(t/epsilon)^2)?
line 312: D^{expand} is only defined in the appendix
line 313: there is no Fact A.5, I guess this should be Fact A.4?

**Limitations:**

Yes

**Strengths And Weaknesses:**

The contribution of this paper is original and of interest to the learning theory community. The paper is for the most part cleanly written, although there are a large number of parameters in the construction and perhaps more intuition could be given about them (e.g. Condition 3.1 and the main theorem). The overview of the prior SQ lower bound construction was particularly helpful, although a few parts were tricky to follow (specifically, the claims in lines 223-227). Lemma 3.3 also could use a bit more explanation (e.g. explaining in plaintext the sampling procedure for x'). The proofs are rather technical and I wasn't able to check them all, but the overview of the various obstacles and how they are overcome makes sense.

The only potential weakness is that this paper is strictly of interest to the learning theory segment of the NeurIPS community; also, while technical and certainly non-trivial I am not sure whether the proof techniques are of broader interest. Nonetheless, within learning theory the problem solved by this paper is of sufficient interest that I would recommend acceptance.

---

> ### Author Response · Authors · 2022-08-02
> **Author Response for Reviewer EeJy**
>
> Thanks for the detailed and constructive comments. We would like to address the point about **"while technical and certainly non-trivial I am not sure whether the proof techniques are of broader interest"**. We believe that our result is of broader interest for a number of reasons.
>
> We would like to point out that the algorithmic result of [DGT19] (that we show is essentially optimal in this work) received the outstanding paper award at NeurIPS’19. This could be considered as evidence that the problem we study is of broader interest beyond learning theory.
> At the technical level, some of the tools we introduce may be of interest to establish hardness for other statistical problems.
>
> We also want to mention that there has recently been a line of work that “turns” SQ hardness results into computational hardness results, including [Dan16], [BB20], and [BRST21]. However, these efforts are still in the early stages and we have no general theory behind them (but only case-by-case studies). Viewed from this perspective, our work is the newest development on this topic and we hope it can lead to/be helpful for a more general theory explaining the connection between SQ hardness and computational hardness. This would answer some important high-level questions such as “why do SQ lower bounds usually reliably approximate computational hardness?”.
>
> We would also like to thank the reviewer for pointing out the typos. Yes, Line 54 should be $2^{-\log^{1-\zeta}(N)}$ where $N$ is the dimension after the Veronese mapping. However, Line 224 should be $\exp(-\Omega(t^2/\epsilon)^2)$, as there are typos on lines 215, 217, and 220 where the $t$ in the intervals should be $t^2$. For line 313, we mean Fact 1.4. We apologize for these typos and will fix them in the camera-ready version.
>
> Reference:
>
> [BB20] M. S. Brennan and G. Bresler. Reducibility and statistical-computational gaps from secret leakage. In Conference on Learning Theory, COLT 2020, volume 125 of Proceedings of Machine Learning Research, pages 648–847. PMLR, 2020.
>
> [BRST21] J. Bruna, O. Regev, M. J. Song, and Y. Tang. Continuous LWE. In STOC ’21: 53rd
> Annual ACM SIGACT Symposium on Theory of Computing, pages 694–707. ACM,
> 2021.
>
> [DGT19] I. Diakonikolas, T. Gouleakis, and C. Tzamos. Distribution-independent PAC learning of halfspaces with massart noise. In Advances in Neural Information Processing Systems 32: Annual Conference on Neural Information Processing Systems 2019, NeurIPS 2019, pages 4751–4762, 2019.

---

> > ### Comment · Reviewer_EeJy · 2022-08-06
> > **Thanks**
> >
> > Thanks for the response. I agree with the point about trying to understand when SQ lower bounds predict computational hardness; this paper gives an additional datapoint in that direction.

---

### Official Review · Reviewer_yBUh · 2022-07-11

**Rating:** 8
**Confidence:** 4
**Soundness:** 4 excellent
**Presentation:** 4 excellent
**Contribution:** 4 excellent

**Summary:**

This paper proves a new cryptographic hardness result for the fundamental problem of learning halfspaces with Massart noise in the distribution-free setting. Specifically, they assume subexponential hardness of the Learning with Errors (LWE) problem and show the following: given a distribution consistent with a halfspace except with $\eta$-bounded Massart noise (think of $\eta = 1/3$), and for which the true OPT is almost polynomially small, no polynomial-time algorithm can achieve 0-1 loss better than $\Omega(\eta)$. Technically, they show that even a natural decision version of this learning problem is hard.

The approach taken is a careful reduction from continuous LWE (itself as hard as LWE) to the problem of learning polynomial threshold functions (PTFs) with Massart noise; by the natural monomials feature map, PTFs are just LTFs over a higher-dimensional space. The reduction builds closely on prior work that showed a superpolynomial SQ lower bound for the same problem. At a high level, the final hard Massart distribution that one wishes to generate is roughly similar in both: one wants a distribution of $(x', y')$ that is consistent with a Massart PTF, and such that the two conditional distributions $x' | y' = +1$ and $x' | y' = -1$ are "pancake distributions", i.e. Gaussian in all directions except one, in which it is instead a mixture of _discrete_ Gaussians that matches many moments with a standard Gaussian. One does this using a very careful kind of rejection sampling; the technical work required is very delicate and nontrivial. Overall, this approach falls into a major line of work on lower bounds for non-Gaussian component analysis and other problems using reductions arising from it, with one of the key foundational works being DKS17.



**Questions:**

- It would be helpful to understand why in the overall proof we end up _needing_ subexponential hardness of LWE. That is, why must the parameters in appendix D be set that way? What would fail if we tried to make do with say quasipolynomial hardness? My guess is that if we tried to set the PTF degree $d = O(t/\epsilon)$ to be say logarithmic in $n$, then OPT would be too large (something like $1/\mathrm{polylog}(N)$ maybe?), but I wonder if it would still at least be nontrivial. The reduction itself seems to only take $\mathrm{poly}(m, N)$ time, so that can't be the bottleneck. Some remarks on this would be useful right in the introduction. (Note that the comparison with known SQ bounds is a slightly different matter; the question is why the _proof in this paper_ requires the strong assumption.)
- A small typo in the statement of Thm 1.2: I think it should be $OPT \leq 2^{-\log^{1-\zeta}N}$.

**Limitations:**

The authors generally do a good job contextualizing their result and its limitations. A few further remarks on how the proof crucially uses subexponential hardness of LWE would not be out of place. I am not aware of any potential negative societal impact of this work.

**Strengths And Weaknesses:**

Strengths:

This is an important hardness result for a fundamental problem in learning theory, perhaps one of the most basic ones in robust PAC learning. The SQ lower bound was already quite convincing, but a cryptographic lower bound has the benefit of not being restricted to a certain model. And while the approach in this paper closely follows the SQ proof, the additional technical work required is still considerable and showcases great mastery of the ideas and techniques involved. The paper is very well-written and clear; the figures in the supplement are particularly helpful. Considering the proofs are quite technical, I do not claim to have verified them in complete detail, but they seem correct to me.

Weaknesses:

One small weakness of this work is that it assumes _subexponential_ hardness of LWE. This is still widely-believed, but it is natural to wonder if a weaker assumption would suffice. The authors do point out how this assumption is consistent with the known SQ lower bounds, though. In terms of novelty, it definitely borrows heavily from the SQ lower bound of DK21, but again it does require significant extra work and ideas.

Overall, I think this is a very good paper that makes a significant contribution to the area.

---

> ### Author Response · Authors · 2022-08-02
> **Author Response for Reviewer yBUh**
>
> We would like to thank the reviewer for the positive assessment and address the question below:
>
> **It would be helpful to understand why in the overall proof we end up needing subexponential hardness of LWE.**
>
> An intuitive explanation for why we need a subexponential LWE assumption would be the following. Suppose we can use a weaker assumption (e.g. quasi-polynomial hardness of LWE) to obtain a non-trivial (superpolynomial) lower bound for learning LTFs with Massart noise. Then using a stronger assumption (subexponential LWE assumption), one would expect the reduction to give an even stronger lower bound – e.g., that there is no quasi-polynomial or subexponential time algorithm for learning LTF with Massart noise. Unfortunately, such a lower bound seems out of reach. For example, it is not known to hold even in the agnostic noise model and in restricted computation models (such as SQ). Therefore, it is reasonable that with our construction, the strongest possible assumption would only allow us to get a super-polynomial lower bound.
>
> At the technical level, our result can be thought of as giving a lower bound for dimension $N=n^d$ and error $\mathrm{OPT}=\exp(-\Omega(t^2/\epsilon)^2)$, where $d=\Theta(t/\epsilon)$. Due to some technical constraints, the parameter $d$ here has to be at least polynomial in $n$. In particular, we need $t^2/\epsilon=\Omega(1)$ for $\mathrm{OPT}$ to be $o(1)$, and we need $t=o(1/n^{0.5})$ for the LWE instance to hold (see Theorem 3.2 and Condition 3.1). Combining these, we get that $d=\Theta(t/\epsilon)=\Omega(1)/t\geq \mathrm{poly(n)}$. Since our dimension is already $N=n^d=n^{\mathrm{poly}(n)}$ (subexponential in $n$), any time superpolynomial in the dimension has to also be subexponential in $n$. In terms of parameters, our result is as strong as the best result we know for learning LTFs even in the *agnostic* model ([Dan16]).
>
> We also want to thank the reviewer for pointing out the typo in Thm 1.2. We will fix it in the camera-ready version.

---

> > ### Comment · Reviewer_yBUh · 2022-08-05
> > **Seeking further clarification**
> >
> > Thank you for the intuitive or "meta" explanation of why we should not expect to get anything better. That makes sense.
> >
> > In terms of the proof of Theorem 3.2 though, I'm still a bit confused on one technical point: could you further clarify exactly why Condition 3.1(iii) (namely the constraint on $t$) is needed and how it is used there? If the constraint on $t$ is justified then I agree the rest makes sense.

---

> > > ### Author Response · Authors · 2022-08-08
> > > **Clarification for (iii) of Condition 3.1**
> > >
> > > We start by explaining the meaning of the parameter $t$ in terms of the hard instance we construct to establish Theorem 3.2. As we elaborate in Section 3.2, the alternative hypothesis distribution in the hard instance ( $D_{t,\epsilon,\psi_+,B_+,\delta}^\mathrm{alternative}$  and $D_{t,\epsilon,\psi_-,B_-,\delta}^\mathrm{alternative}$) is a mixture of noisy “hidden direction discrete Gaussians”. Note that Algorithm 1 generates these noisy “hidden direction discrete Gaussians” by doing rejection sampling on cLWE (continuous LWE, see Lemma 2.4) samples, and then Algorithm 2 takes the mixture of them. These noisy “hidden direction discrete Gaussians” have the property that (see Lemma 3.5):
> > > 1) In the hidden direction $\mathbf{s}$, the distribution is pointwise close to the convolutional sum of a discrete Gaussian and a continuous Gaussian noise.
> > > 2) Moreover, in all the other directions $\perp \mathbf{s}$, the distribution is nearly independent of its value on $\mathbf{s}$ – in the sense that conditioning on any value on $\mathbf{s}$, the distribution on $\perp \mathbf{s}$ always stays pointwise close to a Gaussian.
> > >
> > > The discrete Gaussian here (as defined in Definition 2.2) has support $T\mathbb{Z}+k$, and we will call $T$ the spacing and $k$ the shift of this noisy “hidden direction discrete Gaussian”. Then the parameter $t$ corresponds to the *maximum* spacing of all the noisy “hidden direction discrete Gaussians” used in the mixture for the alternative hypothesis distribution.
> > >
> > > We now explain why the constraint $\frac{1}{t\sqrt{n}} \geq \sqrt{c\log (n/\delta)}$ is necessary here.
> > > Essentially, for Algorithm 1 to be able to generate a noisy “hidden direction discrete Gaussian” using cLWE samples, there is a limit on how large the spacing of it can be. So the constraint ensures that all the distributions used for the mixture do not have spacing exceeding this limit.
> > >
> > > We now explain at the more technical level why Algorithm 1 has this limit. Ignoring some details, the idea of Algorithm 1 is to first massage a cLWE sample $(\mathbf{x},y)$, where $\mathbf{x}\sim U([0,1]^n)$ and $y=\langle \mathbf{s},\mathbf{x}\rangle+\mathrm{noise}$, into $(\mathbf{x}’,y)$ so that $\mathbf{x}’$ is from a Gaussian and $y$ still satisfies $y=\langle \mathbf{s},\mathbf{x}'\rangle+\mathrm{noise}$. (This uses Fact A.4.) After that Algorithm 1 rescales this $\mathbf{x}'$ so the Gaussian has $\Theta(1)\mathbf{I}$ ($\mathbf{I}$ denotes the identity matrix) covariance matrix, and performs rejection sampling on $y$ to obtain the noisy “hidden direction discrete Gaussian” we desire. We note that this spacing in the support of the discrete Gaussian, essentially, comes from the periodicity of the cLWE samples in the hidden direction (one can see for example Figure 1 of [BRST21] for an illustration). The period for the periodic pattern of cLWE on the hidden direction is $1/\sqrt{n}$, since $\|\|\mathbf{s}\|\|_2=\sqrt{n}$. By Fact A.4, for the distribution of $\mathbf{x}’$ to be $(1\pm O(\delta))$-pointwise close to Gaussian, the covariance matrix of $\mathbf{x}’$ has to be at least $\sqrt{c\log (n/\delta)}\cdot\mathbf{I}$. So the rescaling will add on another multiplicative factor of $\sqrt{c\log (n/\delta)}$. This gives the limit that the spacing for noisy “hidden direction discrete Gaussian” cannot be more than $1/\sqrt{c n\log(n/\delta)}$, which is our condition (iii) in Condition 3.1.
> > >
> > > Reference:
> > >
> > > [BRST21] J. Bruna, O. Regev, M. J. Song, and Y. Tang. Continuous LWE. In STOC ’21: 53rd
> > > Annual ACM SIGACT Symposium on Theory of Computing, pages 694–707. ACM, 2021.

---

> > > > ### Comment · Reviewer_yBUh · 2022-08-08
> > > > **Thanks for the clarification**
> > > >
> > > > Interesting, thank you for the detailed clarification. So very roughly, it is a spacing constraint arising from the spacing in the original CLWE problem. That makes sense. I think what you have just written might be valuable to put in this paper as an appendix.

---

### Official Review · Reviewer_UuxY · 2022-07-11

**Rating:** 8
**Confidence:** 3
**Soundness:** 3 good
**Presentation:** 3 good
**Contribution:** 4 excellent

**Summary:**

This paper establishes the cryptographic hardness of learning halfspaces with Massart noise by reducing the Learning with Errors problem to it, which as claimed is widely believed to be subexponential-time hard. This improves the precious statistical-query hardness by [DK21], which only holds for a restricted class of SQ models. The key technique is establishing some hard instances for Massart halfspaces and then performing rejection sampling to produce them from the LWE samples. As a result, the hardness of LWE problem implies the hardness of learning LTFs with Massart noise.


**Questions:**

Can the authors address the essentialness of the subexponential hardness of LWE, and to what extent it holds?


**Limitations:**

There are no negative social impact concerns for this paper.


**Strengths And Weaknesses:**

The result shows that no polynomial-time algorithm can achieve o(\eta) error even though the optimal rate OPT is very small, implying the optimality of the existing Massart halfspaces algorithms ([DGT19,CKMY20,DKT21]). The established lower bound helps to understand the hardness of an interesting and important problem of learning Halfspaces with Massart noise, a noise type that is considered to be between the random classification noise model and agnostic model, which makes it an important progress in the theory of learning. No concerns on the clarity.

---

> ### Author Response · Authors · 2022-08-02
> **Author Response for Reviewer UuxY**
>
> We would like to thank the reviewer for the positive assessment and address the question below:
>
> **Can the authors address the essentialness of the subexponential hardness of LWE, and to what extent it holds?**
>
> Regarding **the plausibility of sub-exponential LWE assumption**, Through a quantum reduction algorithm, any subexponential time algorithm for solving the LWE in our assumption would imply a sub-exponential time quantum algorithm for approximating the shortest vector in a Lattice within a polynomial factor (the GapSVP problem). The GapSVP problem is widely believed to be not approximable for any polynomial factor in sub-exponential time. In particular, there has been much effort trying to solve this problem through different approaches (including [AG11], [BRST21], [AKS01], [MV10], etc.), but all these algorithms have exponential running time.
>
> Furthermore, LWE is a natural extension of the LPN (learning parity with noise) problem, which is also extensively studied in the learning theory community and widely believed to be hard. The currently known best algorithm for LPN is [BKW03] and has a running time of $2^{O(n/\log n)}$. Any subexponential algorithm for LWE would also imply a better algorithm for solving LPN.
>
> We also note that the same sub-exponential LWE assumption has been used in a number of works, including the recent work that we cite [GVV22] (to appear in FOCS’22).
>
> Regarding **to what extent our result hold**, we need at least a sub-exponential lower bound on the LWE to get a superpolynomial lower bound for learning LTFs with Massart noise. An intuitive explanation would be the following. Suppose we can use a weaker assumption (e.g. quasi-polynomial hardness of LWE) to obtain a non-trivial (superpolynomial) lower bound for learning LTFs with Massart noise. Then using a stronger assumption (subexponential LWE assumption), one would expect the reduction to give an even stronger lower bound – e.g., that there is no quasi-polynomial or subexponential time algorithm for learning LTF with Massart noise. Unfortunately, such a lower bound seems out of reach. For example, it is not known to hold even in the agnostic noise model and in restricted computation models (such as SQ). Therefore, it is reasonable that with our construction, the strongest possible assumption would only allow us to get a super-polynomial lower bound.
>
> At the technical level, our result can be thought of as giving a lower bound for dimension $N=n^d$ and error $\mathrm{OPT}=\exp(-\Omega(t^2/\epsilon)^2)$, where $d=\Theta(t/\epsilon)$. Due to some technical constraints, the parameter $d$ here has to be at least polynomial in $n$. In particular, we need $t^2/\epsilon=\Omega(1)$ for $\mathrm{OPT}$ to be $o(1)$, and we need $t=o(1/n^{0.5})$ for the LWE instance to hold (see Theorem 3.2 and Condition 3.1). Combining these, we get that $d=\Theta(t/\epsilon)=\Omega(1)/t \geq \mathrm{poly}(n)$. Since our dimension is already $N=n^d=n^{\mathrm{poly}(n)}$ (subexponential in $n$), any time superpolynomial in the dimension has to also be subexponential in $n$. In terms of parameters, our result is as strong as the best result we know for learning LTFs even in the *agnostic* model ([Dan16]).
>
> Reference:
>
> [AG11] S. Arora and R. Ge. New algorithms for learning in presence of errors. In Automata, Languages and Programming - 38th International Colloquium, ICALP 2011, volume 6755 of Lecture Notes in Computer Science, pages 403–415. Springer, 2011.
>
> [AKS01] M. Ajtai, R. Kumar, and D. Sivakumar. A sieve algorithm for the shortest lattice vector problem. In Proceedings on 33rd Annual ACM Symposium on Theory of Computing, STOC 2001, pages 601–610. ACM, 2001.
>
> [BKW03] A. Blum, A. Kalai, and H. Wasserman. Noise-tolerant learning, the parity problem, and the statistical query model. J. ACM, 50(4):506–519, 2003.
>
> [BRST21] J. Bruna, O. Regev, M. J. Song, and Y. Tang. Continuous LWE. In STOC ’21: 53rd Annual ACM SIGACT Symposium on Theory of Computing, pages 694–707. ACM, 2021.
>
> [MV10] D. Micciancio and P. Voulgaris. A deterministic single exponential time algorithm for most lattice problems based on voronoi cell computations. In Proceedings of the 42nd ACM Symposium on Theory of Computing, STOC 2010, pages 351–358. ACM, 2010.

---

### Author Response · Authors · 2022-08-02
**Joint Response (Part 2)**

**Computational Hardness vs SQ lower bounds:**
We would like to give our perspective on the pros and cons between SQ hardness and computational hardness. Broadly speaking, the ultimate goal of lower bounds is ruling out all algorithms for solving a specific task. Therefore, computational hardness results (under a reasonable hardness assumption) are the best for this purpose, as they rule out all efficient algorithms. However, computational hardness results for statistical problems are usually hard to come by for multiple reasons, including 1) one needs the right kind of assumption to start with, and 2) unlike worst-case computational hardness, average-case computational hardness requires more delicate techniques to preserve the background distribution. On the other hand, SQ lower bounds (or other restricted computational models) only rule out restricted classes of algorithms, but are usually easier to obtain and give us a starting point to probe into the structure of these problems. We also want to point out that algorithms like Gaussian elimination and LLL-based algorithms simply do not fall into the SQ model, and there are no logical implications that these are the only exceptions.

We also want to mention that there has recently been a line of work that “turns” SQ hardness into computational hardness, including [BB20], [Dan16], and [BRST21]. However, these efforts are still in the earlier stages as we have no general theory behind them but only case-by-case studies. From this perspective, our work is the newest development on this topic and we hope it can lead to/ be helpful for a more general theory explaining the connection between SQ hardness and computational hardness. This would answer some important high-level questions such as “why do SQ lower bounds usually reliably approximate computational hardness?”.

Reference:

[AG11] S. Arora and R. Ge. New algorithms for learning in presence of errors. In Automata, Languages and Programming - 38th International Colloquium, ICALP 2011, volume 6755 of Lecture Notes in Computer Science, pages 403–415. Springer, 2011.

[AKS01] M. Ajtai, R. Kumar, and D. Sivakumar. A sieve algorithm for the shortest lattice vector problem. In Proceedings on 33rd Annual ACM Symposium on Theory of Computing, STOC 2001, pages 601–610. ACM, 2001.

[BB20] M. S. Brennan and G. Bresler. Reducibility and statistical-computational gaps from secret leakage. In Conference on Learning Theory, COLT 2020, volume 125 of Proceedings of Machine Learning Research, pages 648–847. PMLR, 2020.

[BKW03] A. Blum, A. Kalai, and H. Wasserman. Noise-tolerant learning, the parity problem, and the statistical query model. J. ACM, 50(4):506–519, 2003.

[BRST21] J. Bruna, O. Regev, M. J. Song, and Y. Tang. Continuous LWE. In STOC ’21: 53rd Annual ACM SIGACT Symposium on Theory of Computing, pages 694–707. ACM, 2021.

[MV10] D. Micciancio and P. Voulgaris. A deterministic single exponential time algorithm for most lattice problems based on voronoi cell computations. In Proceedings of the 42nd ACM Symposium on Theory of Computing, STOC 2010, pages 351–358. ACM, 2010.

---

### Author Response · Authors · 2022-08-02
**Joint Response (Part 1)**

We would like to thank all reviewers for their uniformly positive scores and their detailed and constructive feedback. We would like to start with a brief summary of our work:

In our work, we provide the first cryptographic hardness for distribution-independent learning LTFs with Massart noise under a widely believed cryptographic assumption. Our assumption (sub-exponential LWE) is hard under a *worst-case* assumption (the GapSVP problem) through a quantum reduction, and the parameters match the best known lower bound for learning LTFs in the (harder) *agnostic* learning model ([Dan16]). Thus, our result qualitatively improves the result of [Dan16] in terms of both the noise setting and the hardness assumption.

We would like to address the following points/questions from the reviewers:

**The plausibility of sub-exponential LWE assumption:**
Through a quantum reduction algorithm, any subexponential time algorithm for solving the LWE in our assumption would imply a sub-exponential time quantum algorithm for approximating the shortest vector in a Lattice within a polynomial factor (the GapSVP problem). The GapSVP problem is widely believed to be not approximable for any polynomial factor in sub-exponential time. In particular, there has been much effort trying to solve this problem through different approaches (including [AG11], [BRST21], [AKS01], [MV10], etc.), but all these algorithms have exponential running time.

Furthermore, LWE is a natural extension of the LPN (learning parity with noise) problem, which is also extensively studied in the learning theory community and widely believed to be hard. The currently known best algorithm for LPN is [BKW03] and has a running time of $2^{O(n/\log n)}$. Any subexponential algorithm for LWE would also imply a better algorithm for solving LPN.

We also note that the same sub-exponential LWE assumption has been used in a number of works, including the recent work that we cite [GVV22] (to appear in FOCS’22).

**The necessity of subexponential hardness of LWE:**
An intuitive explanation for why we need a subexponential LWE assumption would be the following. Suppose we can use a weaker assumption (e.g. quasi-polynomial hardness of LWE) to obtain a non-trivial (superpolynomial) lower bound for learning LTFs with Massart noise. Then using a stronger assumption (subexponential LWE assumption), one would expect the reduction to give an even stronger lower bound – e.g., that there is no quasi-polynomial or subexponential time algorithm for learning LTF with Massart noise. Unfortunately, such a lower bound seems out of reach. For example, it is not known to hold even in the agnostic noise model and in restricted computation models (such as SQ). Therefore, it is reasonable that with our construction, the strongest possible assumption would only allow us to get a super-polynomial lower bound.

At the technical level, our result can be thought of as giving a lower bound for dimension $N=n^d$ and error $\mathrm{OPT}=\exp(-\Omega(t^2/\epsilon)^2)$, where $d=\Theta(t/\epsilon)$. Due to some technical constraints, the parameter $d$ here has to be at least polynomial in $n$. In particular, we need $t^2/\epsilon=\Omega(1)$ for $\mathrm{OPT}$ to be $o(1)$, and we need $t=o(1/n^{0.5})$ for the LWE instance to hold (see Theorem 3.2 and Condition 3.1). Combining these, we get that $d=\Theta(t/\epsilon)=\Omega(1)/t\geq \mathrm{poly}(n)$. Since our dimension is already $N=n^d=n^{\mathrm{poly}(n)}$ (subexponential in n), any time superpolynomial in the dimension has to also be subexponential in $n$. In terms of parameters, our result is as strong as the best result we know for learning LTFs even in the *agnostic* model ([Dan16]).

---

### Meta-Review · Area_Chair_4FtM · 2022-08-25

**Recommendation:** Accept
**Confidence:** Certain

**Metareview:**

The paper studies the hardness of PAC learning halfspaces in the presence of Massart noise. Recent work showed that the problem is hard for all algorithms using only statistical queries (the SQ model). While this class contains most learning algorithms, it does not include powerful algorithmic techniques such as Gaussian elimination or certain lattice algorithms. The present work shows similar hardness result for all algorithms under the assumption that learning with errors (LWE) is hard for subexponential-time algorithms. All reviewers appreciate the hardness result on a central problem in learning theory. Some reviewers are slightly concerned that the subexponential-time hard assumption is quite strong but there are evidences linking the problem with other problems at the foundation of lattice-based cryptography and the assumption has also been used in several other previous works.

**Award:**

No

---

### Decision · Program_Chairs · 2022-09-14

Accept